# Predicting Snow-Cover and Frozen Ground Impacts on Large Basin Runoff: Developing Appropriate Model Complexity

**Nan Wu[1,2,3,6], Ke Zhang[1,2,3,4,5,*], Amir Naghibi[6], Hossein Hashemi[6], Zhongrui Ning[2,3,6],**

**Qinuo Zhang[1], Xuejun Yi[7], Haijun Wang[7], Wei Liu[7], Wei Gao[7], Jerker Jarsjö[8]**

[1]National Key Laboratory of Water Disaster Prevention, Hohai University, Nanjing, Jiangsu, 210024, China

[2]Yangtze Institute for Conservation and Development, Hohai University, Nanjing, Jiangsu, 210024, China

[3]College of Hydrology and Water Resources, Hohai University, Nanjing, Jiangsu, 210024, China

[4]China Meteorological Administration Hydro-Meteorology Key Laboratory, Hohai University, Nanjing, Jiangsu, 210024, China

[5]Key Laboratory of Water Big Data Technology of Ministry of Water Resources, Hohai University, Nanjing, Jiangsu, 210024, China

[6]Division of Water Resources Engineering, LTH, Lund University, Lund, 22100, Sweden

[7]Hydrological Center of Shandong Province, Jinan, Shandong, 250002, China

[8]Department of Physical Geography, Stockholm University, Stockholm, 10691, Sweden

*Corresponding author:* Ke Zhang (kzhang@hhu.edu.cn)

**Abstract.** In cold regions, snow cover and seasonally frozen ground (SFG) exert substantial influence on hydrological processes, yet their effects—especially at large basin scales—remain insufficiently understood due to limited observations and process-based analysis. To address this, we extended the widely used Grid Xinanjiang (GXAJ) hydrological model by developing two physically meaningful yet computationally efficient modules: (i) the GXAJ-S model, which incorporates snowmelt processes, and (ii) the GXAJ-S-SF model, which additionally accounts for freeze-thaw cycles of SFG. These modules strike a balance between physical representation and simplicity, making them applicable in data-sparse cold regions. Model performance was evaluated using multi-source remote sensing/reanalysis data and observed daily runoff, enabling a systematic investigation of how snow and SFG jointly regulate key hydrological processes. The results demonstrate that: (1) including both snowmelt and freeze-thaw processes significantly improves runoff simulation, especially during cold seasons; (2) snow dynamics directly modulate the development of soil freeze-thaw cycles, thereby altering the hydrothermal state of the vadose zone; and (3), the inclusion of the SFG module in the model variant that already accounted for snowmelt, increased predicted surface runoff by 39–77% during cold months, reduced evapotranspiration by approximately 85%, and substantially modified interflow processes, particularly during the early spring thaw period. These findings provide quantitative evidence of the critical role of SFG in shaping the

seasonal hydrological regime of large cold-region basins. Moreover, the modular and transferable design of the snow and SFG components allows for straightforward integration into other hydrological models, offering a valuable tool for hydro-climatic assessments and water resource management in mountainous regions under changing climate conditions.

**Keywords:** Frozen ground, Snow, Hydrological Modeling, Cold Regions. Climate change

## 1. Introduction

Seasonally Frozen Ground (SFG) has significant implications for the energy balance and water equilibrium of the land surface, which in turn affects ecosystems, hydrologic processes, soil properties, and biological activity worldwide. Seasonal freezing occurs across extensive areas, with approximately 25% of the Northern Hemisphere's land surface experiencing seasonal topsoil freezing in permafrost regions, i.e., the active layer, and an additional 25% outside the permafrost zone (Zhang et al., 2003). While the hydrological impacts of permafrost thaw and active layer changes have been extensively investigated over the past decade (Ford and Frauenfeld, 2016; Qin et al., 2017; Song et al., 2022; Streletskiy et al., 2015), the hydrological impacts of SFG in permafrost-free regions have received less attention (Ala-Aho et al., 2021). The hydrological response to SFG is controversial and appears to be highly site- and time-specific (Appels et al., 2018). A systematic review by Ala-Aho et al. (2021) concluded that the impact of SGF on runoff processes is profound in many small-scale applications.

However, large knowledge gaps remain, not least regarding the complex and less clear responses on larger scales for which the presence and absence of SFG may show considerable

spatial variation. The possible, spatially complex impacts of SFG on runoff in large basins may furthermore vary considerably within the year (Song et al., 2022). Shiklomanov (2012) similarly noted that despite the large scale and significant importance of SFG in cold regions, it has not received much attention due to the lack of long-term observational time series. Additionally, climate change is expected to alter frozen ground conditions and extent (Wang et

al., 2019), increasing the frequency of freeze-thaw events in cold regions (Venäläinen et al., 2001). Thus, understanding the hydrological impacts of SFG under a warming climate, where permafrost is being transformed into SFG, is becoming increasingly important.

It is generally accepted that frozen ground, whether seasonally frozen or permafrost, constrains hydrological interactions to some extent. However, the hydrological response within

75 permafrost regions differs significantly from areas where only the surface soil freezes seasonally. Permafrost extends deeply into the subsurface, impeding or even completely preventing deep groundwater runoff (Walvoord et al., 2012), leading to shallow groundwater runoff and rapid surface water runoff during snowmelt if the active layer of permafrost has not yet thawed (Hinzman et al., 1991). In contrast, the effects of SFG typically remain shallow in

depth, increasing surface water runoff and reducing groundwater recharge during snowmelt if the topsoil is frozen (Ireson et al., 2013). This suggests that SFG disrupts surface-subsurface hydraulic connectivity in winter and spring while increasing hillslope runoff into the stream channels (Covino, 2017). This study focuses on SFD, which, at the regional scale, can serve as

a crucial indicator of climate change and frozen ground conditions in cold regions.

SFG regions generally experience seasonal snow cover, which significantly influences the soil freeze-thaw process. Due to the low thermal conductivity, high latent heat of melting, and high albedo of snow, changes in snow cover substantially alter the impact of air temperature on the thermal state of the soil (Goncharova et al., 2019), thereby affecting the soil freeze-thaw dynamics (Biskaborn et al., 2019). In areas of thin or transient snow cover in the SFG regions,

thermal coupling between the ground and the atmosphere is more likely to increase the frequency and intensity of soil freezing while potentially reducing the duration of the freeze (Fuss et al., 2016). Consequently, soil in these regions may freeze more frequently and deeply but thaw more quickly due to weaker snowpack insulation. The seasonal effect of deep snowpack on ground temperatures depends on the thermal history of the ground, air

temperature, and solar radiation that isolates the ground from the atmosphere (Maurer and Bowling, 2014). In a warming climate, a decrease in late-season snowpack may lead to increased soil freezing (Hardy et al., 2001). This phenomenon, termed "soil cooling in a warm world" (Groffman et al., 2001), emphasizes the complex effects of climate change on soil freezing and thawing processes. Therefore, the hydrological impacts of snow and SFG should

be considered together as the two processes interact (Qi et al., 2019).

      The impact of SFG and snow cover on hydrological processes can be simulated using process-based hydrological models (Gao et al., 2022; Qi et al., 2019). Physical process-based cold regions hydrological models such as the Geomorphology-Based Eco-Hydrological Model (GBEHM) (Yang et al., 2015), the Water and Energy Budget-based Distributed Hydrological

Model (WEB-DHM) (Wang et al., 2009), the Variable Infiltration Capacity (VIC) model (Liang et al., 1996), and the Cold Region Hydrological Model (CRHM) (Pomeroy et al., 2007) have been developed to assess various hydrological impacts of SFG and snow cover (Jafarov et al., 2018; Qi et al., 2016; Walvoord et al., 2019). While these models offer rigorous physical interpretations, they require a number of high-quality input data, and are hindered by parameterization complexities that induce simulation uncertainties (Gao et al., 2018), and exhibit slow computational speeds. Moreover, challenging climate and environmental conditions in cold regions pose difficulties for field observations, exacerbating local parameterization challenges.

Conventional hydrological models such as SWAT (Arnold et al., 1995), HBV model (Krysanova et al., 1999), TOPMODEL (Beven and Kirkby, 1979), and Xinanjiang model (Zhao, 1984) predominantly focus on soil moisture conditions, neglecting the impacts of snowmelt and soil freeze-thaw processes. However, the soil freeze-thaw cycle traverses runoff processes, including infiltration, evaporation, and water migration, constituting a pivotal aspect of the hydrological cycle in cold regions (Guo et al., 2022). Although efforts have been made to integrate soil freeze-thaw processes into conventional hydrological models (Ahmed et al., 2022; Huelsmann et al., 2015; Kalantari et al., 2015), most of them are based on changing relevant parameters and are unable to reflect the key physical processes in cold regions. Snow cover and SFG exhibit significant spatiotemporal heterogeneity and are influenced by numerous interconnected factors. The translation of point/slope-scale frozen processes into their basin-scale hydrological implications remains largely unexplored (Gao et al., 2022). Furthermore,

there is also a lack of lack of mechanistic and quantitative studies on how snow and SFG affect key hydrological processes.

The Tibetan Plateau, the source region for many major rivers in Asia, provides water for billions of people and downstream ecosystems, earning the title "Asian Water Tower" (Immerzeel et al., 2010). The cryosphere of the Tibetan Plateau, consisting primarily of snow, permafrost, and glaciers (Qi et al., 2019), is highly sensitive to climate change. Seasonal snow cover and frozen ground significantly influence the hydrological processes in cold alpine regions, exhibiting pronounced intra-annual regulatory effects (Gao et al., 2023). Consistent with that, Pomeroy et al. (2007) recommended considering the coupling of seasonal freeze-thaw cycles with precipitation (snowfall) as a potential primary control on hydrological processes. The Xinanjiang model and its derivatives are considered the most commonly used practical flood forecasting models in China (Yao et al., 2014), with significant experience accumulated in operational flood forecasting (Chen et al., 2023); However, its adaptability in cold regions is relatively poor because it does not account for the influence of snow cover and frozen ground on the hydrological process.

To address these limitations, this study develops two enhanced hydrological models based on the Gridded Xinanjiang (GXAJ) framework. The enhancements are achieved through additions of a snowmelt-enhanced module (GXAJ-S) and by the further addition of a seasonally frozen ground module (GXAJ-S-SF). A main innovation lies in explicitly coupling the physical mechanisms of snowmelt and freeze–thaw processes into a distributed hydrological model. In particular, SFG influences the partitioning of water into ice and liquid phases, modifies the

vadose and humus layer thickness used in runoff generation, and thereby alters seasonal runoff dynamics. The spatial distribution of SFG is strongly influenced by snow cover, and together, they regulate evapotranspiration and soil water availability. A related main novel aspect introduced in this work is how the additional processes are accounted for, taking advantage of the modular model design in a three-step manner (i.e., considering (i) the baseline model with no snow/SFG, (ii) adding the snow module, and (iii) further adding the SFG module), with the modules being grounded in well-established physical principles. This allows for increasing the complexity while transparently checking the model performance of each step. In particular, any potential increases in model performance can then be related to the dynamics created by the additional module (and the corresponding account for a new process). To the best of our knowledge, this has not been done earlier in large cold region basins. This is because previous comparisons have regarded models that differ in either structure (Gao et al., 2018; Li et al., 2018b; Song et al., 2022), or structure as well as complexity (e.g., Ahmed et al., 2022; Gao et al., 2018; Guo et al., 2022). In both cases, differences in model performance may then partly be due to fundamental structural or parametrization differences between models, introducing uncertainty in how performance may be linked to complexity (i.e., inclusion or omission of processes), which is avoided by the current approach. We aim to provide scientific and practical guidance for the appropriate level of model complexity needed for large-scale cold region hydrological applications, especially where data limitations persist.

# 2. Methodology

## 2.1 Cold region runoff mechanisms

The critical importance of ground freezing in the runoff generation of cold regions lies in the transformation of pre-existing water in soil pores into ice, which inhibits vertical water connectivity (Ala-Aho et al., 2021). Consequently, in areas with frozen ground, runoff processes are influenced not only by precipitation and soil moisture but also by ground freezing conditions driven by temperature variations (Wang et al., 2017). Based on the dynamic changes associated with seasonal freeze-thaw cycles and snow accumulation-melt dynamics, the runoff generation process are divided into four stages (Guo et al., 2022): initial freezing stage (IFS), stable freezing with snow stage (SFS-S), initial thawing stage (ITS), and complete thawing stage (CTS) (Fig. 1).

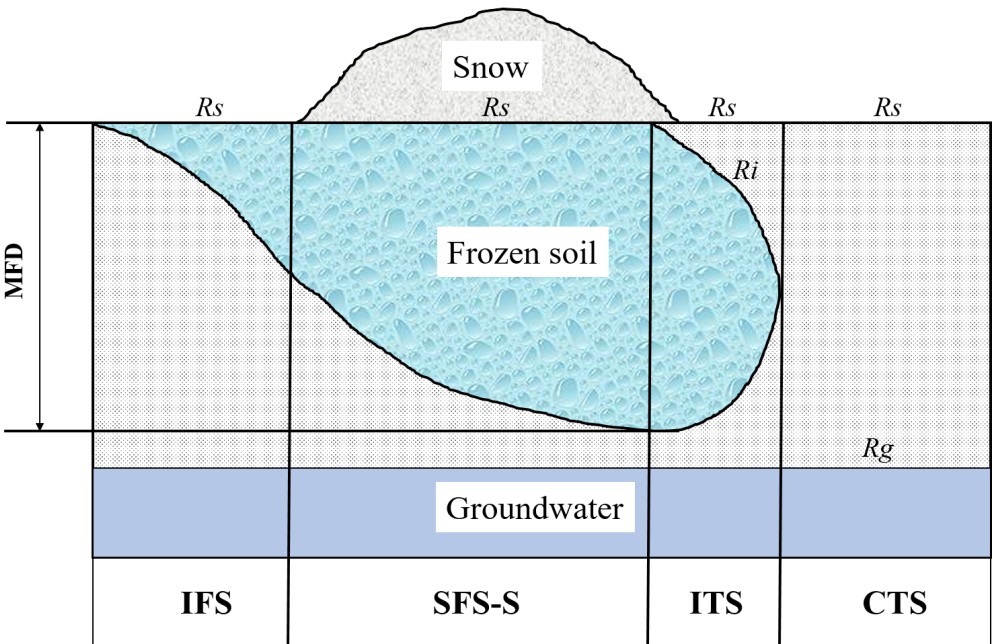

**Figure 1.** Runoff generation model in seasonally frozen ground/snow regions. $R_s$, $R_i$, and $R_g$ represent surface water runoff, interflow, and groundwater runoff, respectively; MFD means maximum seasonal frozen ground

depth.

   **i)** During the IFS, temperatures are low, but no snowfall occurs. The ground freezes from the surface downwards (Thomas et al., 2009), significantly inhibiting the evaporation of soil moisture into the air and making it difficult for vegetation to absorb it. Due to the frozen surface layer, groundwater recharge is restricted. The precipitation during this stage mainly generates

surface water runoff ($R_s$), which becomes the primary runoff component.

   **ii)** Persistent low temperatures cause the depth of the frozen ground to increase while snow accumulates on the surface, maintaining the frozen state. The snow protects the cold ground from solar radiation despite warmer temperatures (Rush and Rajaram, 2022) until the snow completely melts. In the SFS-S, groundwater remains active beneath the frozen layer (Gao et

al., 2022), soil evapotranspiration is nearly zero, and $R_s$ generated by snowmelt or rainfall remains the main runoff component.

   **iii)** During the ITS, as the temperature continues to rise and snow completely melts, the surface frozen ground begins to thaw, receiving substantial inputs from precipitation and snowmelt. During this stage, vegetation transpiration is very limited, and soil evaporation

occurs only in the thawed surface layer. As a result, the surface layer easily saturates, generating saturation-excess runoff $R_s$. With increasing thaw depth, interflow ($R_i$) appears above the thaw front. Runoff during this stage primarily consist of a mix of $R_s$ and $R_i$.

   **iv)** In the CTS, the atmospheric and soil layers restore vertical connectivity. Increased rainfall events replenish groundwater, and evapotranspiration gradually increases. Runoff

processes in this stage include $R_s$, $R_i$, and groundwater runoff ($R_g$).

In SFG/snow covered regions, precipitation and snowmelt are the primary sources of runoff. Temperature influences the seasonal freeze-thaw cycles of snow and frozen ground, and their interaction further affects soil water/ice content and evapotranspiration. Lower elevations generally experience higher temperatures compared to higher elevations, and south-facing slopes are generally warmer than north-facing slopes. Such local to regional temperature differences cause spatial variability in runoff, with transitions in runoff components across different freeze-thaw stages forming the fundamental runoff patterns in SFG regions.

## 2.2 Modeling approach

The GXAJ model (Yao et al., 2012) uses the concept of a saturated runoff mechanism, meaning that during rainfall, runoff will only occur once the soil water storage reaches the field capacity, with all incoming water being absorbed by the soil before that point. In the GXAJ model, the tension water storage capacity ($W_M$) (mm) of any grid cell is determined by the geomorphological features and underlying surface conditions such as soil and vegetation (Stephens, 1996; U. S. Department of Algriculture, 2002). The potentially uneven distribution of $W_M$ within a grid cell is not considered. The measured precipitation in the computation period is first adjusted by subtracting the corresponding period's evapotranspiration, vegetation canopy interception, and river precipitation. Then the upstream inflow is considered to check if it can replenish the soil moisture in the current grid cell. This results in an effective precipitation ($P_e$) that is used for runoff ($R$) calculation.

The runoff ($R$) from a grid cell is divided into three components: surface runoff $R_s$, interflow $R_i$, and groundwater runoff $R_g$. The GXAJ model assumes that the surface soil of the

capillary zone is humus layer (determined by geomorphological features and soil, vegetation, and other surface conditions) (Li et al., 2004). The bottom of the humus layer is considered to be "relatively impermeable." A portion of the runoff generates $R_i$ in the humus layer, while another part infiltrates further to produce $R_g$. When the free water in the humus layer becomes saturated, surface runoff occurs. Similarly, the uneven distribution of free water storage capacity ($S_M$) within the grid cell is not considered.

The GXAJ model calculates evapotranspiration using a three-layer model. The soil is divided into upper, lower, and deep layers, with each layer having corresponding tension water storage capacities of $W_{UM}, W_{LM}$ and $W_{DM}$ (mm). When calculating actual evapotranspiration in a grid cell, canopy interception is evaporated based on its evapotranspiration capacity. If the interception is less than the evapotranspiration capacity, the three-layer model is used. The calculation principle of the three-layer evapotranspiration model is as follows: The upper layer evaporates according to its capacity. If the upper layer's water content is insufficient, the remaining evapotranspiration capacity is used by the lower layer, which evaporates proportionally to the lower layer's water content and inversely to its water storage capacity. The ratio of the calculated lower layer evapotranspiration to the remaining evapotranspiration capacity must not be less than the deep-layer evapotranspiration coefficient ($C$). Otherwise, the deficit is replenished by the lower layer's water content, and when the lower layer is insufficient, it is supplemented by the deep layer's water content.

In summary, the GXAJ model partitions runoff into three components, i.e., $R_s$, $R_i$, and $R_g$, by calculating the tension water storage capacity ($W_M$) in the vadose zone and the free water

storage capacity ($S_M$) in the humus layer (the spatial distribution is shown in Fig. S1). The $W_M$

determines whether a grid cell generates runoff and the runoff volume (i.e., saturation-excess

runoff), while the free water content of the surface soil differentiates the runoff components

into $R_i$ and $R_g$. When the free water content reaches saturation, $R_s$ is produced, as illustrated in

Fig. S2 (a). For actual evapotranspiration calculation, the soil within each grid cell is divided

into three layers: upper, lower, and deep, with corresponding soil moisture and

evapotranspiration labeled as $W^u$, $W^l$, and $W^d$, and $E^u$, $E^l$, and $E^d$, respectively, as shown in Fig.

S2 (b). Confluence processes follow the calculation order between grids, sequentially routing

various water sources to the watershed outlet. For details, refer to Yao et al. (2009).

However, the original GXAJ model does not account for the impacts of snow cover

and freeze-thaw processes on runoff generation; studies have shown that this model is not

suitable for seasonally cold regions (Yao et al., 2009, 2012). To address this, we here

introduce the snowmelt runoff process (SNOW17) and the freeze-thaw cycle processes

into the GXAJ model, investigating if and to which extent the related expanded GXAJ-S

model and GXAJ-S-SF model could better represent cold region hydrological processes

(Fig. 2). Specifically, these processes explicitly account for the accumulation and melting

of seasonal snow, as well as the spatiotemporal variations in soil freeze-thaw depth, using

grid-based temperature and precipitation inputs. The SNOW17 model (Anderson, 1973)

was chosen for snowmelt runoff calculation due to its minimal input requirements and

clear representation of the most critical physical processes within the snowpack.

Additionally, the Stefan equation was employed to predict seasonal soil freeze and thaw

depths (Peng et al., 2017). The Stefan equation is widely used in conjunction with process-

based models due to its simplicity and flexibility (Kurylyk, 2015).

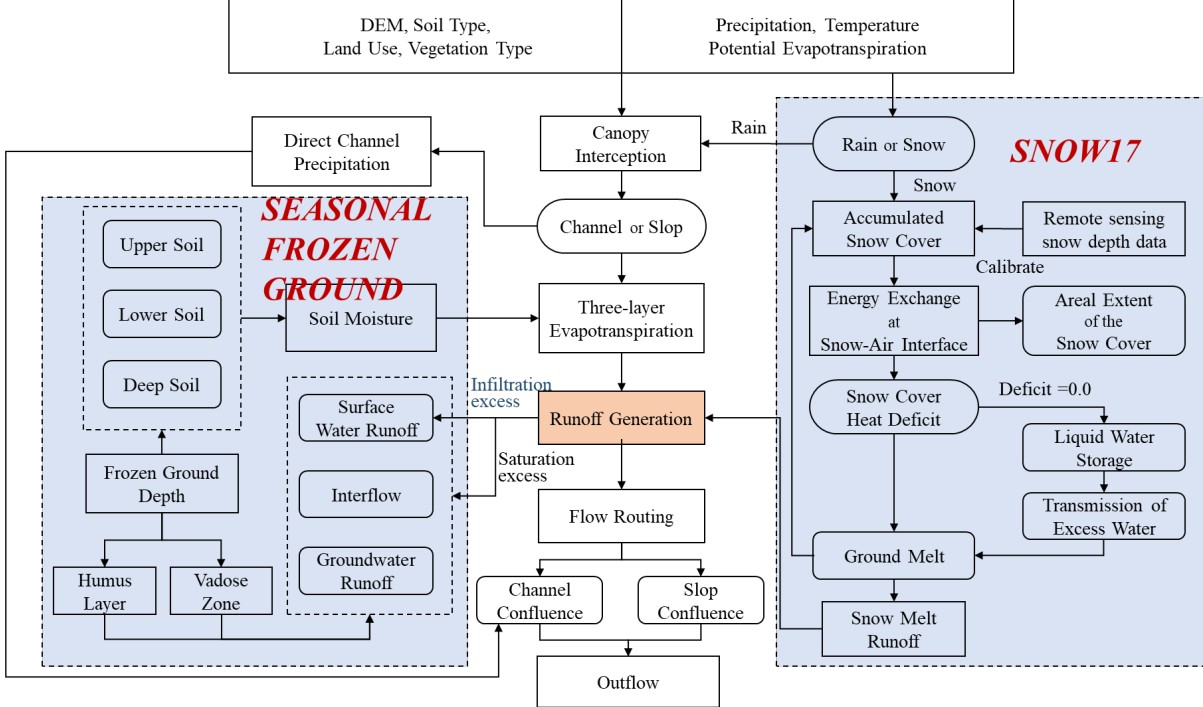

**Figure 2.** The schematic framework of the GXAJ-S-SF model.

## 2.2.1 Snow accumulation and melting runoff

Before snowfall occurs, if ground temperatures remain below freezing (0°C) for an extended period, the soil is subject to freezing (IFS) conditions. In related snow accumulation phases, as long as the snow cover remains relatively thin, most solar radiation is reflected by the snow cover due to its high albedo, while it yet does not insulate the ground, due to insufficient thickness. In contrast, thick snow covers, with their low thermal conductivities, can completely isolate the ground from the surrounding air temperature (Rush and Rajaram, 2022). Research has proposed a snow depth threshold of 30-40 cm (Hill, 2015), above which air temperature is not expected to affect ground temperature. At the lowest negative accumulated temperature, the maximum frozen depth is reached, with soil water retained as ice. As

temperatures rise, the surface snow begins to melt first (Fig. S3).

The SNOW17 model (Anderson, 1973), developed as part of the National Weather Service

river forecast system in the United States, was used for snowmelt prediction. The model

description in this section is adapted from the latest references of the model (Anderson, 2006).

The SNOW17 is an empirical lumped model that uses average daily temperature as the sole

index to simulate snow accumulation, heat storage, snowmelt, liquid water retention, and

meltwater transmission, determining energy exchange at the snow-air interface based on

empirical relationships (He et al., 2011). The model outputs are snow depth and runoff time

series. The snow accumulation and melting amount for each grid cell are calculated based on

the snow-covered area. The SNOW17 model calculates snowmelt with and without rainfall,

producing the total runoff during the snow cover period ($O_s$, mm).

The snow surface melting equation with rainfall is:

$$M_r = \sigma \cdot \Delta t_p \cdot [(T_a + 273)^4 - 273^4] + 0.0125 \cdot P \cdot f_r \cdot T_r + 8.5 \cdot UADJ$$
$$\cdot \left(\Delta t_p / 6\right) \cdot [(0.9 \cdot e_{sat} - 6.11) + 0.00057 \cdot P_a \cdot T_a] \quad (1)$$

where, $M_r$ is the melt during rain-on-snow time intervals (mm), $\sigma$ represents the

Stefan-Boltzman constant (6.12·10$^{-10}$ mm/°K/hr), $\Delta t_p$ is the time interval of precipitation data

(hour), $T_a$ is the air temperature (°C), 273 represents 0°C on the Kelvin scale, $f_r$ is the

fraction of precipitation in the form of rain, $T_r$ is the temperature of rain (°C), $UADJ$

represents the average wind function (mm/mb/6 hr), and $e_{sat}$ and $P_a$ are saturated vapor

pressure at $T_a$ (mb) and atmospheric pressure (mb), respectively.

The snow surface melting equation without rainfall is:

$$M_{nr} = M_f \cdot (T_a - MBASE) \cdot \frac{\Delta t_p}{\Delta t_t} + 0.0125 \cdot P \cdot f_r \cdot T_r \qquad (2)$$

where, $M_{nr}$ is the melt during non-rain periods (mm), $M_f$ is the melt factor (mm/°C/$\Delta t_t$), $\Delta t_t$ is the time interval of temperature data (hours), and $MBASE$ is the base temperature (°C).

Most soil moisture exists in the form of solid ice, and the presence of frozen ground obstructs the infiltration of snowmelt water, resulting in surface water runoff ($R_s^*$, mm) as shown in Fig. S3 (a). In the presence of snow cover, soil moisture evaporation is generally impeded. The snow cover prevents the evaporation of moisture from the soil surface, while moisture on the snow surface is released into the atmosphere through sublimation (i.e., snow surface evaporation) as described by the SNOW17 model. Therefore, soil moisture evaporation is typically restricted under snow cover. Additionally, the frozen ground beneath the snow prevents soil moisture from being released into the atmosphere through evaporation, further limiting soil moisture evaporation. The soil moisture status at this time is shown in the Fig. S3 (b).

**2.2.2 Freeze-thaw process**

The GXAJ-S-SF model employed the Stefan equation to estimate the approximate solution for the freeze-thaw depth. The Stefan equation is a temperature index-based freeze-thaw algorithm that assumes the sensible heat in soil freeze-thaw simulations can be neglected (Xie and Gough, 2013):

$$SFD = \sqrt{\frac{2 \cdot 86400 \cdot K_f \cdot F}{L \cdot \omega \cdot \rho}} \qquad (3)$$

where $SFD$ is the freeze-thaw depth (cm), $K_f$ is the thermal conductivity of the soil (W(mK)$^{-1}$), $F$ is the surface freezing-thawing index, with the freezing index being the

cumulative negative ground temperature during freezing and the thawing index being the cumulative positive ground temperature during thawing. $L$ is the latent heat of fusion for ice $(3.35\times10^5 Jkg^{-1})$, $\omega$ is the water content, and $\rho$ is the bulk density of the soil (kg-m$^{-3}$). We set the thermal conductivity to 2W(mK)$^{-1}$, the water content $\omega$ to 0.12 (as a fraction of dry soil weight), and the bulk density $\rho$ to 1000 kg-m$^{-3}$ (Gao et al., 2022). Due to the lack of ground temperature data, a conversion factor was used to transform air temperature into ground temperature. During the freezing period, this factor was 0.6, while during thawing, it was assumed that ground temperature equaled air temperature (Gisnas et al., 2016).

To account for the insulating effect of snow cover on frozen ground, a threshold of 30 cm was used: if the snow depth exceeded 30 cm (Hill, 2015), the air temperature effect on frozen ground was ignored, regardless of whether low temperatures caused soil freezing or high temperatures caused thawing. If the snow depth was below this threshold and the snow cover duration ranged between 60-140 days (Wu et al., 2024), the snow depth variable was added to the Stefan equation (Wang & Chen, 2022):

$$SFD^* = \sqrt{\frac{2 \cdot 86400 \cdot K_f \cdot F}{L \cdot \omega \cdot \rho}} / \sqrt[3]{ASD} \tag{4}$$

where *ASD* is the average snow depth.

In this study, the Stefan equation was driven by distributed temperature data, enabling us to simulate the soil freeze-thaw processes for each grid cell. The spatiotemporal variation of frozen soil depth affects runoff components, including soil water/ice, and soil evapotranspiration. We distinguish between four different possible type cases regarding associated runoff generation, each of which is associated with different modeling routines:

**Case (a)**: When the surface soil is frozen, as shown in Fig. S4 (a), rainfall and snowmelt primarily generate surface water runoff ($R_s^*$). Soil water/ice content is shown in Fig. S5 (a). When the soil is in a frozen state, soil moisture cannot evaporate because the frozen ground forms an ice layer that prevents upward moisture evaporation.

**Case (b):** When the surface soil has thawed and the thawing depth is less than the depth of the humus layer (Fig. S4 (b)), the surface soil moisture exists in the form of liquid water. In this case, the thawed soil layer is considered to be the "new" vadose zone and the humus layer. The bottom of the thawed layer (impermeable layer) generates interflow ($R_i^*$), and since the thawed soil layer is relatively thin, surface saturation runoff ($R_s^*$) is easily generated:

$$R = P_e + W_0^* - W_M^* \tag{5}$$

$$R_i^* = K_i \times S^* \tag{6}$$

$$R_s^* = R + S^* - S_M^* \tag{7}$$

where $P_e$ is the net rainfall during the period used for runoff calculation, mm; $W_0^*$ is the initial soil moisture content of the thawed soil layer, mm; $W_M^*$ is the tension water storage capacity of the thawed soil layer, $S^*$ is the free water content in the thawed surface soil, $K_i$ is the outflow coefficient of the surface soil free water content to the interflow, and $S_M^*$ is the free water storage capacity in the thawed surface soil.

Among them, the variables with * represent relevant variables in the thaw layer, and their values are related to the temporal and spatial changes of the frozen soil depth:

$$W_0^* = \frac{(L_a - SFD^*)}{L_a} W_0 \tag{8}$$

$$S_0^* = \frac{(L_h - SFD^*)}{L_h} S_0 \tag{9}$$

$$W_M^* = \frac{(L_a - SFD^*)}{L_a} W_M = (L_a - SFD^*) \times (\theta_{fc} - \theta_{wp}) \tag{10}$$

$$S_M^* = \frac{(L_h - SFD^*)}{L_h} S_M = (L_h - SFD^*) \times (\theta_s - \theta_{fc}) \tag{11}$$

$L_a$ and $L_h$ are the thickness of the vadose zone and humus layer, respectively, which can

be estimated by a soil moisture constant corresponding to the terrain index and soil type, mm;

$W_0, S_0, W_M, S_M$ are the corresponding water contents when there is no frozen soil (Yao et al.,

2009).

At this time, there are two scenarios for soil moisture (Figs. S5 (b1) and S5 (b2)). As

shown in Fig. S5 (b1), when the bottom of the thawed layer is in the upper soil, the upper soil

moisture includes both liquid water $W_w^u$ and frozen solid ice $W_i^u$. Evapotranspiration only

affects the liquid water in the upper layer, while evapotranspiration in the lower and deep layers

is zero. When $W_w^u$ is sufficient; the upper layer evapotranspiration $E^u$ is:

$$E^u = K \times E_M \tag{12}$$

where $K$ is the evapotranspiration coefficient, and $E_M$ is the water surface evaporation during

the period, mm.

When the bottom of the thawed layer reaches the lower soil layer (Fig. S5 (b2)), the entire

upper soil is thawed, and the lower soil contains both solid and liquid water. At this time, the

thawed lower layer is also affected by the evapotranspiration process. If the upper layer is dry

and the lower thawed soil moisture content $W_w^l$ is sufficient, the upper and lower layers are

affected by the evapotranspiration, $E^u$ and $E^l$, respectively:

$$E^u = K \times E_M \tag{13}$$

$$E^l = (K \times E_M - E^u) \times W_w^l / W_{LM}^* \tag{14}$$

where $W_{LM}^*$ is the tension water storage capacity of the lower thawed soil layer (mm), which is related to the proportion of the lower thawed soil layer to the whole lower layer:

$$W_{LM}^* = \frac{(L_M - SFD^*)}{L_M} W_{LM} = (L_M - SFD_{LM}^*) \times (\theta_{fc} - \theta_{wp}) \tag{15}$$

$L_M$ represents the depth of the lower layer soil, $SFD_{LM}^*$ is the frozen depth of the lower layer soil.

**Case (c):** When the humus layer is completely thawed (Fig. S4 (c)), the thawed soil layer is considered to be the "new" vadose zone. According to the original GXAJ model's runoff generation theory, the bottom of the humus layer (relatively impermeable layer) generates $R_i$. At this time, there are two components of interflow: $R_i$ and $R_i^*$. When the humus layer is saturated, $R_s$ is generated. It is noteworthy that no groundwater runoff is generated throughout the frozen soil period.

$$R = P_e + W_0^* - W_M^* \tag{16}$$

$$R_i = K_i \times S \tag{17}$$

$$R_i^* = K_g \times S \tag{18}$$

$$R_s = R + S - S_M \tag{19}$$

where $S$ is the free water content in the surface soil $L_h$, $K_g$ is the outflow coefficient of $S$ to groundwater runoff, $S_M$ is the free water storage capacity of $L_h$.

Soil moisture is present in two scenarios, with the bottom of the thawed layer appearing in the lower soil (Fig. S5 (c1)) and the deep soil (Fig. S5 (c2)). The evapotranspiration calculation for the first scenario (Fig. S5 (c1)) is consistent with Fig. S5 (b2). When the bottom of the thawed layer deepens to the deep soil (Fig. S5 (c2)), if the soil moisture in the upper and

lower layers is also insufficient, it is necessary to calculate the deep layer thawed soil

evapotranspiration $E^d$:

$$E^u = K \times E_M \tag{20}$$

$$E^l = (K \times E_M - E^u) \times W_w^l / W_{LM} \tag{21}$$

$$E^d = C \times (K \times E_M - E^u) - E^l \tag{22}$$

where $C$ is the deep-layer evapotranspiration coefficient.

**Case (d):** Until the frozen soil is completely thawed, as shown in Fig. S5 (d), runoff

calculation is performed according to the original GXAJ model (Fig. S2).

**2.2.3 Model parameters and calibration**

The original GXAJ model operates on a daily time step and includes 18 parameters (Table

1), 13 of which are spatially distributed and estimated based on vegetation type, soil texture,

and topographic characteristics. The remaining 5 parameters are calibrated and derived from

long-term empirical experience with the model. To incorporate snow and freeze-thaw processes

without compromising model parsimony, we adopted a flexible approach by integrating the

SNOW17 snowmelt module and a simplified freeze-thaw cycle module into the GXAJ model.

The SNOW17 module contains 10 parameters in total (Table 2), of which only 4 are key

parameters requiring calibration. These core parameters can be initially estimated using

empirical guidelines (Anderson, 2002), and the remaining secondary parameters, which have

limited influence on model performance, can be assigned based on general climate

characteristics of the study area with minimal adjustment.

**Table 1.** GXAJ model parameters and their descriptions.

| Module | Parameter | Description | Source or Calibration |
|---|---|---|---|
| Canopy interception | $LAI_{max}$ | Maximum *LAI* for the vegetation in a year | Derived from LDAS based on vegetation types |
| | $h_{lc}$ | Height of vegetation (m) | Derived from LDAS based on vegetation types |
| Channel precipitation | $W_{ch}$ | Channel width within a cell (km) | Estimated based on measured cross sections |
| Evapotranspiration | $W_{UM}$ | Tension water capacity of upper layer (mm) | Estimated based on initial $W_M$ |
| | $W_{LM}$ | Tension water capacity of lower layer (mm) | Estimated based on initial $W_M$ |
| | $C$ | Evapotranspiration coefficient of deeper layer | Estimated based on *LAI* and $h_{lc}$ of vegetation |
| | $K$ | Ratio of potential evapotranspiration to pan evaporation | Calibrated (prior range: $0-1$) |
| Runoff generation | $W_M$ | Tension water capacity (mm) | Estimated using $\theta_{fc}, \theta_{wp}$ and vadose zone thickness |
| | $\theta_s$ | Saturated moisture content | Obtained from literature based on soil types |
| | $\theta_{fc}$ | Field capacity | Obtained from literature based on soil types |
| | $\theta_{wp}$ | Wilting point | Obtained from literature based on soil types |
| | $S_M$ | Free water capacity (mm) | Estimated using $\theta_s, \theta_{fc}$ and humus layer thickness |
| | $K_i$ | Outflow coefficient of free water storage to interflow | Estimated based on soil properties |
| | $K_g$ | Outflow coefficient of free water storage to groundwater | Estimated based on soil properties |
| Flow routing | $C_i$ | Recession constant of interflow storage | Calibrated (prior range: $0-1$) |
| | $C_g$ | Recession constant of groundwater storage | Calibrated (prior range: $0-1$) |
| | $C_s$ | Recession constant in the lag and route technique | Calibrated (prior range: $0-1$) |
| | $L_{ag}$ | Lag time | Calibrated (prior range: $\geqslant 0$) |

**Table 2.** SNOW17 model parameters and their descriptions.

| | Parameter | Description | Calibration or Fixed Value |
|---|---|---|---|
| Major parameters | *SCF* | Snow correction factor, or gage catch deficiency adjustment factor | 0.7 - 1.6 (calibrated) |
| | *MFMAX* | Maximum solar melt factor during non-rain periods, assumed to occur on June 21 (mm·°C-1·6hr-1) | 0.5 - 2.0 (calibrated) |
| | *MFMIN* | Minimum solar melt factor during non-rain periods, assumed to occur on December 21 (mm·°C-1·6hr-1) | 0.05 - 0.49 (calibrated) |
| | *UADJ* | The average wind function during rain-on-snow periods (mm·mb$^{-1}$) | 0.03 - 0.19 (calibrated) |
| Minor parameters | *NMF* | Maximum negative melt factor (mm·mb$^{-1}$·6hr$^{-1}$) | 0.45 (fixed value) |
| | *TIPM* | Antecedent temperature index parameter | 0.9 (fixed value) |
| | *PXTEMP* | The temperature that separates rain from snow (˚C) | 0 (fixed value) |
| | *MBASE* | Base temperature for snowmelt computations during non-rain periods (˚C) | 0 (fixed value) |
| | *PLWHC* | Percent liquid water holding capacity for ripe snow (decimal fraction) | 0.1 (fixed value) |
| | *DAYGM* | Constant daily amount of melt which takes place at the snow-soil interface whenever there is a snow cover (mm·day-1) | 0.7 (fixed value) |

To prevent overfitting and ensure model improvements stem from enhanced physical

process representation rather than increased parameter freedom, the SNOW17 model was

first run independently. Remotely sensed snow depth data were used as observational

constraints to calibrate the four major snow parameters, ensuring that snow simulations

aligned with observed snow dynamics. Once calibrated, the snowmelt model was coupled

with the GXAJ model to form the GXAJ-S model. Importantly, this integration did not

introduce any new parameters to the original GXAJ structure. The freeze-thaw process was

implemented using a simplified module based on the Stefan equation with five empirical

parameters (see Section 2.2.2). These were used to adjust soil moisture availability and

runoff generation under frozen ground conditions. The resulting GXAJ-S-SF model thus

includes only a limited number of additional parameters, all of which have clear physical

interpretations and are easy to calibrate, making the model especially suitable for data-scarce

regions.

All model configurations (GXAJ, GXAJ-S, GXAJ-S-SF) were calibrated using the

Shuffled Complex Evolution algorithm (SCE-UA; Duan et al., 1992). This global

optimization algorithm samples from the parameter space using different prior

configurations, reducing the risk of local minima and enhancing robustness. Only major

parameters were subject to calibration, thereby reducing the risk of over-parameterization

and ensuring model efficiency. Importantly, the snow and freeze-thaw modules developed

here are model-independent and can be integrated into other hydrological models.

Adaptation requires only alignment with the target model's soil layering and runoff

generation structure—for example, setting the humus layer thickness $L_h$ to zero if interflow

is not considered, and the three-layer evapotranspiration scheme can be directly embedded. The flexible design preserves overall simplicity while ensuring physical consistency and adaptability, making the approach especially suitable for cold-region studies in ungauged or data-limited basins.

## 2.3 Model implementation and evaluations

### 2.3.1 Study area

The Yalong River is located in the southeastern part of the Tibetan Plateau and is the largest tributary of the Jinsha River. The main river stretches 1,571 km with a natural drop of 3,830 meters. Rich in hydroelectric resources, 21 hydropower stations are planned along the river, primarily concentrated in the downstream region. This study focuses on the mid-upper reaches of the Yalong River Basin (29.94°-34.21°N, 96.82°-101.63°E), with the Yajiang hydrological station serving as the outlet flow measurement (Fig. 3), covering an area of approximately 67,000 km². The elevation ranges from 2,500 to 5,900 meters, with a general south-north orientation with a high elevation in the northwest and low in the southeast, predominantly mountainous. Most precipitation occurs in summer, with limited snowfall in winter. Due to the complex terrain, meteorological observations in the study area are constrained. Seasonally frozen ground is widespread, with some areas containing sporadic permafrost (Ran et al., 2012). Seasonal snow significantly affects spring runoff, with about 50% of runoff directly fed by precipitation and the rest from glacier melt and groundwater (Wu et al., 2024). This pattern may change in the future due to global warming (Yao et al., 2022).

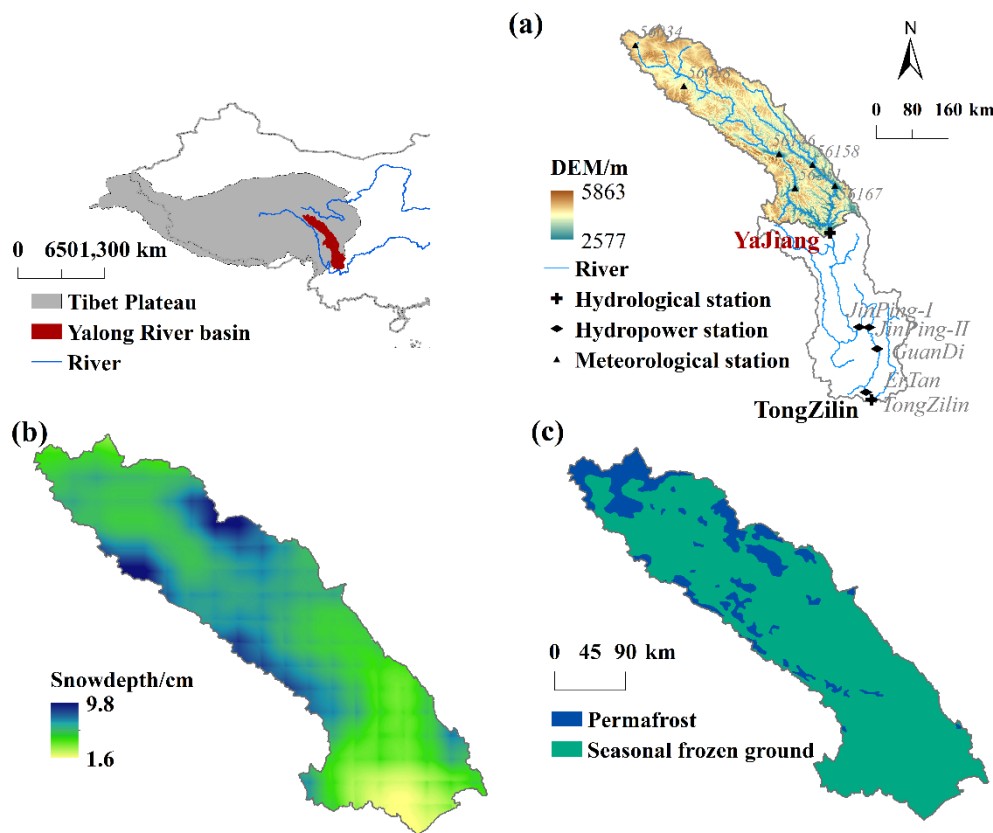

**Figure 3.** The mid-upper reaches of the Yalong River Basin in the southeastern Qinghai-Tibet Plateau, China, (a) topographic features, (b) annual average snow depth distribution, (c) seasonal frozen ground areas (https://doi.org/10.3972/westdc.0078.2013.db.).

### 2.3.2 Data collection, pre-processing and implementation

The data collection and description are presented in Table 3. Considering the computational efficiency of the model, the precision of precipitation, air temperature, snow depth, and all other data were resampled to 0.05°. The hydrological simulation performance of the original models (GXAJ and SNOW17) and the further developed models (GXAJ-S and GXAJ-S-SF) were evaluated in the mid-upper reaches of the Yalong River Basin. First, the SNOW17 model was calibrated (2000-2010) and validated (2011-2018) using remote sensing snow depth data to determine snowmelt parameters, with the freeze-thaw processes determined

through empirical formulas. Then, the developed models GXAJ-S and GXAJ-S-SF were used

to simulate runoff during the same period, focusing on the snowmelt runoff period from March

to June, and compared with the original GXAJ model. The impact of the two components

(SNOW17 and SFG) on the runoff process, including runoff sources, components, and

evapotranspiration, was also analyzed. Various statistical criteria, including Nash-Sutcliffe

Efficiency (NSE), BIAS, Relative Error (RE), and Root Mean Squared Error (RMSE), were

used to evaluate model performance. These criteria are defined in equations S1-S4.

**Table 3.** Data collection and description.

| Data | Spatial resolution | Source | Description |
|---|---|---|---|
| Runoff | - | China Hydrology Yearbook from Ministry of Water Resources of China (http://www.mwr.gov.cn/) . | Daily runoff data (2000-2018) at the Yajiang hydrological station |
| Precipitation and air temperature | 0.05°× 0.05° | China Meteorological Administration (CMA, http://data.cma.cn) | Precipitation and air temperature at meteorological stations were interpolated to 0.05° and corrected by post-processing analysis. |
| Ground temperature | - | China Meteorological Administration (CMA, http://data.cma.cn) | Site data |
| Potential evapotranspiration | 0.25°×0.25° | - | Potential evapotranspiration was estimated using the Penman-Monteith model (Allen et al., 1998) |
| Atmospheric pressure, relative humidity, and sunshine duration | 0.25°×0.25° | CN05.1 dataset (New et al., 2000) | Daily data (1961-2020), based on site data |
| Snow depth | 0.05°× 0.05° | National Tibetan Plateau Data Center | Refer to (Yan et al., 2022) |
| Digital Elevation Model | 1km×1km | U.S. Geological Survey (USGS) (GTOPO30) | https://www.usgs.gov/centers/eros/science/usgs-eros-archive-digital-elevation-global-30-arc-second-elevation-gtopo30 |
| Vegetation cover | 1km×1km | University of Maryland Food and Agriculture Organization | Refer to (Potapov et al., 2022) |
| Soil type | 10km×10km | | Refer to (Fischer et al., 2008) |
| Maximum thickness of seasonally frozen ground | 1km×1km | National Tibetan Plateau Data Center (https://cstr.cn/18406.11.Cryos.tpdc.300955) | Maximum thickness of seasonally frozen ground every 10 years from 1961 to 2020 was simulated using the Stefan equation based on remote sensing surface temperature data |
| Snow cover | 500m×500m | Daily fractional snow cover dataset over High Asia (2002 – 2016) | http://www.sciencedb.cn/dataSet/handle/457 |
| Soil temperature | 0.1° x 0.1° | ERA5-Land hourly data from 1950 to present | https://cds.climate.copernicus.eu/datasets/reanalysis-era5-land?tab=overview |

# 3. Results

## 3.1 Simulation of snow accumulation and freeze-thaw process

At the basin scale, the SNOW17 model was first applied to determine the model parameters. The average daily snow depth simulated during the calibration period (2000-2010) and the validation period (2011-2018) was compared with remote sensing data. As shown in Fig. 4, the simulated snow depth closely followed the trend observed in the remote sensing data. Although the model slightly overestimated snow depth overall, it demonstrated

reasonable accuracy in capturing the dynamics of snow depth. The model performed better during the validation period (RMSE = 1.6 cm, BIAS = 0.3 cm) compared to the calibration period (RMSE = 2.1 cm, BIAS = 0.9 cm). The model simulation error is relatively large when the snow depth is high, which may be attributed to a more complex snow melting process under deep snow conditions. Shallower snow depths may reduce errors related to model

simplifications of complex snowmelt process under deep snow conditions, thereby improving the simulation accuracy. This may also be the reason why the simulation accuracy is higher in the validation period (shallower snow depth) than in the calibration period (deeper snow depth). The trend lines in Fig. 4 indicate a declining trend in snow depth from 2000 to 2018 in the mid-upper reaches of the Yalong River Basin, which is evident in both the remote sensing data and

the model simulation results. Overall, the SNOW17 model showed satisfactorily simulations results of snow depth.

      This study systematically validated the simulation results of frozen soil depth based on the Stefan empirical formula through multi-source data comparison. Fig. 5 presents the frozen depth derived from ERA5 reanalysis data using four soil temperature layers (0–7 cm, 7–28 cm,

28–100 cm, and 100–289 cm; freezing occurs when layer temperatures fall below 0°C). The

seasonal freeze-thaw depths calculated by the Stefan formula exhibit high consistency with

ERA5-derived results in both freeze-thaw timing and variation trends. Notably, the ERA5-

based frozen depths display a stepwise variation pattern, with the maximum freezing depth

terminating at the 100 cm layer, likely attributable to the freezing inhibition effect caused by

higher temperatures in the deep soil layer (100–289 cm). The simulations indicate that the

freezing process initiates in late September, reaches the maximum depth of 1.4 m by late March

of the following year, and completes thawing by late May. This temporal pattern aligns closely

with ground temperature observations from basin meteorological stations (Fig. S6; mean errors

of ≤5 days for initial freezing dates and ≤10 days for initial thawing dates).

To further evaluate the model's spatial performance, the 2000–2018 mean maximum

frozen depth distribution was compared with contemporaneous data from the National Tibetan

Plateau Data Center (Table 3; Fig. S7). The Stefan formula-based simulations, incorporating

station-based temperature interpolation, demonstrate smoother spatial transitions—a

characteristic linked to model parameterization. Both datasets reveal a gradient pattern of

deeper frozen depths in upstream valley regions and shallower depths in downstream areas,

with a spatial correlation coefficient of 0.89. Furthermore, the observed decreasing trend in

frozen depth during 2000–2018 corresponds with accelerated snowmelt patterns (Fig. 4),

highlighting the coupled response of the cryosphere to climate change.

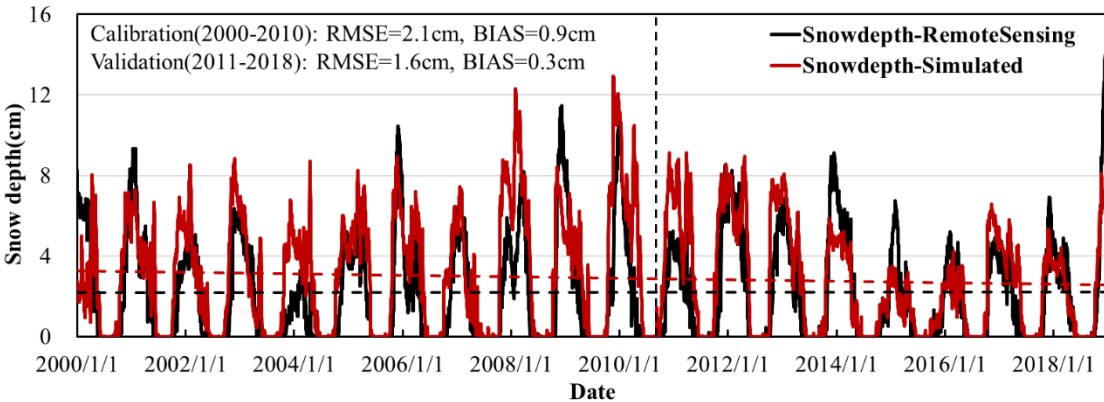

**Figure 4.** Comparison of simulated and observed basin-average snow depth in the Yalong River Basin during the calibration (2000-2010) and validation (2011-2018) periods, and the dashed lines represent the trend of snow depth.

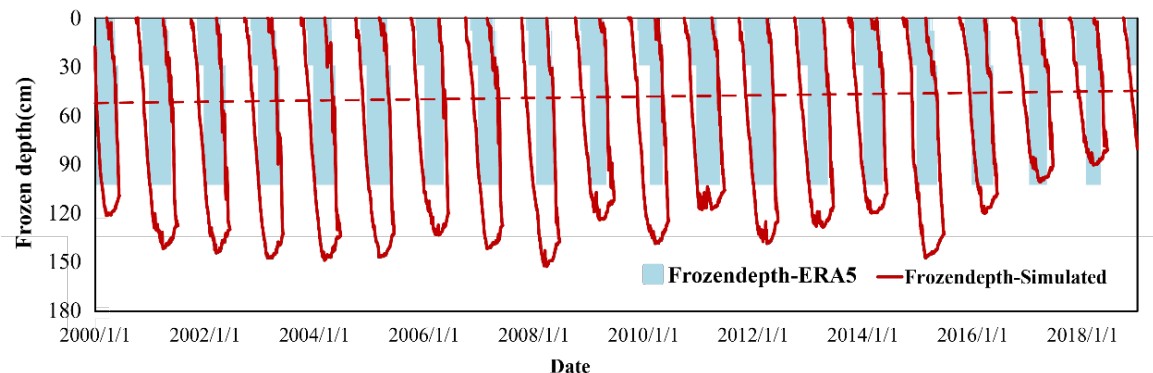

**Figure 5.** Seasonal freeze-thaw depth changes calculated using the Stefan empirical formula and ERA5data in the study area, and the dashed lines represent the trend of frozen depth.

To further illustrate impacts of freeze-thaw processes, Fig. 6 shows the annual variation of basin-average snow depth, frozen ground, effective humus layer, effective vadose zone, and soil water/ice content in 2001. The figure shows that the formation of frozen ground preceded the occurrence of snow. In particular, during periods of little or shallow snow depth (October–December), the rate of ground freezing was relatively fast. However, as snow depth increased (enhancing its insulating effect), the freezing rate gradually slowed down. Snow depth reached its maximum value (approximately 9 cm) in February and then rapidly decreased to 3 cm. Only

when the snow depth was small did the ground freeze begin to melt. Therefore, the ground freezing and thawing trends were closely aligned with changes in snow depth.

Moreover, Fig. 6(b) demonstrates that frozen ground freezes part of the vadose zone, significantly reducing the effective vadose zone thickness of the Yalong River basin, particularly during cold months (October–December and January–May), with the humus layer even becoming entirely frozen. When the temperature rises, the surface frozen ground melts rapidly, and there are frequent and short freeze-thaw cycles. The humus layer and the vadose

zone melt in turn and return to an unfrozen state. Fig. 6(c) further illustrates a notable increase in soil ice content due to ground freezing, as well as a corresponding decrease in soil water content. These solid-liquid transformation processes of the Yalong River basin hence exert a critical influence on the water storage capacity of the vadose zone, alters infiltration pathways, and consequently affects the partitioning of runoff into surface water and groundwater

components.

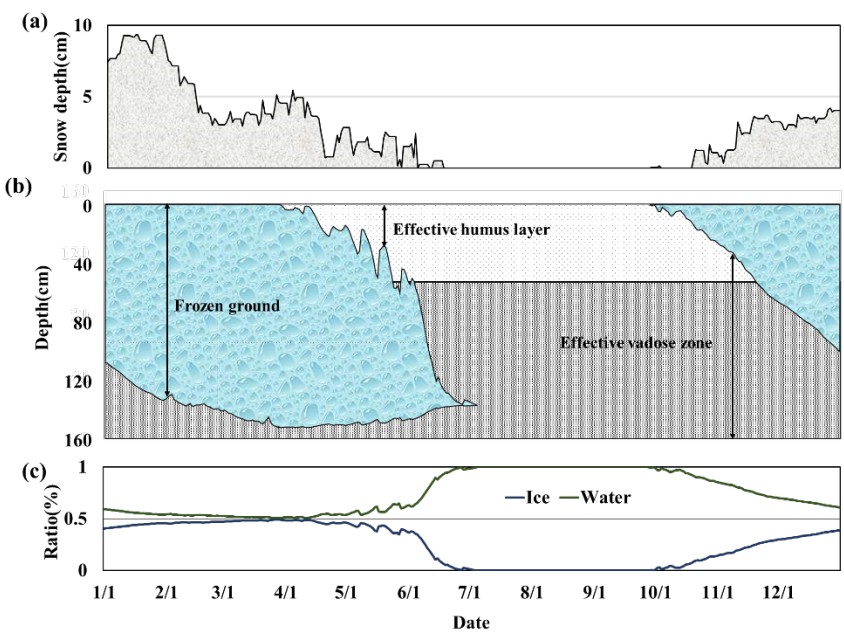

**Figure 6.** (a) Annual variation of basin-average snow depth; (b) impact of frozen ground on the basin-

average depths of the effective vadose zone and humus layer; (c) basin-average ratio of water / ice content

in the vadose zone, taking 2001 as an example

## 3.2 Calibration and validation of the streamflow

Fig. 7 (a) shows the simulated daily streamflow at the Yalong station of the GXAJ model

from 2000 to 2018, without considering the effects of snow and seasonally frozen ground

(SFG). The model did not distinguish between rainfall and snowfall, all incoming water was

treated as rainfall. The model performed relatively well during both the calibration period

(2000-2010) and the validation period (2010-2018), with NSE around 0.8. However,

streamflow was often underestimated in winter and spring, which can be related to the impacts

of frozen ground and snow. To further understand the model's performance in specific periods,

the streamflow simulation results from March to June were analyzed separately (Fig. 7 (b)).

The results then showed that the GXAJ model had considerable inaccuracies in simulating

spring snowmelt, especially during the validation period, where NSE decreased to 0.44 and RE

reached -0.50. These metrics reflect that the GXAJ model calculated spring streamflow solely

based on rainfall, failing to reflect the delayed effect of snowmelt on streamflow, which hence

led to streamflow underestimation.

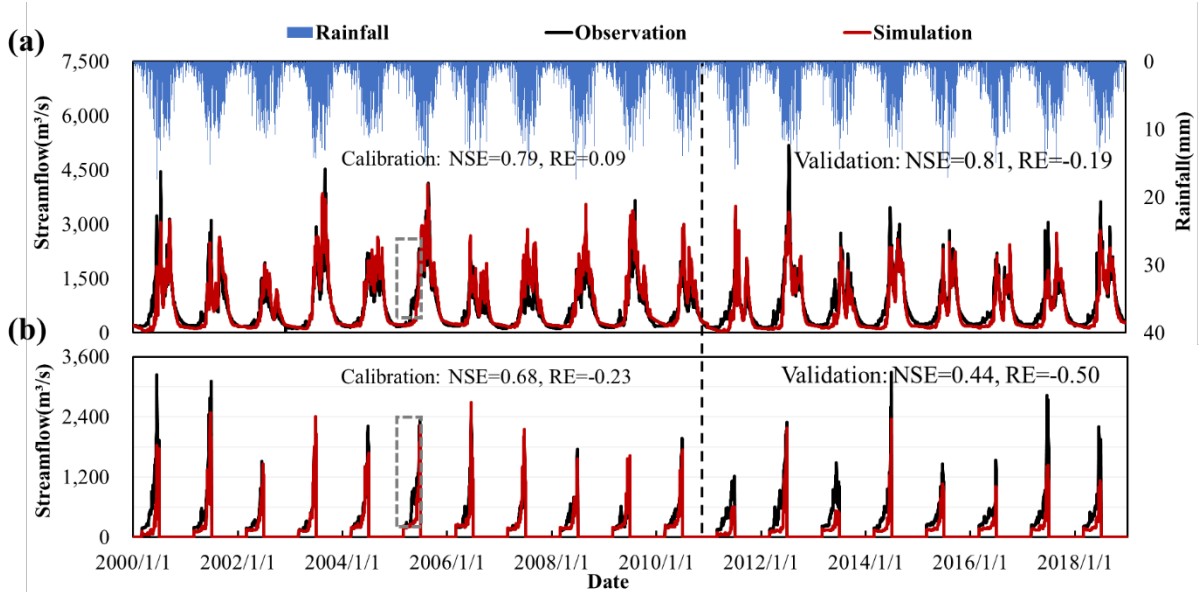

Figure 7. (a) Daily observed streamflow at the Yalong station and simulated streamflow by the GXAJ model during the calibration (2000-2010) and validation (2011-2018) periods, (b) with spring snowmelt from March to June highlighted (within dashed rectangle).

When snow cover effects were considered in the GXAJ-S model, the accuracy of daily streamflow simulation during 2000-2018 significantly improved (Fig. 8 (a)), especially during the calibration period (NSE=0.82, RE=0.05), indicating that a better performance of the GXAJ-S model in simulating snow accumulation and its hydrological effects, as compared to the original GXAJ model. However, as shown in Fig. 8 (b), the model still showed inaccuracies during the spring snowmelt period, particularly in the validation stage (NSE=0.68, RE=-0.36). The decrease in accuracy during the validation period may be partially related to changes in applicability of model assumptions and parameter values between the calibration and validation periods. It probably also reflects that the model has not yet fully considered the interaction between snow and frozen ground on runoff, with the delayed water retention effect of frozen ground during the spring snowmelt period likely being a major source of error.

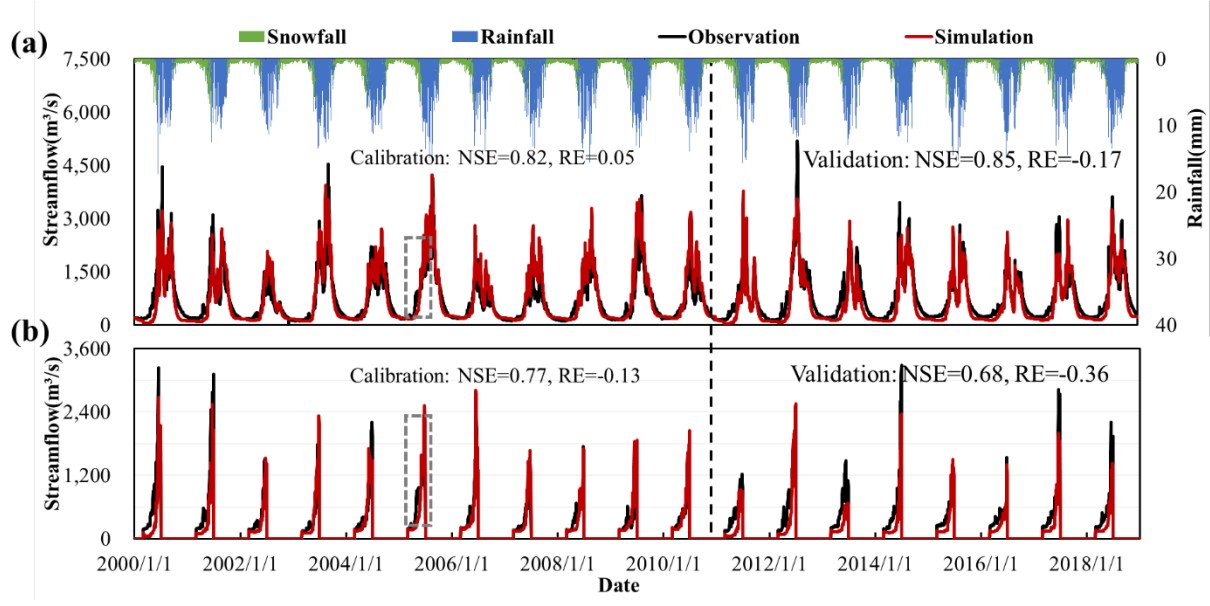

**Figure 8.** (a) Comparison of GXAJ-S model simulation results with observed values, (b) highlighting spring

snowmelt from March to June.

Considering both snow cover and SFG effects, the GXAJ-S-SF model demonstrated

excellent performance in overall daily runoff simulation (Fig. 9 (a)). The NSE values for both

the calibration and validation periods exceeded 0.8, and the RE values were close to zero,

indicating a high degree of fit between the model and observed runoff time series. Compared

to the GXAJ-S model, the GXAJ-S-SF model was more accurate in simulating daily runoff,

especially during the calibration period, showing higher accuracy. In simulating spring

snowmelt runoff (Fig. 9 (b)), the GXAJ-S-SF model showed improvements over the previous

models, particularly during the calibration phase, achieving higher accuracy. Although some

580 underestimation remained in the validation period, the GXAJ-S-SF model demonstrated higher

accuracy compared to the other two models.

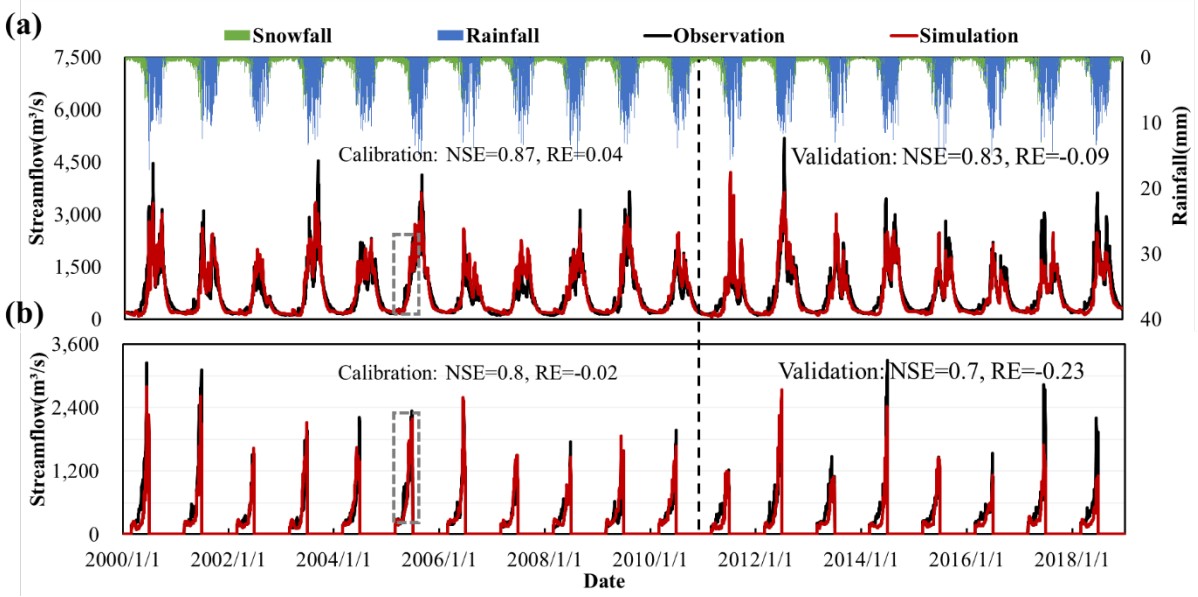

**Figure 9.** (a) Comparison of GXAJ-S-SF model simulation results with observed values, (b) highlighting spring snowmelt from March to June.

To provide a more comprehensive comparison of the three models, we have included an evaluation of computational efficiency. Table S1 presents the calibration and simulation times for GXAJ, GXAJ-S, and GXAJ-S-SF. The results indicate that while GXAJ-S-SF provides improved physical representation, it requires longer computation time compared to GXAJ and GXAJ-S. This information is useful for users who may prioritize efficiency over accuracy in certain applications.

## 3.3 Model differences in simulated runoff components and soil evapotranspiration

Fig. 10 illustrates differences in the simulation of surface water runoff, interflow, and groundwater runoff among different models. The GXAJ and GXAJ-S models simultaneously reached the minimum percentage of interflow and maximum percentage of surface runoff in June and May, respectively, possibly due to the modelled soil saturation in both cases reaching relatively high values during the rainy summer season, thereby increasing surface runoff. Overall, the runoff components simulated by the GXAJ and GXAJ-S models were similar, with interflow accounting for the largest proportion (55-70%), followed by groundwater runoff (20-

26%). The similarities between these two cases suggest that the omission (in GXAJ) or inclusion (in GXAJ-S) of snow processes in the modelling had a relatively limited impact on the simulated runoff dynamics. However, the GXAJ-S-SF model exhibited significant simulation differences. Fig. 10 (c) shows that during the cold months (January-March, November-December), the proportion of surface water runoff increased significantly to 48-83%, mainly influenced by SFG (39-77%) as seen in Fig. 6b, while interflow and groundwater runoff decreased substantially. This was because SFG interrupted the connection between surface water and groundwater, preventing infiltration and leading to more surface water runoff. Additionally, the impact of SFG on interflow was most evident from March to May. As the surface soil thawed from top to bottom, the thawed soil layer tended to produce interflow. Groundwater runoff was hindered by frozen ground, remaining low during the cold season until frozen soil completely melted in summer, when groundwater runoff returned to its unfrozen state. This dynamic change indicates that SFG processes plays a critical role in regulating runoff composition over time. Moreover, SFG has a pronounced "decoupling effect" on surface runoff and groundwater runoff during cold months, interrupting their connection and restricting groundwater recharge and deep percolation.

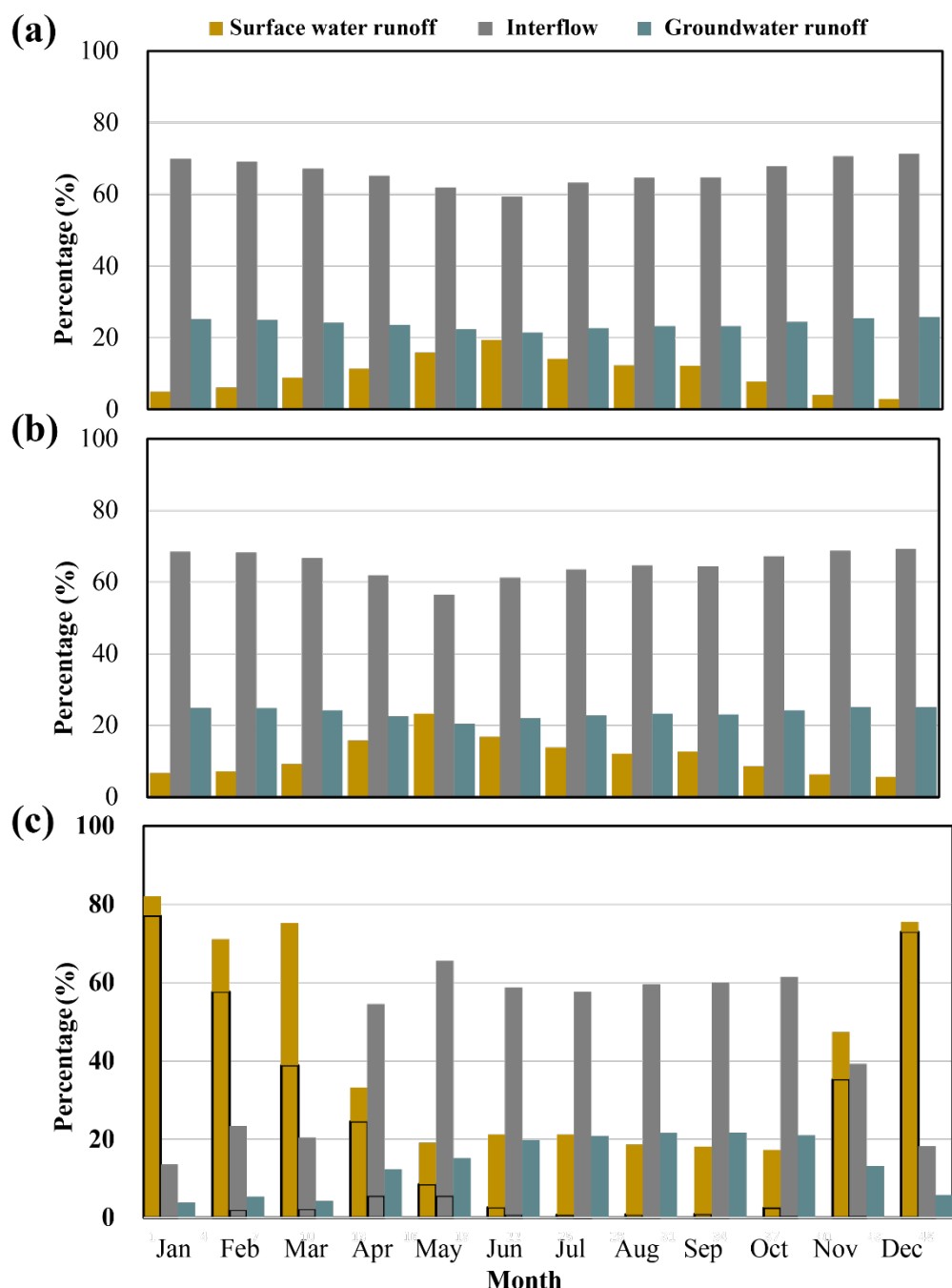

**Figure 10.** Comparison of simulated runoff components by models: (a) GXAJ, (b) GXAJ-S, and (c) GXAJ-S-SF, with the black box in (c) indicating runoff components influenced by SFG. The percentage of the y-axis represents the percent contribution of the considered runoff component (surface water runoff, interflow and groundwater runoff) to the total runoff.

Based on the model comparison results shown in Fig. 11, the suppression effect of snow and frozen ground on soil evapotranspiration during cold months exhibited significant temporal

variability. During the cold period (November to March), evapotranspiration in the GXAJ-S-SF model remained generally below 5 mm, whereas in the GXAJ model, it ranged between 10 and 30 mm, with an average reduction of approximately 85%. This substantial decrease was primarily attributed to two mechanisms: first, snow cover effectively inhibited soil moisture

evaporation, leading to snow loss primarily through sublimation rather than direct evapotranspiration; second, the formation of frozen ground created a barrier within the soil, restricting upward water transport and significantly reducing soil moisture loss. As temperatures rose, evapotranspiration across the basin gradually intensified, and in May, the difference between the two models reached its maximum, approximately 30 mm. At this time,

the snow had mostly melted, but frozen ground remained, continuing to influence soil moisture transport and evapotranspiration, thereby maximizing the discrepancy between the two models. During summer (July to September), the influence of snow and frozen ground gradually diminished, and the difference in simulated evapotranspiration between the two models decreased to within 5 mm, indicating that the effects of freezing had essentially disappeared.

As shown in Fig. 11, within the dashed rectangular area representing the summer of 2010, the simulation results of both models converged, suggesting that even in high-altitude regions, the residual effects of frost and snow on basin-wide evapotranspiration were negligible. Overall, the comparison between the GXAJ-S-SF and GXAJ models clearly revealed the significant regulatory role of snow and frozen ground in soil evapotranspiration during cold seasons. This

effect was particularly pronounced in winter, effectively preserving soil moisture and reducing water loss by suppressing evapotranspiration. However, as temperatures rose, this influence gradually weakened and eventually disappeared in the warm season.

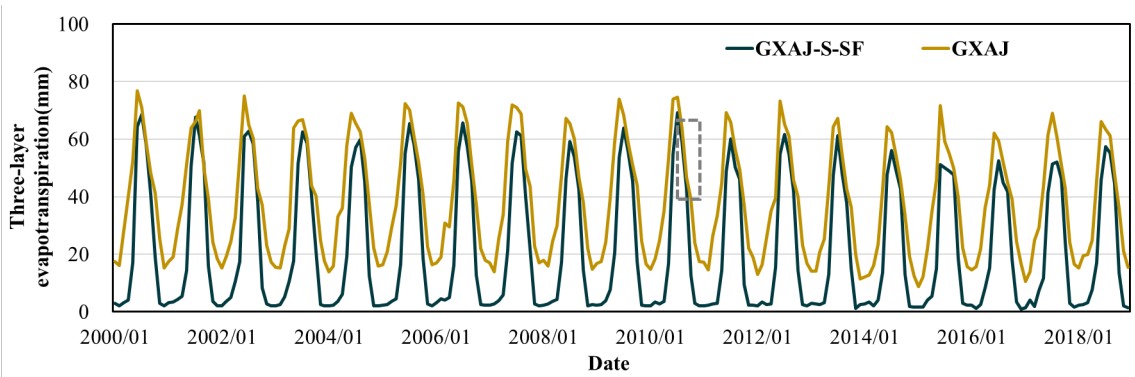

**Figure 11.** Simulated monthly evapotranspiration series during the study period. The dashed rectangle represents 2010 summer evapotranspiration.

## 4. Discussion

### 4.1 Key limitations in hydrological models in relation to their process complexity

A limitation in the application of the GXAJ base model, which neglects impacts of snow and ice, is related to the fact that the parameters of its modules are determined based on historical basin characteristics. Although such models without frozen ground components can, through appropriate calibration or optimization of parameters, in some cases successfully reproduce historical hydrological processes in cold regions under stable conditions (Li et al., 2011; Zhang et al., 2017), they may not be suitable for evaluating the consequences of future changes as their calibrated values do not represent new conditions of the basin, and as the model lacks physical representation of key drivers of change. Our study demonstrates that incorporating the effects of seasonally frozen ground (SFG) and snow into a basic model can provide robust and physically consistent results in simulating large-scale hydrological processes in cold regions, which can be particularly important for predicting hydrological impacts of future climate change scenarios.

Although significant progress has been made in physical models that account for snow

and freeze-thaw processes, their application in cold-region hydrology remains challenging. The spatial heterogeneity in topography, vegetation, and soil properties in cold regions introduces substantial uncertainty in the energy balance and surface heat flux simulations (Gao et al., 2018). Errors in estimating surface albedo, net radiation, and snow thermal properties can cascade into inaccuracies in ground temperature and freeze–thaw simulations (Wang et al., 2024). Moreover, physical models often require high-resolution spatial inputs and detailed parameterization (e.g., soil hydraulic conductivity, canopy structure, and snow thermal conductivity) (Gao et al., 2018; Song et al., 2022), which are rarely available for large-scale and high-altitude basins like the Yalong River. The diversity in climatic and geographic conditions further reduces model transferability (Yong et al., 2023; Zhou et al., 2021).

In contrast, our proposed GXAJ-S and GXAJ-S-SF models adopt a distributed framework that integrates key cold-region hydrological processes (i.e., snowmelt and seasonally frozen ground dynamics) based on physically grounded but simplified formulations. For example, the snow module is adapted from the SNOW17 model, and the frost depth is calculated using the Stefan equation, incorporating snow insulation effects. In particular, our three-step approach (involving the GXAJ, GXAJ-S and GXAJ-S-SF models) implies that a limited number of additional parameters are introduced in each performance evaluation step, which enables the identification of well-functioning levels of model complexity while involving only a small number of parameters. This greatly reduces the risk of overfitting. We also considered the risk of coincidental good performance by potentially overfitted models by evaluating in which way the addition of process-based modules alters the model behavior in multiple sub-catchments and over multiple seasons. We could then for instance see that, rather than increasing the sub-

catchment and seasonal performance in random ways, the addition of the snow and SFG

modules specifically increased cold-season performance in low-temperature (high-altitude)

parts of the study area, which is consistent with the expected effects of the considered processes.

This hence provides a logical explanation as to why the here demonstrated simulation

performance was strong (e.g., with high NSE) despite being based on few parameters as

compared with e.g. VIC and SWAT applications. In a direct comparison using the same study

period (2007–2011), the VIC model yielded NSE values of 0.75 (calibration) and 0.65

(validation) in the Yalong River basin (Li et al., 2018b), whereas our model achieved NSE

values of 0.87 and 0.74, respectively. This suggests that our approach is more suited to this

data-scarce mountainous basin, where excessive model complexity may not translate into

improved predictive accuracy (Wang et al., 2024). In turn this may be related to increased

demands of uncertain input data of complex physical models (Gao et al., 2018; Qin et al., 2017;

Wang et al., 2024).

In complex mountainous cold regions, observation remains a bottleneck (Gao et al., 2022).

Due to limitations in measured data on frozen soil and snow depth in the considered Yalong

River basin, this study used multi-source remote sensing data and reanalysis data for calibration

and verification from multiple perspectives. In particular, errors in remote sensing snow depth

data (Yan et al., 2022; Zou et al., 2014) can propagate to the model output. However, previous

studies have specifically investigated the here used remote sensing dataset for the Yalong River

basin showing that its accuracy is high (Wu et al., 2024), which suggests that model errors

should be relatively low. This study further compared MODIS snow cover data with model

simulations, revealing that snow cover extended over up to half of the study area, with daily

snow cover fraction exhibiting a high correlation coefficient of 0.91 between the two datasets.

Figure S8 illustrates the spatial distribution of simulated snow depth and MODIS-derived snow cover on December 1, 2015, demonstrating strong consistency in coverage patterns. We also recognize that the use of surface/soil temperature and maximum frozen ground depth to verify the freeze-thaw process introduces some uncertainty (Li et al., 2022). Since the GXAJ-S and GXAJ-S-SF model variants used the same temperature, snow and frozen ground data in the present simulations, they can be expected to share similar data errors, However, due the non-linear nature of the modeled processes, such data errors may still not cancel completely when comparing different models. Nevertheless, observed differences in model performance between these models are mainly expected to reflect differences in model capabilities rather than differences in input datasets. Future work should focus on improving remote sensing data quality and exploring the long-term robustness of the model to further enhance performance and improve our understanding of the freeze-thaw processes in complex mountainous cold regions.

Hydrological modeling typically prioritizes model fitness, which in theory can be improved by introducing more fitting parameters. However, this study highlights differences that are due to addition of process-based modules (regarding snow and frozen ground). This implies that improvements in model fit and differences in associated model output (e.g. runoff and evapotranspiration) reflect how the considered snow and/ or frozen ground processes more concretely alter hydrological flows. This therefore increases the understanding of underlying hydrological processes (Gao et al., 2022) in large-scale applications such as the Yalong River basin that additionally has a complex topography with large elevation differences yielding high

spatiotemporal heterogeneity in snowmelt and freeze-thaw cycles of soil.

**4.2 The impact of seasonal frozen ground/snow on hydrological processes**

SFG is a thermally driven phenomenon dependent on ground heat. As previously mentioned, it is clear that SFG in many cases has crucial impact locally, as ground freezing causes ice to block previously water-filled soil pores, restricting water flow through them. This process directly affects the seasonal permeability of the vadose zone and groundwater recharge (Ge et al., 2011). Our study similarly found that the formation of frozen ground not only significantly reduces the effective thickness of the vadose zone but also leads to the complete freezing of the humus layer (Fig. 6). Additionally, snow cover plays a key role in modulating frozen ground development through its thermal insulation effect: when snow cover is shallow, the freezing rate is accelerated; however, as snow depth increases, the freezing rate of the frozen ground slows down (Fig. 6). This finding aligns with Iwata et al. (2018), who suggested that despite subzero air temperatures, thick early-winter snow cover can significantly reduce or even completely prevent ground freezing.

The impact of soil freeze-thaw cycles on basin runoff generation varies seasonally (Fig. 6; Gao et al., 2023). Previous studies have shown that spring runoff is primarily composed of surface runoff and interflow, while summer thawing of frozen ground enhances groundwater recharge (Huelsmann et al., 2015). Through multi-model comparisons, this study further quantified these processes: when accounting for SFG effects, the proportion of surface runoff from November to March increased by 39% to 77% compared to the baseline model without SFG. Additionally, the influence of SFG on interflow was most pronounced in spring (Fig. 10). This is largely due to the relatively impermeable surface frozen ground, which directly

generates substantial surface runoff. Even as temperatures rise and the surface soil gradually

thaws, the effective vadose zone remains highly susceptible to saturation (Guo et al., 2022;

Huelsmann et al., 2015; Ireson et al., 2013; Wang et al., 2017), leading to the formation of

interflow at the base of the thawed layer (Fig. S4). Overall, the multi-model simulations of

daily runoff processes therefore provided important insights into key factors governing basin

hydrology under seasonal variations in cold regions.

Furthermore, the freeze-thaw process complicates soil water movement within the vadose

zone (Yu et al., 2018). Within the frozen soil layer, water movement is minimal, resulting in

negligible upward evaporation. Above the freezing interface, water moves upward and

evaporates. As the thawed layer thickens, evaporation and infiltration capacities gradually

increase (Yu et al., 2018). The simulation results from the GXAJ-S-SF model in this study

further reflected significant seasonal differences in the suppression effect of the snow-frozen

ground interaction on evapotranspiration (Fig. 11): during the freezing period (December–

March), evapotranspiration decreased by 85%, while after thawing (July–September), the

difference was reduced to within 5 mm. This process not only highlights the barrier effect of

frozen ground but also demonstrates the suppression of snow sublimation (Anderson, 1973).

These processes, including freeze-thaw dynamics, soil moisture movement, and the effects of

snow and SFG on evapotranspiration, can influence the hydrological cycle and ecosystems by

altering water availability and flow patterns. These effects, particularly during freeze-thaw

periods, may lead to changes in water storage, infiltration, and runoff, which can alter regional

water resource management and ecosystem resilience.

In addition, snowmelt runoff is a vital component of spring runoff in the Yalong River

Basin, as further demonstrated in this study (Fig. S8). Snow cover varies with elevation, exhibiting significant spatiotemporal heterogeneity (Li et al., 2018). Under the backdrop of global warming, rising average temperatures are expected to affect the composition and duration of snow cover (Fig. S9; IPCC, 2021). Changes in snowmelt volume can influence

downstream runoff, impacting water resource management and ecological balance. Incorporating the effects of snow into this study has improved the predictive accuracy of hydrological simulations for daily runoff and spring snowmelt runoff (Fig. 7, 8). Both remote sensing data and model simulation results in this study showed a decreasing trend in snow/frozen depth from 2000 to 2018 (Figs. 4, 5), which is consistent with the results in similar

study areas (Qin et al., 2017; Song et al., 2022). Winter snowmelt water typically infiltrates the upper soil layer, forming an almost impermeable "concrete frost" layer at the interface between the ground and snow layer upon refreezing (Dunne and Black, 1971). Due to warming, the ice content in SFG is denser, potentially altering the hydrological response of SFG during major spring snowmelt periods (Hardy et al., 2001). The snowfall process profoundly impacts ground

thermal conditions, with some proposing that we might even see "colder soils in warmer climates" (Halim and Thomas, 2018). In summary, predicting future changes in SFG and its hydrological importance remains challenging due to the complex interactions between climate, land, water, ecosystems, and human activities. The hydrological relevance of SFG may increase due to factors such as reduced snow cover and changes in snow insulation capacity, more

frequent freeze-thaw cycles, rain-on-snow events, and land cover changes (Cuo et al., 2015). Such may therefore significantly impact the spatial and temporal availability of water resources in SFG regions.

This study quantitatively analyzed the impact of seasonal snow and frozen ground on hydrological processes based on the hydrological model, and its validity was confirmed not only by measured runoff but also by multi-source data, especially the trends in snow and frozen soil changes. Although the model developed based on GXAJ has great potential for application in other cold regions, its use should be based on a thorough understanding of the assumptions and structural limitations of the model. Snow and seasonal frozen ground are only part of the hydrological drivers in cold environments, with other important factors such as glacial melt, geological conditions, and soil thermal properties also playing significant roles (Du et al., 2022; Gao et al., 2023), but these are often difficult to observe or measure directly. Additionally, topographic and vegetation dynamics can significantly impact runoff, infiltration, and evapotranspiration processes (Lazo et al., 2019). While these factors are not currently incorporated into our modeling system, future work could address this by integrating corresponding glacier runoff modules, vegetation-hydrology modules, or improving the representation of frozen ground. The empirical parameters in the SNOW17 model and the Stefan equation have clear physical significance and have been validated by previous studies (Anderson, 2006; Ran et al., 2022; Zou et al., 2014), but when applied to other regions, recalibration of key parameters is still necessary. Therefore, expanding the application of complex hydrological models requires careful attention to the local and regional variability of environmental conditions. This may increase the difficulty of modeling but also greatly enhance the understanding of hydrological processes and the generalizability of the assumptions. In cold and data-scarce regions, extending the application of complex hydrological models must strike a balance between model complexity and data availability to

ensure their applicability and reliability.

## 5. Conclusions

The understanding of cold-region hydrology remains incomplete, primarily due to limited observational data, which also constrains quantitative analyses of water resources, especially in complex mountainous basins like the Tibetan Plateau. This study developed and applied two

enhanced versions (GXAJ-S, which incorporates snowmelt, and GXAJ-S-SF, which additionally considers freeze-thaw processes) based on the original GXAJ model. The models were calibrated and validated using measured daily runoff (2000–2018) obtained at the Yajiang discharge station in the Yalong River basin. The results showed that the GXAJ-S-SF model achieved the highest simulation accuracy, with significant improvements in NSE and RE for

total runoff and runoff during snowmelt conditions. These enhanced models integrate multiple key cold-region hydrological processes while maintaining low parameter complexity, making them particularly suitable for cold regions with complex hydro-meteorological conditions and scarce data availability.

Further analysis revealed the tightly coupled interactions between snow dynamics, freeze-

thaw cycles, and unsaturated zone processes. Snow accumulation and subsequent melting were found to directly influence the depth and duration of soil freezing, altering the thermal and hydrological state of the vadose zone. The presence of frozen ground significantly reduced soil permeability and water-holding capacity, affecting runoff partitioning. During cold months (November–March), SFG processes led to a 39–77% increase in simulated surface runoff while

interflow and groundwater recharge were substantially reduced or entirely suppressed. As thawing progressed (March–May), interflow became the most affected runoff component.

Additionally, the model captured an average 85% reduction in soil evapotranspiration during the frozen period relative to the baseline model, with the largest difference observed in May (~30 mm), attributed to restricted moisture movement in frozen soils and insulating effects of snow cover.

By comparing multiple model configurations, this study provides valuable insights into the role of cold-region processes in shaping water balance components. The findings emphasize that the improved modeling framework not only enhances runoff simulation but also assesses the impact of snow and frozen soil on runoff generation and water resource availability. The developed snow and SFG components are designed to be flexible and adaptable, allowing seamless integration with hydrological models beyond GXAJ. A comparative analysis between the here investigated set of models and (even) more complex physically based models illustrates that the data limitation in the Yalong basin is likely to currently constrain the performance of physically based models. This hence suggests the need to expand observational efforts before expanding modelling efforts to further improve predictive capacity.

**Author Contributions**

N.W. and J.J. conceived the idea and designed the research framework. Z.N., X.Y. and W.L. carried out data collection, preprocessing, and method determination. H.W., H.H. and Q.Z. performed data analysis, graphical visualization, and manuscript preparation. K.Z., A.N. and W.G. contributed to the manuscript refinement. All authors have read and agreed to the published version of the manuscript.

**Declaration of Competing Interest**

The authors declare that they have no known competing financial interests or personal

relationships that could have appeared to influence the work reported in this paper.

**Acknowledgments**

This study was supported by the Fundamental Research Funds for National Key Research and Development Program of China (2023YFC3006500), the Special Fund Project of Jiangsu Province Science and Technology Program (BZ2024035), the fund of National Key Laboratory of Water Disaster Prevention (524015222), and Project "Applied Scientific Research on the 'Three-Line Defense' Strengthening Foundation Project for Rainfall and Water Monitoring & Forecasting in Shandong Province" (3700000025001720240235). The first author also received a grant from the China Scholarship Council to study at Lund University in Sweden. The authors thank Ministry of Water Resources of China (http://www.mwr.gov.cn/) for providing the natural and observed streamflow, the China Meteorological Administration (CMA) for providing the climatic data (http://data.cma.cn/). The code and data used in this paper are available from the first author's GitHub repository (https://github.com/NanWu16/ ) or by contacting the corresponding author (kzhang@hhu.edu.cn).

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
