# Peer review of "Predicting Snow-Cover and Frozen Ground Impacts on Large Basin Runoff: Developing Appropriate Model Complexity"

_Hydrology and Earth System Sciences, 2024_

## Author Comment (AC1)

**General comments**

**Comments 1:** This manuscript is well written, and the work done appears quite meticulous and informative from a methodological point of view, but I am not fully convinced of the novelty of this manuscript. The manuscript shows that GXAJ-S-SF outperforms GXAJ. It is self-evidently almost certain that a more accurate hydrological partitioning can be achieved when two important physical processes snowmelt and freeze-thaw are included into the model. Thus, it is certainly expected that GXAJ-S-SF will outperform GXAJ in a region that experiences the S and SF processes. Furthermore, since GXAJ consistently underestimates the runoff, and since the physical processes modeled in SF can only increase the runoff but not decrease it, it is a foregone conclusion that upon calibrating SF you will arrive at a better fit for GXAJ-S-SF than GXAJ. As far as I can tell, there are no novel or interesting findings regarding hydrological processes in this manuscript, nor are there meaningful analyses about the utility and information content of the hydrological models used beyond the goodness-of-fit metrics NSE, RBE, RMSE. Therefore, I recommend that after major revisions addressing the concerns I have elaborated below, this manuscript could be suitable for publication as a technical note.

**Response 1:** Thank you for your evaluation and detailed feedback on our manuscript. We highly value your comments regarding the novelty and contributions of the study, which has helped us further clarifying the innovative aspects and novel results of our research during the revision process.

For instance, we now state upfront in the introduction - based e.g. on the topical systematic review of Ala-Aho et al. (2021) - that the impact of SFG on runoff processes has been shown to be profound in many small-scale applications. This would indeed suggest that improved performance of the GXAJ-S-SF model, which incorporates snowmelt and freeze-thaw processes, over the original GXAJ model, which neglects such processes, may be somewhat expected. However, we now also clarify that large knowledge gaps remain, e.g. regarding the complex and less clear impacts of SFG on runoff in large basins (e.g., Ala-Aho et al., 2021). At such larger scales, the question has hence remained relatively open regarding the required model complexity for capturing dominant hydrological processes and producing sufficiently accurate runoff simulations in presence of snow and SFG. The present systematic analyses of the performance of models of different complexity contribute to addressing this knowledge gap, as now emphasized e.g. in the

manuscript's revised title, as well as in the discussion and conclusion sections. Overall, we believe the main innovations of this study lie in the following aspects:

- **Innovation in the snow-freeze-thaw coupling approach:**

A key aspect is how the physical mechanisms of snowmelt and freeze-thaw cycles are coupled the model, in a way that enables quantitative analyses of the impacts of snowmelt and  frozen ground on runoff, soil moisture dynamics and evapotranspiration. In particular, the developed snow-freeze-thaw coupling method has clear physical significance, which supports the use of a relatively low number of additional (fitting) parameters.

- **Assessment of dominant hydrological processes in a large basin subject to SFG:**

 In the light of the above-mentioned considerable knowledge gaps on large-scale impacts of SFG on runoff, an additional novel aspect of the manuscript is related to the performed systematic comparison between simplified models (having no combined snow-SFG extensions, or accounting for snow processes only) and extended models that account for combined impacts of snow and SFG. As explained in the revised introduction, this comparison aims at increasing the understanding regarding to which extent SFG processes play a significant role in large basin runoff, e.g. providing guidance regarding the necessary level of complexity in predictive models. These quantitative results for instance show that SFG can indeed significantly increase large basin runoff during cold months (with an increase of 39–77% compared to models that neglect SFG) while reducing interflow and groundwater runoff.

- **Integration of process understanding and practical application:**

By analyzing and quantifying the effects of varying frozen soil depths and their spatiotemporal distributions on hydrological processes, this study highlights the complex feedback mechanisms of frozen ground on hydrological systems. These analyses not only deepen our understanding of the dynamic interactions among snow, freeze-thaw, and hydrological processes but also provide important references for predicting hydrological changes under future climate change scenarios in cold mountainous regions.

**Comments 2:** Would it not be more meaningful to compare GXAJ-S-SF to a different hydrological model that also includes snowmelt and seasonal freeze-thaw? For example one of those models you mentioned in L96 – 112. Even if an actual model comparison is not done, it would be useful

to discuss the differences and similarities in model processes between GXAJ-S-SF and other similar models with snowmelt and freeze-thaw functions.

**Response 2:** Thank you very much for your valuable suggestions on our research!

We follow your advice and have extended the discussion section with comparisons of the present GXAJ-S-SF model processes with other hydrological models that incorporate snowmelt and freeze-thaw processes. Therbey we elaborate on the similarities and differences in hydrological process simulations among the different models. This also contributes to clarifying the GXAJ-S-SF model's innovations in representing hydrological processes.

**Comments 3:** It would be more rigorous to re-run the models with different priors. For example, there could be a configuration of GXAJ, with soil property related parameters set at an "annual average effective value" taking into account that the soil is frozen for 9 months of the year. This hypothetical configuration of GXAJ could possibly produce results as good as GXAJ-S-SF, but it is possible that this configuration of GXAJ was not tested because the optimization algorithm was stuck in a local minimum. Given the highly nonlinear processes involved in this model, I think that calibrating from a single set of priors may be insufficient.

**Response 3:** Thank you for your valuable comments on our research. We greatly appreciate your feedback and have addressed the relevant issues with detailed revisions and additions as follows:

**On Optimization Algorithm and Parameter Settings:**

Regarding your concerns about model optimization and the use of "prior parameters," we understand the potential limitations of calibrating the model using a single prior parameter configuration. In our study, we employed the SCE-UA optimization algorithm to precisely calibrate key parameters and obtain the optimal solution for the model. To avoid the risk of local optima, we set a range of prior parameters and randomly selected different configurations within the allowed range, running the optimization algorithm multiple times. While this approach was not explicitly described in the manuscript, we will supplement the relevant content to clarify that we considered different prior parameters during optimization. Through this approach, we enhanced the model's stability and minimize reliance on a single configuration, addressing the limitations you raised.

**On Soil Characteristics and Related Parameter Settings:**

We fully agree with your observation that the presence of frozen ground significantly alters soil moisture dynamics, which, in turn, affects the storage capacities of soil tension water and free water and their spatial-temporal distribution. In our study, we did not rely on a simplistic "annual effective value" to account for frozen ground effects. Instead, we dynamically adjusted the distribution of these underlying surface parameters by comparing the depth of frozen ground (characterized by spatial-temporal heterogeneity) with the corresponding soil layer thickness (as categorized in four specific cases in the methodology). This adjustment supports the model in reflectíng key effects of frozen ground. Through this approach, we fully considered the spatial-temporal heterogeneity induced by frozen ground, thereby improving the accuracy of the simulation results. In the revised manuscript, we will further elaborate on this section, detailing how changes in frozen ground depth dynamically influence soil layer thickness and related parameters, enhancing the model's capacity to simulate frozen ground dynamics.

We believe that these revisions will better demonstrate the scientific validity and rationality of our approach, and we thank the reviewer for their insightful suggestions.

**Comments 4:** L409 – 417: "The accuracy in simulating the initial freeze and initial thaw dates was validated against ground temperature data from meteorological stations within the basin (Fig. S5), indirectly confirming the simulated soil freeze-thaw processes."

Could you provide citations or a more detailed discussion to support the validity of this point? Since freezing and melting both start from the top, and since the temperature data for verification was measured at the ground surface, simulating the correct initial freeze and initial thaw dates does not help confirm that the model has simulated the freezing depth correctly over the 9 months with frozen soils.

**Response 4:** We have identified several studies supporting the use of surface temperature to validate the initial freeze and thaw dates, providing a theoretical basis for applying this method in regions without observational data on frozen soil depth. However, due to the lack of measured frozen soil depth data in the study area, we are currently unable to directly validate the simulated frozen soil depth. Consequently, this study primarily uses available surface temperature data to verify the initial freeze and thaw dates, thereby indirectly supporting the reliability of the simulated freeze-thaw processes.

To further evaluate the model's reliability, we compared the spatial distribution of the maximum frozen soil depth simulated in this study with data from the Tibetan Plateau Permafrost Dataset (1961–2020) for the 2000s. The comparison revealed a high degree of consistency in both spatial distribution patterns and magnitude, with a correlation coefficient of 0.89. These results provide additional validation support, and we will present these comparative analyses in the revised manuscript.

We also recognize that using surface temperature to validate freeze-thaw processes introduces some uncertainty, as the freezing and thawing processes propagate downward from the surface, and these data only partially reflect the dynamics of deeper frozen soil layers. To address this, we will expand the discussion section to further analyze this uncertainty and its potential impacts. At the same time, leveraging available data resources to validate hydrological processes remains practically meaningful in data-scarce regions. This approach provides a robust foundation for supporting the regional applicability of the model.

**Comments 5:** I think that the "modular approach" that you emphasize several times, including in the abstract and conclusion, is reinventing the wheel as it is just another name for loose coupling or one-way coupling, which is a basic hydrological concept.

**Response 5:** Thank you for your valuable comments and for providing us with the opportunity to clarify the term "modular approach." After carefully considering your feedback, we agree that the term "modular" may overlap to some extent with concepts such as loosely coupled or unidirectional coupling, especially in the context of hydrological modeling. However, our use of "modular" aims to emphasize the flexibility, scalability, and reusability of the model components. Specifically, the snow and frozen ground modules in our study were designed as independent components that can be enabled or disabled depending on environmental conditions (e.g., the presence of snow or seasonally frozen ground). These components are not hard-coded into the GXAJ model; instead, they can be integrated into other hydrological frameworks without requiring significant modifications to the core structure of the model. This design approach enhances the model's flexibility and adaptability, allowing researchers to extend or modify it to suit different cold-region environments.

To address your comments, we will revise the manuscript to avoid potential misunderstandings caused by the term "modular approach." Instead, we will describe the design philosophy of the

model components more accurately. In the abstract and conclusion, we will refer to the approach as "flexible and adaptable" rather than "modular," ensuring that the core idea of the design is conveyed clearly and without ambiguity.

We hope this clarification and the corresponding revisions will adequately address your concerns.

**Comments 6:** After reading through the manuscript several times, I recognize that the bulk of the scientific contribution of this manuscript lies in the freeze-thaw process module in section 2.2.2. As shown in the results, it fits well with the measurements. However, I think some parts should be explained more clearly. What is the purpose of using two different representations of the soil layers in one model? Why not use the same layers for the computation of runoff, moisture and ET (Figure S1)? Does this mean that in the simulations, the humus layer could sometimes overlap with both the "upper soil" and part of the "lower soil"? And can the "upper soil" sometimes overlap with both the humus layer and the vadose zone? Does this not then imply that you need to interpolate some effective soil parameter values that may be inappropriate for the actual individual soil layers? How would this affect the runoff and discharge predictions? Furthermore, wouldn't this mean that the parameter values you calibrate from field data do not have a proper physical meaning? I think that in order to reconcile the two different representations of the soil layers, it is inevitable that the calibrated parameter values are smoothed interpolations of the values that would actually describe each individual soil layer.

**Response 6:** Thank you for your positive feedback on the freeze-thaw process in section 2.2.2 and for your valuable suggestions. Regarding the use of two different soil layer representations in the model, we adopted this approach based on the design philosophy of the original GXAJ model. We did not provide sufficient explanation in the original manuscript, which caused some confusion, and we will clarify this in the revised version.

Specifically: When calculating runoff for a grid cell, soil saturation refers to the soil water content reaching the field capacity, not the saturation water content. The GXAJ model uses a saturation runoff mechanism, meaning that runoff only occurs when the soil's unsaturated zone reaches field capacity. Before this point, all incoming water is absorbed by the soil without generating runoff. In the GXAJ model, the tension water storage capacity ($W_M$, in mm) of a grid cell is determined by the watershed's topography, as well as soil, vegetation, and other surface conditions. We do not consider the uneven distribution of tension water content within the grid cell. To calculate the

actual precipitation ($P_e$) available for runoff, we subtract the evaporation, canopy interception, and river precipitation from the measured rainfall during the calculation period, then check if upstream inflow replenishes the soil water content of the current grid cell.

When calculating the sources of runoff (Figure S2(a)), the runoff in the grid cell is divided into three components: surface runoff, interflow, and groundwater runoff. The GXAJ model treats the upper soil layer in the unsaturated zone as the humus layer (determined by topography, soil, vegetation, and other surface conditions), with the bottom of the humus layer considered a "relatively impermeable layer." Some of the runoff generates interflow, while part continues to percolate, generating groundwater runoff. When the free water in the humus layer reaches saturation, surface runoff is produced. Similarly, we do not consider the uneven distribution of free water storage in the grid cell.

In summary, the GXAJ model (Yao et al., 2012) calculates the tension water storage capacity ($W_M$) in the unsaturated zone (Figure S1) and the free water storage capacity ($S_M$) in the humus layer to divide runoff into three components: $R_s$, $R_i$, and $R_g$. $W_M$ determines whether runoff occurs and the amount of runoff (saturation excess runoff), while the free water content in the surface soil splits runoff into $R_i$ and $R_g$. When the free water content reaches saturation, $R_s$ is produced, as shown in Figure S2(a).

For evapotranspiration (Figure S2(b)), the GXAJ model uses a three-layer evapotranspiration model, dividing the soil (vadose zone) into upper, middle, and lower layers, with corresponding tension water storage capacities: $W_{UM}, W_{LM}$ and $W_{DM}$ (in mm). During actual evapotranspiration calculation, canopy interception is evaporated based on evapotranspiration capacity. When the intercepted water is less than the evapotranspiration capacity, the three-layer model is applied. The calculation principle is that the upper layer evaporates according to its evapotranspiration capacity. If the upper layer cannot supply enough water for evapotranspiration, the remaining capacity is drawn from the middle layer, with evapotranspiration in the middle layer proportional to the remaining capacity and inversely proportional to the middle layer's storage capacity. The ratio of middle layer evapotranspiration to the remaining capacity cannot be less than the deep layer evapotranspiration coefficient, $C$. If the middle layer cannot supply enough, the deep layer water will supply the deficit. The corresponding soil moisture and evapotranspiration are labeled as $W^u$, $W^l$, and $W^d$, and $E^u$, $E^l$, and $E^d$.

The original GXAJ model used different soil layers to better simulate the role of soil layers at different depths in hydrological processes. This means that the humus layer may overlap with the "upper soil" and part of the "lower soil," as you understand, but the specific situation may vary depending on the surface conditions. However, the soil surface parameters, such as tension water storage capacity and free water storage capacity, are derived from the physical properties of the soil (e.g., soil type and structure), as well as topography and vegetation, and thus have physical significance. Applying these concepts to a single-layer soil system would simplify the calculation, treating the entire soil layer as the unsaturated zone for runoff calculation and using a single-layer evapotranspiration model.

I hope this explanation resolves your doubts, and we will further improve the manuscript in the revision.

**Specific comments**

**Comments 7:** L192: Is saturation excess runoff a reliable way to partition snowmelt fluxes, which are fast and may often exceed the infiltration capacity?

**Response 7:** Thank you for the valuable feedback. The melting rate of snowmelt water is usually fast, and in the presence of a frozen soil surface, the permeability of the soil is limited, which easily leads to surface runoff. We fully agree with this point, and it has been thoroughly considered in Sections 2.1 and 2.2.1 of our study. In our model, we specifically focused on the interaction between water movement and soil during the snowmelt process, considering the potential freezing of the soil surface. When the snowmelt water encounters the frozen soil layer, due to the low permeability of the frozen soil, the snowmelt water cannot rapidly infiltrate into the soil, which results in significant surface runoff. The detailed consideration of this process in the model ensures the rapid generation of runoff from the snowmelt water.

**Comments 8:** L277: If you divide the SFD by the cube root of ASD to get SFD*, then the units of SFD* are $[cm]^{2/3}$. What does that physically mean?

**Response 8:** Thank you for your attention. The empirical formula for frozen soil depth used in our study is derived from the research "Influence of snow cover on soil freeze depth across China," which utilizes observational data from 378 meteorological stations across China (1980–2014). This study quantified the relationship between snow cover and the maximum seasonal freeze depth

(MSFD), as well as the contribution of snow cover to MSFD. The results indicated that in areas with thin snow cover or short snow duration, the impact on freeze depth is minimal. However, in regions with thick snow and longer snow duration, the snow cover reduces the frozen soil depth, and this relationship can be reasonably reflected by dividing the freeze depth (SFD) by the two-thirds power of the snow depth (ASD). Although this formula does not have a strict physical unit explanation, it has demonstrated high accuracy in multiple station validations, making it a practical method for describing the influence of snow cover on frozen soil depth.

**Comments 9:** L335: Please be consistent with terminology, do not interchangeably use primary parameters and major parameters.

**Response 9:** Thank you for the reviewer's correction. We will unify the terminology in the revised manuscript to avoid confusion between 'primary parameters' and 'major parameters' and ensure consistent expression.

**Comments 10:** L346 – 349: It would be helpful to mark in Figure 2 which processes in SNOW17 were calibrated with measured data, and which were not.

**Response 10:** Thank you for your suggestion. In the revised manuscript, we will annotate Figure 2 (the SNOW17 model diagram) to indicate which processes were calibrated using observational data and which were not, in order to more clearly present the calibration scope and methodology of the model.

**Comments 11:** L349 – 352: I am not sure what this actually means. You definitely need more parameters for GXAJ-S than GXAJ, because you are adding physical processes. Are you saying that just because the -S module is compartmentalized in a module that means that you do not add more parameters to GXAJ? I think that this is a confusing way to describe one-way coupling.

**Response 11:** Thank you for the valuable comments. In our study, the goal is to improve the model performance by introducing new physical processes while minimizing the introduction of unnecessary additional parameters. Specifically, regarding the issue of "increasing parameters" that you mentioned, we provide the following explanation:

On the issue of increasing parameters: Indeed, after introducing snowmelt (-S) and freeze-thaw (SF) processes, the model requires additional parameters to describe these physical processes.

However, during the model improvement process, although new physical processes were introduced, we did not add extra adjustable parameters to the model. To maintain model simplicity, we adopted a fixed-parameter strategy during the model construction. Specifically, when introducing the snowmelt process, the related parameters were fixed after initial calibration and were not adjusted further. The aim of this strategy is to ensure that the model improvements are achieved solely by introducing new physical processes, rather than by adding new free parameters. Freeze-thaw process parameters: For the freeze-thaw cycle process, we used empirical parameters, which are also fixed in the model and were not introduced as additional degrees of freedom during the coupling process. Therefore, although the model considers more physical processes, by fixing the parameters of these processes, we avoid adding new adjustable parameters to the model.

By using this approach, we ensure improved model performance while maintaining consistency and simplicity in the model parameter settings. This approach prevents unnecessary parameter additions and emphasizes the technical improvements brought by the introduction of physical processes.

We will further clarify this point in the revised manuscript to ensure a clearer understanding of the model parameter settings. Once again, we appreciate the reviewer's attention to our work and their feedback.

**Comments 12:** L402 – 404: I think that the evidence of robustness is that the model *did not perform worse* during the validation period. Performing *better* during the validation period is not evidence of robustness. Conversely, performing *better* during the validation period suggests that you made some assumptions about the physical processes hard coded into the model, that were more valid during the validation period. Please discuss this in more detail if possible.

**Response 12:**

Thank you for the reviewer's comments. We would like to clarify that no new assumptions were introduced during either the calibration or validation periods. The model is based on four main parameters, all of which have clear physical meanings and remain consistent throughout the study. These specific parameters are:

- SCF (Snowfall Correction Factor)
- MFMAX (Maximum Snowmelt Factor during Non-Rainfall Period)
- MFMIN (Minimum Snowmelt Factor during Non-Rainfall Period)

- UADJ (Average Wind Speed Factor during Rain-Snow Period)

The better performance during the validation period is not due to any pre-set assumptions in the model, but may be related to the simpler hydrological conditions during this period. Specifically, the snow depth during the validation period was smaller, which simplified the complexity of snowmelt and freeze-thaw processes compared to the calibration period. Under shallower snow conditions, the errors introduced by complex snowmelt-freeze-thaw interactions, which occur under deeper snow, were significantly reduced. As a result, the model was able to more accurately capture these relatively simple hydrological processes, leading to better performance during the validation period.

Although this is a possible explanation, we acknowledge that other factors may have contributed to the model performance. If necessary, we can explore this further. However, we believe that the model's performance during both periods demonstrates a certain level of stability and its ability to adapt to different snow conditions and simulate snowmelt and runoff processes.

We will add a clarification in the revised manuscript, explaining that the improvement in model performance during the validation period may be related to the simplified hydrological conditions. If you have any further suggestions, please let us know.

**Comments 13:** Figure 4: What are the dashed lines?

**Response 13:** Thank you for the reviewer's question. The dashed line in Figure 4 represents the trend of snow depth changes. We will provide a more detailed explanation of the legend in the revised manuscript to avoid confusion and ensure that the information in the figure is clearer and more intuitive for the readers.

**Comments 14:** Figure 5: What is the dashed line?

**Response 14:** Thank you for the reviewer's question. The dashed line in Figure 5 represents the trend of permafrost depth changes. We will provide a more detailed explanation of the legend in the revised manuscript to avoid confusion and ensure that the information in the figure is clearer and more intuitive for the readers.

**Comments 15:** L427: You earlier defined an RBE, but not an RE.

**Response 15:** Thank you for the reviewer's reminder. We noticed that the definition of RE (Relative Error) was not properly provided in the original manuscript. In the revised version, we will correct its definition and description, ensuring consistency in the use of terminology to avoid confusion for the readers.

**Comments 16:** L539 – 534: I think that the formation of a saturated layer above ground under these circumstances is possible only for very coarse soils that are inefficient at soil water redistribution. This is unlikely to be a general behavior. If you are referring to a specific soil type, please describe it. If you are claiming this as a general behavior, please provide references.

**Response 16:** Thank you for the reviewer's valuable suggestion. Regarding the phenomenon of a saturated layer forming above the surface under freeze-thaw conditions, we believe this phenomenon is mainly due to the low permeability of frozen soil, which causes runoff from snowmelt or rainfall to accumulate above the frozen layer. When the upper frozen layer is thin, moisture tends to accumulate at the freeze-thaw interface, forming a saturated layer. This phenomenon has been mentioned in many studies, especially during the freeze-thaw period, when the accumulation of moisture at the freeze-thaw interface can lead to the formation of a saturated layer above the frozen ground. Furthermore, some studies, such as "What conditions favor the influence of seasonally frozen ground on hydrological partitioning? A systematic review," suggest that soil type is unlikely to be the determining factor influencing the hydrological response of seasonally frozen ground, even though soil type is important for overall hydrological responses. We will include the relevant references in the revised manuscript to further support this argument.

**Comments 17:** L540: If matric potential is the primary driver of moisture movement, then how does gravity cause a saturated layer to emerge at the frozen interface?

**Response 17:** Thank you for the reviewer's thoughtful question. We understand the concern regarding the relationships between the driving factors of soil water movement. During the freezing period, soil water movement in the unsaturated zone is influenced not only by matric potential but also primarily by temperature potential. During the thawing period, water movement is controlled by matric potential, gravitational potential, and temperature potential. Above the freeze-thaw interface, water moves upward and evaporates due to matric potential, while

gravitational water moves downward, accumulating and filling soil pores at the thaw interface. This process results in the formation of a saturated layer above the frozen ground.

In the revised manuscript, we will clarify this point more explicitly. We appreciate the reviewer's detailed feedback, which has helped us better explain this complex hydrological process.

**Comments 18:** L548 – 549: Which processes are you referring to, and what impacts? Are the processes you study not already naturally part of the local hydrological cycle and ecosystem?

**Response 18:** Thank you for your valuable feedback. The "processes" mentioned in the text, including freeze-thaw dynamics, soil water movement, and the impact of snow and seasonally frozen ground (SFG) on evapotranspiration, are indeed integral parts of the local hydrological cycle and ecosystem. However, we would like to emphasize that this study specifically focuses on how these processes—especially the freeze-thaw cycle and its effects on soil moisture and evaporation rates—can vary under different environmental conditions.

For instance, during the freezing period, frozen ground and snow cover suppress evapotranspiration, potentially significantly reducing water supply to plants. In contrast, during the thawing period, the formation of a saturated layer above the frozen soil may alter soil permeability and runoff patterns. These changes are particularly important in the context of climate change, as variations in the freeze and thaw periods could impact water resources and the stability of ecosystems over the long term.

In the revised manuscript, we will clarify these points more explicitly and further explain the potential impacts of these processes.

**Comments 19:** L580 – L590: I feel that this is self-evident. It is a rehash of the widely known problem that data-calibrated hydrological models are often 'right for the wrong reasons'. It is a nice discussion that fits the work done, but does not contribute new knowledge.

**Response 19:** Thank you for your valuable feedback on our discussion. We understand your point, and indeed, it is widely recognized in the field of hydrology that "models calibrated with existing data may perform well under specific conditions but may not necessarily reflect future changes." However, we believe that in this study, by considering the effects of frozen ground and snow on hydrological processes, we are able to physically address the limitations of such models, providing

a more robust and physically consistent framework for hydrological simulations under future climate change scenarios.

Therefore, in the revised manuscript, we will further emphasize the background of these issues and highlight the advantages of the model, especially its potential for predicting hydrological changes under future climate conditions. We will also elaborate on the importance of the model in forecasting future hydrological processes, particularly in the context of climate change.

**Comments 20:** L600 – 602: This argument is valid only if the modeled processes are linear. The processes you have modeled are potentially too nonlinear and have too many interactions for this argument to hold.

**Response 20:** Thank you for your valuable feedback. We fully understand your point, and indeed, for hydrological processes with strong non-linearity and interactions, assuming that errors are completely canceled out is an oversimplification. In the revised manuscript, we will modify the relevant sections based on your suggestions and further discuss these non-linear processes and their impact on error propagation. We will emphasize that, although errors may not be entirely canceled out, since both models use the same snow depth data, the error impacts are likely to be relatively consistent. Therefore, we can still draw meaningful conclusions from the model performance comparison.

Once again, thank you for your thoughtful comments, which have helped us improve the paper.

**Comments 21:** L603 – 604: I agree that remote sensing errors would probably not affected the core conclusions of this manuscript, but not for the reasons you provide in L600 – 602. As I explained in my general comments, I think your conclusions are mostly self-evident.

**Response 21:** Thank you for your valuable feedback on our manuscript. Regarding your comment that "most of the conclusions are self-evident," we understand your perspective and will further clarify the innovation and contributions of this study to address this point.

While we believe the impact of remote sensing data errors on the core conclusions is minimal, we recognize that model output errors are not entirely linear, and the effect of remote sensing errors may not fully cancel out. As you pointed out, the model processes themselves have nonlinear

characteristics, so the influence of remote sensing errors cannot be ignored. In the revision, we will more clearly explain the sources and potential impacts of errors, and emphasize that, despite some uncertainty, our core conclusions are still validated through comparisons with other similar models.

Regarding your point about "self-evident," we understand your view that it is reasonable for the GXAJ-S-SF model to perform better in cold regions than the GXAJ model, which does not consider snowmelt and freeze-thaw processes. However, we believe the innovation of this study lies not just in the "reasonableness" of the results, but in our development of a method that systematically couples snowmelt, frozen ground, and hydrological processes. Additionally, we quantitatively analyze the spatiotemporal dynamic impacts of seasonally frozen ground and snow on hydrological processes. These quantitative analyses and deeper understanding of hydrological processes are important additions to existing models and predictive methods.

In the revision, we will further emphasize this innovation and include more technical discussions and process analyses to ensure the contributions of the paper to hydrological modeling in cold regions are clearly presented.

Once again, thank you for your valuable comments, and we look forward to your further guidance.

**Comments 22:** L606 – 615: The benchmark model GXAJ you refer to is not a different model, but it is just GXAJ-S-SF without the snow and freezing capacities. This discussion is not meaningful because it is self-evident.

**Response 22:** Thank you for your feedback. We understand your point about the "GXAJ model not being a distinct model but merely a version of the GXAJ-S-SF model without the snow and frozen ground capabilities." We agree that in certain cases, the difference between the base and the improved model might seem quite straightforward.

However, the reason we refer to the GXAJ as the baseline model is to clearly illustrate the impact of snow and frozen ground processes on hydrological simulation results. While the comparison between the GXAJ, which does not account for these processes, and the GXAJ-S and GXAJ-S-SF models, which do, may seem self-evident, this comparison is still crucial in highlighting the contribution of the newly added features to the hydrological process simulation. Specifically, this

comparison allows us to emphasize the improvements in model simplification, computational efficiency, and the adaptation to hydrological processes in cold regions.

In the revision, we will further clarify this discussion, focusing on the innovation of our study— how the introduction of snow-frozen ground coupling processes enhances the model's ability to simulate the impacts of seasonally frozen ground. We will refine this section to ensure that the contributions of our research are presented more clearly and meaningfully, avoiding redundant conclusions.

Once again, thank you for your valuable suggestions. We will make the necessary adjustments in the revised manuscript.

**Comments 23:** L615 – 617: What is the modular approach being contrasted against? What results did you show that support this statement?

**Response 23:** Thank you for your valuable feedback. Regarding the issue of "comparing the modular approach with other methods," you have pointed out an important detail. We understand your concern and will further clarify our position in the revision.

In the original text, the term "modular approach" referred to the integration of individual physical processes (such as snow, frozen ground, and hydrological processes) as separate modules in the model, facilitating future expansion and improvement. However, in the revised version, we have moved away from further modularization and emphasized the model's superiority in achieving simplicity, computational efficiency, and adaptability to various environmental conditions.

We will revise this section to explicitly clarify how our choice of a "simplified design" approach, rather than a modular structure, better supports the model's broad applicability, especially in the diverse environments of cold regions. The revised description will highlight the model's strong performance, ease of implementation, and low data requirements, demonstrating the advantages of this approach for practical applications.

We will further refine this part to ensure that our research contributions and innovations are clearly explained. Thank you again for your feedback, and we will make the necessary improvements based on your suggestions.

**Comments 24:** L630 – 634: This is a great point, and could be expanded to make the discussion more interesting.

**Response 24:** Thank you for your valuable suggestion. We fully agree with your point that soil and geological complexity is indeed a critical factor in watershed modeling, significantly impacting the model's applicability and accuracy. In the revised manuscript, we will expand on this discussion and emphasize the necessity of recalibrating model parameters based on different watershed characteristics, such as soil type, moisture retention capacity, topography, and vegetation cover.

Your suggestion has been very helpful in enriching this part of the discussion, and we will ensure that these important factors are adequately addressed in the revision. Thank you once again for your constructive feedback.

---

## Author Comment (AC2)

**General comments**

**Comments 1:** The authors present a modeling study on the hydrological impacts of snow and frozen ground dynamics in a topographically complex basin. The topic of cryospheric changes and their impacts on hydrology is both significant and timely. However, the authors should address several key issues in the current manuscript to enhance its quality before it can be considered further.

I think the novelty of this study is not sufficiently distinctive or well-highlighted. There have already been numerous modeling studies on snow and frozen ground dynamics in the Tibetan Plateau region, both the basin-scale and regional-scale studies are conducted. Moreover, the models employed in previous studies provided more advanced representations of snow and frozen ground processes, particularly in terms of frozen ground dynamics, compared to the model used in this study. Therefore, the authors need to consider how to better emphasize the unique contributions of this study in comparison to prior research.

**Response 1:** Thank you for your thoughtful evaluation and constructive feedback. We appreciate your comments regarding the novelty of our study, which have prompted us to clarify and highlight the unique contributions of our research more effectively.

We believe the uniqueness of this study is reflected in the following aspects:

- **The hydrological impacts of snow and frozen ground in large basins**

As you mentioned, the impacts of snow and frozen ground on runoff processes have been confirmed in many small-scale studies. However, we would like to further clarify that significant knowledge gaps remain regarding the complex and less well-understood effects of seasonally frozen ground (SFG) on runoff in large-scale basins (e.g., Ala-Aho et al., 2021). Therefore, at such a large scale, the question of how model complexity influences the ability to capture key hydrological processes and produce sufficiently accurate runoff simulations under the presence of snow and SFG remains relatively unresolved. This study aims to fill this knowledge gap through a systematic analysis of the performance of models with different levels of complexity. We will further emphasize this point in the revised title, discussion, and conclusion sections.

- **A simple and data-efficient snow and freeze-thaw coupling method**

A key innovation of this study is the integration of the physical mechanisms of snowmelt and freeze-thaw cycles into the hydrological model, enabling the quantitative analysis of the impacts of snowmelt and frozen ground on runoff, soil moisture dynamics, and evapotranspiration. The

developed snow and freeze-thaw coupling module is physically meaningful, requires relatively few additional parameters, and has low dependence on input data. This feature is particularly important for cold regions, where data is often limited.

35  •  **Quantitative assessment of the impacts of SFG on hydrological processes in large basins**

Given the significant knowledge gaps regarding the large-scale impacts of SFG on runoff, another novel aspect of this study lies in the systematic comparison of simplified models (without coupled snow-SFG modules or only considering snow processes) and extended models that include the

40  combined effects of snow and SFG. Such comparisons enhance our understanding of the extent to which SFG processes play a role in large-basin runoff and provide guidance on the necessary level of model complexity. For example, our quantitative results indicate that SFG can significantly increase surface runoff in large basins during cold months (by 39%-77% compared to models that ignore SFG) while reducing interflow and groundwater runoff. These findings will be further

45  highlighted in the revised introduction and discussion sections.

•  **Combining hydrological cycle process understanding with practical applications**

Through the analysis and quantification of frozen soil depth and its spatiotemporal distribution impacts on hydrological processes, this study reveals the complex feedback mechanisms of frozen ground on hydrological systems. These analyses not only deepen our understanding of the dynamic

50  interactions between snow, freeze-thaw processes, and hydrological processes but also provide critical references for predicting hydrological changes in cold mountainous regions under future climate change scenarios.

We will further emphasize these unique contributions in the revised manuscript. Once again, we sincerely thank the reviewer for your valuable comments.

55

**Specific comments**

**Comments 2:** In Figure 3c, it is evident that a significant portion of the study area is covered by permafrost. However, the Stefan model mentioned in the methodology is designed to model seasonal frozen ground. Did the authors separately account for the dynamics of permafrost in their

60  study? If not, this could be a critical limitation that needs to be addressed or clarified.

**Response 2:** Thank you for highlighting this important aspect. As shown in Figure 3c, the study area is dominated by seasonal frozen ground, while permafrost accounts for less than 10% and is

sparsely distributed along the edges of the study area. Therefore, our study primarily focuses on seasonal frozen ground, and the improved module is more efficient in regions dominated by seasonal frozen ground. The simulation results proved that the model achieves high accuracy in these areas.

However, we acknowledge that even a small portion of permafrost may influence hydrological processes and runoff simulations. We will further discuss this potential impact, as well as the limitations of the proposed model in permafrost regions and the uncertainties introduced by this limitation, in the revised manuscript. Thank you for bringing this to our attention, which will help us further strengthen our study.

**Comments 3:** Line 272-275: How was this threshold 30cm determined? Was a sensitivity analysis conducted to assess the impact of this threshold on the results? Providing such an analysis would help evaluate the robustness of the study's findings.

**Response 3:** Thank you for the valuable question! The 30 cm threshold mentioned in Lines 272-275 is based on findings from previous studies. Many studies have explored different snow depth thresholds. For example, Brooks et al. (1995, 1999) and Cline (1995) suggested that when snow depth reaches 30-40 cm, air temperature is unlikely to significantly affect ground temperature. Building on this, Hill (2015) proposed a conceptual model indicating that thick snow cover (>30 cm) effectively insulates the ground, keeping it thawed year-round and enabling groundwater recharge. This also leads to an earlier hydrological response compared to thin snow cover (<30 cm), where the ground may remain seasonally frozen during the snowmelt season, limiting groundwater recharge and resulting in a delayed hydrological response later in the summer. In our study, we adopted Hill's (2015) proposed 30 cm threshold for snow depth, supported by the above literature. We will include the relevant references and further elaborate on this background in the revised manuscript. We sincerely thank the reviewer for highlighting this important point!

**Comments 4:** In Table 3, the authors utilized several data products from other studies. However, the accuracy of these datasets, particularly the snow depth data, which is critical for this study, has not been clarified.

**Response 4:** Thank you for pointing out this important aspect. We acknowledge that the accuracy of the datasets used, particularly the snow depth data, is critical for this study. In the revised

manuscript, we will include additional details about the accuracy and validation of the data products utilized, especially the snow depth dataset. Furthermore, we will expand the discussion section to address the uncertainties introduced by the snow depth data and their potential impact on our findings.

We sincerely appreciate the reviewer's insightful suggestion, which will help improve the robustness and clarity of our study.

**Comments 5:** Line 309-404: In points with high snow depth, there are significant discrepancies between the model results and the remote sensing data. The authors should investigate the underlying causes of these differences.

**Response 5:** Thank you for your valuable comment. We used the SNOW17 model to simulate snow depth and compared the results with remote sensing data. We noticed discrepancies between the model results and the remote sensing data in areas with high snow depth. One potential reason is that hydrological processes in areas with high snow depth are more complex. The model employs simplified parameterization methods to simulate snow accumulation and melting processes in these regions, which may not fully capture the spatial heterogeneity of snow processes. Additionally, remote sensing data may have limitations in capturing extreme snow depths, such as signal saturation or terrain occlusion in mountainous areas, which could introduce errors.

These factors may also explain the lower simulation accuracy of spring runoff shown in Figures 6 to 8. However, despite the discrepancies in areas with high snow depth, the calibration and validation results demonstrate relatively low RMSE and BIAS values, indicating that the model performs well overall in simulating snow depth dynamics, particularly in areas with moderate snow depth. Furthermore, the improved model shows significant advancements in simulating snowmelt runoff compared to the original model.

We acknowledge that snowmelt is a complex hydrological process, and under limited data conditions, we strive to utilize available observational and remote sensing data to simulate the snowmelt process with higher accuracy. While certain limitations and errors are inevitable, we believe that such efforts are meaningful and valuable, particularly in cold regions where data scarcity presents significant challenges.

In the revised manuscript, we will further investigate and discuss the potential causes of these discrepancies, including the limitations of both the remote sensing data and the model itself. Additionally, we will include a more comprehensive discussion in the uncertainty analysis section to address these issues and propose directions for future improvements.

**Comments 6:** Line 412:415: For the simulation of frozen ground processes, verifying only the accuracy of the initial freeze and initial thaw dates is far from sufficient. It is also essential to validate the simulated soil temperature and soil moisture (including liquid water content and soil ice content). These variables are key to understanding how freeze-thaw processes influence basin hydrology. Therefore, the authors should provide validation results for these variables to demonstrate the reliability of the study's findings.

**Response 6:** Thank you for your valuable comments. Due to the lack of measured frozen ground depth data, soil temperature, and soil moisture (which is common in most cold regions), we are currently unable to directly validate the simulated frozen ground depth and freeze-thaw processes. Therefore, this study primarily uses available surface temperature data to validate the initial freeze and thaw dates, thereby indirectly supporting the reliability of the simulated freeze-thaw processes, a method that has been supported by several studies.

To further assess the reliability of the model, we compared the spatial distribution of the maximum frozen ground depth simulated in this study with the 2000s Tibetan Plateau Permafrost Dataset (1961–2020). The comparison results show a high level of consistency in both spatial distribution patterns and magnitudes, with a correlation coefficient of 0.89. These results provide additional validation support for the model, which we will present in the revised manuscript.

We also acknowledge that using surface temperature to validate the freeze-thaw process introduces some uncertainty, as the freeze-thaw process propagates from the surface downward, and surface temperature data only partially reflect the dynamics of deeper frozen layers. To address this, we will expand the discussion section to further analyze this uncertainty and its potential impacts. At the same time, utilizing existing data resources to validate hydrological processes in data-scarce regions remains highly meaningful. This approach provides a strong foundation for supporting the model's applicability in regional assessments.

**Comments 7:** The 'Results' section is too brief and lacks depth in describing the characteristics of snow and frozen ground changes and their hydrological effects. For instance, there is insufficient discussion on how frozen ground processes alter soil temperature and moisture conditions, thereby influencing hydrological processes, as well as how snow changes directly impact runoff. Additionally, the manuscript does not adequately address how snow affects frozen ground processes and thereby indirectly impacts hydrology. Moreover, compared to the analysis of the differences in runoff simulations using various modules, I believe it would be more meaningful to explore how the synergistic changes in snow and frozen ground under climate change influence runoff in the study area during the past decades.

**Response 7:** We sincerely thank the reviewer for the valuable suggestions. We fully acknowledge your recommendations and will expand the "Results" section in the revised manuscript to provide a more comprehensive description of the characteristics of snow and frozen ground changes and their impacts on hydrological processes. In the current study, we primarily focused on the influence of frozen ground on soil moisture conditions, and this part will be further supplemented in the revised manuscript. Additionally, we will deepen the discussion in the "Results" section to explore how snow changes directly affect runoff and how snow indirectly influences hydrology through its impact on frozen ground processes.

Regarding your suggestion to investigate the impact of synergistic changes in snow and frozen ground under climate change on runoff over the past decades, we fully recognize the importance of this research direction. However, the main objective of this study is not only to analyze differences in runoff simulation using different modules but also to develop and evaluate an enhanced hydrological model suitable for cold regions. This model integrates snowmelt and freeze-thaw processes, features modular design, is computationally simple, and has wide applicability. Furthermore, we quantitatively analyzed the contribution of snowmelt to runoff generation and the impact of frozen ground on soil conditions, runoff components, and evapotranspiration. The primary focus of this study is to verify the applicability and reliability of the developed model, providing a practical tool for hydrological and climatic assessments and predictions in cold mountainous regions. By quantifying the roles of snow and frozen ground in hydrological processes, our study also contributes to a deeper understanding of the complex hydrological processes in cold regions. It is not, however, aimed at systematically analyzing the

long-term impact of climate change on runoff. We will further clarify this research objective in the revised manuscript.

Due to the lack of observed runoff data in the study area, our current analysis is limited to hydrological simulations over a 19-year period from 2000 to 2018. More extensive observational data would be required for analyzing runoff changes on a longer temporal scale.

Once again, we sincerely thank you for your valuable suggestions. We will further clarify and expand the relevant discussions in the revised manuscript to better address your feedback.

**Comments 8:** In the 'Discussion' section, the authors should focus on how their findings represent an advancement over previous research and then call back to the research questions outlined in the Introduction. Rather than including an extensive literature review, the discussion should emphasize the novel contributions of this study and its implications for the field.

**Response 8:** Thank you for the valuable suggestion. In the revised manuscript, we will restructure the "Discussion" section to better align with the research questions outlined in the Introduction and to emphasize the novel contributions of this study. Specifically, we will:

- Highlight the key findings of this research and explain how they advance the field beyond previous studies. For instance, we will focus on the development and application of the GXAJ-S-SF model, which integrates snowmelt and freeze-thaw processes to improve hydrological simulations in cold regions, and discuss its significance.
- Streamline the discussion by reducing the emphasis on literature review and instead focus on the unique contributions of this study, such as quantifying the impacts of snow and seasonally frozen ground on runoff, evapotranspiration, and soil moisture dynamics.
- Clearly articulate the broader implications of our findings for hydrological modeling, water resource management, and climate change impact assessments, particularly for cold mountainous regions.

By incorporating these adjustments, we aim to provide a more focused and impactful narrative that underscores the significance of our findings and their relevance to the field.

---

## Author Response (AR1)

**Response to RC1:**

**General comments**

**Comments 1:** This manuscript is well written, and the work done appears quite meticulous and informative from a methodological point of view, but I am not fully convinced of the novelty of this manuscript. The manuscript shows that GXAJ-S-SF outperforms GXAJ. It is self-evidently almost certain that a more accurate hydrological partitioning can be achieved when two important physical processes snowmelt and freeze-thaw are included into the model. Thus, it is certainly expected that GXAJ-S-SF will outperform GXAJ in a region that experiences the S and SF processes. Furthermore, since GXAJ consistently underestimates the runoff, and since the physical processes modeled in SF can only increase the runoff but not decrease it, it is a foregone conclusion that upon calibrating SF you will arrive at a better fit for GXAJ-S-SF than GXAJ. As far as I can tell, there are no novel or interesting findings regarding hydrological processes in this manuscript, nor are there meaningful analyses about the utility and information content of the hydrological models used beyond the goodness-of-fit metrics NSE, RBE, RMSE. Therefore, I recommend that after major revisions addressing the concerns I have elaborated below, this manuscript could be suitable for publication as a technical note.

**Response 1:** Thank you for your detailed evaluation of our manuscript and your constructive feedback. We highly appreciate your comments on the novelty and contributions of the study, which have provided us with valuable guidance for improving the manuscript. Below, we address your concerns and clarify the innovative aspects and unique findings of our research.

We acknowledge your point that the inclusion of snowmelt and freeze-thaw processes (SFG) in the GXAJ-S-SF model makes performance improvements over the original GXAJ model somewhat expected in regions experiencing these processes. Indeed, prior studies (e.g., Ala-Aho et al., 2021) have shown the significant impact of SFG on runoff at small scales. However, as emphasized in the revised introduction and discussion sections (see lines 50-70, 135-155), there are still substantial knowledge gaps regarding the complex impacts of SFG processes on runoff in large basins, especially concerning the level of model complexity required to accurately simulate these processes. Our study systematically addresses these gaps, making the following key contributions:

- **Innovation in snow-freeze-thaw coupling:**

We developed a physically meaningful method to couple snowmelt and freeze-thaw processes within the hydrological model. This coupling allows for a quantitative analysis of how snowmelt and frozen ground influence runoff, soil moisture dynamics, and evapotranspiration. Importantly, the coupling approach is computationally efficient, requiring relatively few additional parameters while retaining strong physical interpretability.

- **Assessment of dominant hydrological processes in a large basin subject to SFG:**

Unlike previous studies that often focus on small-scale applications, our study systematically compares simplified models (e.g., without snow-SFG extensions or considering snow only) with extended models that account for both snow and SFG processes. These comparisons reveal that SFG processes significantly alter runoff dynamics at large basin scales, with quantitative results showing an increase in surface runoff by 39–77% and significant reductions in interflow and groundwater runoff compared to models during cold months that neglect SFG processes. This provides critical insights into the role of SFG in shaping hydrological regimes and highlights the necessity of incorporating such processes in predictive models for cold regions.

- **Integration of process understanding with practical applications:**

By analyzing the effects of varying frozen soil depths and their spatiotemporal distributions on hydrological processes, our study provides a comprehensive understanding of the complex feedback mechanisms of frozen ground on hydrological systems. These insights not only enhance the understanding of snow-SFG-hydrology interactions but also offer practical guidance for predicting hydrological responses to future climate change in cold mountainous regions.

In summary, while the improved performance of GXAJ-S-SF may appear intuitive, our study goes beyond performance metrics by systematically addressing the scientific question of how and to what extent SFG processes impact hydrological dynamics in large basins, an area with significant gaps in current knowledge. We have revised the manuscript to emphasize these contributions more explicitly, including updates to the title, introduction, discussion, and conclusion sections.

**Comments 2:** Would it not be more meaningful to compare GXAJ-S-SF to a different hydrological model that also includes snowmelt and seasonal freeze-thaw? For example one of those models you mentioned in L96 – 112. Even if an actual model comparison is not done, it would be useful to discuss the differences and similarities in model processes between GXAJ-S-SF and other similar models with snowmelt and freeze-thaw functions.

**Response 2:** Thank you very much for your valuable suggestions regarding our research! We have carefully followed your advice and expanded the discussion section, particularly in Section 4.1 "*Key limitations in hydrological models in relation to their process complexit*y". In this section, we have compared the processes of the current GXAJ-S-SF model with other hydrological models that include snowmelt and freeze-thaw functions.

Specifically, we elaborated on the similarities and differences in how these models simulate hydrological processes, highlighting the unique features and advantages of the GXAJ-S-SF model. These comparisons have also helped us to clarify the innovations and improvements that the GXAJ-S-SF model offers in representing hydrological processes.

Additionally, we discussed how the process complexity and computational efficiency of the GXAJ-S-SF model compare to other models. This provides further context for understanding the trade-offs between model simplicity and accuracy, as well as the potential for applying our model to other cold-region catchments. We believe this discussion addresses your concerns and adds value to the manuscript by offering a more comprehensive perspective.

We sincerely appreciate your constructive feedback, which has helped us improve the quality of our manuscript.

**Comments 3:** It would be more rigorous to re-run the models with different priors. For example, there could be a configuration of GXAJ, with soil property related parameters set at an "annual average effective value" taking into account that the soil is frozen for 9 months of the year. This hypothetical configuration of GXAJ could possibly produce results as good as GXAJ-S-SF, but it is possible that this configuration of GXAJ was not tested because the optimization algorithm was stuck in a local minimum. Given the highly nonlinear processes involved in this model, I think that calibrating from a single set of priors may be insufficient.

**Response 3:** Thank you for your valuable comments on our research. We greatly appreciate your feedback and have addressed the relevant issues with detailed revisions and additions as follows:

**On Optimization Algorithm and Parameter Settings:**

Regarding your concerns about model optimization and the use of "prior parameters," we understand the potential limitations of calibrating the model using a single prior parameter configuration. In our study, we employed the SCE-UA optimization algorithm to precisely calibrate key parameters and obtain the optimal solution for the model. To avoid the risk of local optima, we set a range of prior parameters and randomly selected different configurations within the allowed range, running the optimization algorithm multiple times. While this approach was not explicitly described in the original manuscript, we have now supplemented the relevant content to clarify that we considered different prior parameters during optimization (Section 2.2.3, lines 404-414). Through this approach, we enhanced the model's stability and minimized reliance on a single configuration, addressing the limitations you raised.

**On Soil Characteristics and Related Parameter Settings:**

We fully agree with your observation that the presence of frozen ground significantly alters soil moisture dynamics, which, in turn, affects the storage capacities of soil tension water and free water and their spatial-temporal distribution. In our study, we did not rely on a simplistic "annual effective value" to account for frozen ground effects. Instead, we dynamically adjusted the distribution of these underlying surface parameters by comparing the depth of frozen ground (characterized by spatial-temporal heterogeneity) with the corresponding soil layer thickness (as categorized in four specific cases in the methodology). This adjustment supported the model in reflecting key effects of frozen ground. Through this approach, we fully considered the spatial-temporal heterogeneity induced by frozen ground, thereby improving the accuracy of the simulation results. In the revised manuscript, we have further elaborated on this section (see Section 2.2.2 *Freeze-thaw proces*s), detailing how changes in frozen ground depth dynamically influence soil layer thickness and related parameters, enhancing the model's capacity to simulate frozen ground dynamics.

We believe that these revisions will better demonstrate the scientific validity and rationality of our approach, and we thank the reviewer for their insightful suggestions.

**Comments 4:** L409 – 417: "The accuracy in simulating the initial freeze and initial thaw dates was validated against ground temperature data from meteorological stations within the basin (Fig. S5), indirectly confirming the simulated soil freeze-thaw processes."

Could you provide citations or a more detailed discussion to support the validity of this point? Since freezing and melting both start from the top, and since the temperature data for verification was measured at the ground surface, simulating the correct initial freeze and initial thaw dates does not help confirm that the model has simulated the freezing depth correctly over the 9 months with frozen soils.

**Response 4:** Thank you for your insightful comments. We have identified several studies supporting the use of surface temperature to validate the initial freeze and thaw dates, which provide a theoretical basis for applying this method in regions lacking observational data on frozen soil depth (Li et al., 2022). However, due to the absence of measured frozen soil depth data in our study area, we were unable to directly validate the simulated frozen soil depth. Consequently, this study primarily utilized available surface temperature data to verify the initial freeze and thaw dates, thereby indirectly supporting the reliability of the simulated freeze-thaw processes.

To further evaluate the model's reliability, we compared the spatial distribution of the maximum frozen soil depth simulated in this study with data from the Tibetan Plateau Permafrost Dataset (1961–2020) for the 2000s. The comparison revealed a high degree of consistency in both spatial distribution patterns and magnitude, with a correlation coefficient of 0.89. These results provided additional validation support, and we presented these comparative analyses in the revised manuscript (Section 3.1 "*Simulation of snow accumulation and freeze-thaw process*," lines 475-486).

We also recognize that the use of surface temperature and maximum frozen ground depth to verify the freeze-thaw process introduces some uncertainty (Li et al., 2022), Since the GXAJ-S and GXAJ-S-SF model variants used the same temperature, snow and frozen ground data in the present simulations, they can be expected to share similar data errors, However, due the non-linear nature of the modeled processes, such data errors may still not cancel completely when comparing different models. Nevertheless, observed differences in model performance between these models are mainly expected to reflect differences in model capabilities rather than differences in input datasets. Future work should focus on improving remote sensing data quality and exploring the long-term robustness of the model to further enhance performance and improve our understanding of the freeze-thaw processes in complex mountainous cold regions (Section 4.1 *"Key limitations in hydrological models in relation to their process complexity"* lines 642-659).

**Comments 5:** I think that the "modular approach" that you emphasize several times, including in the abstract and conclusion, is reinventing the wheel as it is just another name for loose coupling or one-way coupling, which is a basic hydrological concept.

**Response 5:** Thank you for your valuable comments and for providing us with the opportunity to clarify the term "modular approach." After carefully considering your feedback, we agreed that the term "modular" may overlap to some extent with concepts such as loosely coupled or unidirectional coupling, especially in the context of hydrological modeling. However, our use of "modular" aimed to emphasize the flexibility, scalability, and reusability of the model components. Specifically, the snow and frozen ground modules in our study were designed as independent components that could be enabled or disabled depending on environmental conditions (e.g., the presence of snow or seasonally frozen ground). These components were not hard-coded into the GXAJ model; instead, they were designed to be integrated into other hydrological frameworks without requiring significant modifications to the core structure of the model. This design approach enhanced the model's flexibility and adaptability, allowing researchers to extend or modify it to suit different cold-region environments.

To address your comments, we revised the manuscript to avoid potential misunderstandings caused by the term "modular approach." Instead, we described the design philosophy of the model components more accurately. In the abstract and conclusion, we replaced the term "modular approach" with "flexible and adaptable," ensuring that the core idea of the design is conveyed clearly and without ambiguity.

We hope this clarification and the corresponding revisions have adequately addressed your concerns.

**Comments 6:** After reading through the manuscript several times, I recognize that the bulk of the scientific contribution of this manuscript lies in the freeze-thaw process module in section 2.2.2. As shown in the results, it fits well with the measurements. However, I think some parts should be explained more clearly. What is the purpose of using two different representations of the soil layers in one model? Why not use the same layers for the computation of runoff, moisture and ET (Figure S1)? Does this mean that in the simulations, the humus layer could sometimes overlap with both the "upper soil" and part of the "lower soil"? And can the "upper soil" sometimes overlap with both the humus layer and the vadose zone? Does this not then imply that you need to interpolate some effective soil parameter values that may be inappropriate for the actual individual soil layers? How would this affect the runoff and discharge predictions? Furthermore, wouldn't this mean that the parameter values you calibrate from field data do not have a proper physical meaning? I think that in order to reconcile the two different representations of the soil layers, it is inevitable that the calibrated parameter values are smoothed interpolations of the values that would actually describe each individual soil layer.

**Response 6:** Thank you for your positive feedback on the freeze-thaw process in section 2.2.2 and for your valuable suggestions. Regarding the use of two different soil layer representations in the model, we adopted this approach based on the design philosophy of the original GXAJ model. We did not provide sufficient explanation in the original manuscript, which caused some confusion, and we have now clarified this in the revised version (Section 2.2 *"Modeling approach"* lines 200-256, 327-379).

Specifically: When calculating runoff for a grid cell, soil saturation refers to the soil water content reaching the field capacity, not the saturation water content. The GXAJ model uses a saturation runoff mechanism, meaning that runoff only occurs when the soil's unsaturated zone reaches field capacity. Before this point, all incoming water is absorbed by the soil without generating runoff. In the GXAJ model, the tension water storage capacity ($W_M$, in mm) of a grid cell is determined by the watershed's topography, as well as soil, vegetation, and other surface conditions. We do not consider the uneven distribution of tension water content within the grid cell. To calculate the actual precipitation ($P_e$) available for runoff, we subtract the evaporation, canopy interception, and river precipitation from the measured rainfall during the calculation period, then check if upstream inflow replenishes the soil water content of the current grid cell.

When calculating the sources of runoff (Figure S2(a)), the runoff in the grid cell is divided into three components: surface runoff, interflow, and groundwater runoff. The GXAJ model treats the upper soil layer in the unsaturated zone as the humus layer (determined by topography, soil, vegetation, and other surface conditions), with the bottom of the humus layer considered a "relatively impermeable layer." Some of the runoff generates interflow, while part continues to percolate, generating groundwater runoff. When the free water in the humus layer reaches saturation, surface runoff is produced. Similarly, we do not consider the uneven distribution of free water storage in the grid cell.

In summary, the GXAJ model (Yao et al., 2012) calculates the tension water storage capacity ($W_M$) in the unsaturated zone (Figure S1) and the free water storage capacity ($S_M$) in the humus layer to divide runoff into three components: $R_s$, $R_i$, and $R_g$. $W_M$ determines whether runoff occurs and the amount of runoff (saturation excess runoff), while the free water content in the surface soil splits runoff into $R_i$ and $R_g$. When the free water content reaches saturation, $R_s$ is produced, as shown in Figure S2(a).

For evapotranspiration (Figure S2(b)), the GXAJ model uses a three-layer evapotranspiration model, dividing the soil (vadose zone) into upper, middle, and lower layers, with corresponding tension water storage capacities: $W_{UM}, W_{LM}$ and $W_{DM}$ (in mm). During actual evapotranspiration calculation, canopy interception is evaporated based on evapotranspiration capacity. When the intercepted water is less than the evapotranspiration capacity, the three-layer model is applied. The calculation principle is that the upper layer evaporates according to its evapotranspiration capacity. If the upper layer cannot supply enough water for evapotranspiration, the remaining capacity is drawn from the middle layer, with evapotranspiration in the middle layer proportional to the remaining capacity and inversely proportional to the middle layer's storage capacity. The ratio of middle layer evapotranspiration to the remaining capacity cannot be less than the deep layer evapotranspiration coefficient, $C$. If the middle layer cannot supply enough, the deep layer water will supply the deficit. The corresponding soil moisture and evapotranspiration are labeled as $W^u$, $W^l$, and $W^d$, and $E^u$, $E^l$, and $E^d$.

The original GXAJ model used different soil layers to better simulate the role of soil layers at different depths in hydrological processes. This means that the humus layer may overlap with the "upper soil" and part of the "lower soil," as you understand, but the specific situation may vary depending on the surface conditions. However, the soil surface parameters, such as tension water storage capacity and free water storage capacity, are derived from the physical properties of the soil (e.g., soil type and structure), as well as topography and vegetation, and thus have physical significance. Applying these concepts to a single-layer soil system would simplify the calculation, treating the entire soil layer as the unsaturated zone for runoff calculation and using a single-layer evapotranspiration model.

I hope this explanation resolves your doubts, and we will further improve the manuscript in the revision.

**Specific comments**

**Comments 7:** L192: Is saturation excess runoff a reliable way to partition snowmelt fluxes, which are fast and may often exceed the infiltration capacity?

**Response 7:** Thank you for the valuable feedback. The melting rate of snowmelt water is usually fast, and in the presence of a frozen soil surface, the soil's permeability is limited, which easily leads to surface runoff. We fully agree with this point, and it has been thoroughly considered in Sections 2.1 and 2.2.1 of our study. In our model, we specifically focused on the interaction between water movement and soil during the snowmelt process, considering the potential freezing of the soil surface. When the snowmelt water encounters the frozen soil layer, due to the low permeability of frozen soil, the snowmelt water cannot rapidly infiltrate into the soil, resulting in significant surface runoff. The detailed consideration of this process in the model ensures the rapid generation of runoff from snowmelt water. Additionally, even without considering the effects of frozen soil, the runoff generation mechanism in our study is based on saturation excess and infiltration excess (Figure 2), which also accounts for the possibility of infiltration excess runoff from snowmelt fluxes.

**Comments 8:** L277: If you divide the SFD by the cube root of ASD to get SFD*, then the units of SFD* are $[cm]^{2/3}$. What does that physically mean?

**Response 8:** Thank you for your attention. The empirical formula for frozen soil depth used in our study is derived from the research "*Influence of snow cover on soil freeze depth across China*" which utilized observational data from 378 meteorological stations across China (1980–2014). This study quantified the relationship between snow cover and the maximum seasonal freeze depth (MSFD), as well as the contribution of snow cover to MSFD. The results indicated that in areas with thin snow cover or short snow duration, the impact on freeze depth is minimal. However, in regions with thick snow and longer snow duration, snow cover significantly reduces the frozen soil depth. This relationship can be reasonably reflected by the formula dividing the freeze depth (SFD) by the cube root of snow depth (ASD).

From a physical perspective, the units of SFD* ($[cm]^{2/3}$) may not have strict physical meaning. However, the inverse relationship between SFD and the cube root of ASD reflects the empirical nature of this relationship, which has been validated with high accuracy across multiple stations. This makes it a practical and reliable method for describing the influence of snow cover on frozen soil depth, despite the lack of a physical explanation for the units.

**Comments 9:** L335: Please be consistent with terminology, do not interchangeably use primary parameters and major parameters.

**Response 9:** Thank you for the reviewer's correction. To avoid confusion between "primary parameters" and "major parameters," we have standardized the terminology to "major parameters" throughout the revised manuscript to ensure consistency in expression.

**Comments 10:** L346 – 349: It would be helpful to mark in Figure 2 which processes in SNOW17 were calibrated with measured data, and which were not.

**Response 10:** Thank you for your suggestion. In Figure 2, the SNOW17 model utilized remote sensing snow depth data to calibrate the simulated snow data. We have updated Figure 2 in the revised manuscript to clearly indicate the scope and method of calibration within the model.

**Comments 11:** L349 – 352: I am not sure what this actually means. You definitely need more parameters for GXAJ-S than GXAJ, because you are adding physical processes. Are you saying that just because the -S module is compartmentalized in a module that means that you do not add more parameters to GXAJ? I think that this is a confusing way to describe one-way coupling.

**Response 11:** Thank you for your valuable comments. We appreciate the opportunity to clarify this point. In our study, the goal was to improve model performance by introducing new physical processes while minimizing unnecessary adjustable parameters. While incorporating snowmelt (-S) and freeze-thaw (SF) processes required additional parameters to describe these physical processes, we ensured that no extra adjustable parameters were added to the model by adopting a fixed-parameter strategy during model construction. Below, we address your comments in more detail:

• Snowmelt process parameters: To enhance the effectiveness of the model improvement and avoid the possibility that the introduction of additional parameters could potentially improve simulation results, the SNOW17 model was initially run independently. Remote sensing snow depth data (considered as "measured values") were used as input, and the parameters were adjusted to align the model-simulated snow depth with the "measured values," thereby determining the snow parameters for the study area. This approach allowed the integration of the SNOW17 model with the GXAJ model to form the GXAJ-S model for calculating snowmelt runoff in grid cells, ensuring that no new parameters were added to the GXAJ-S model compared to the GXAJ model (Section 2.2.3 "*Model parameters and calibration*" lines 394-402).

- Freeze-thaw process parameters: We utilized empirical parameters, which were also fixed in the model and not treated as additional adjustable parameters during the coupling process. By doing so, the freeze-thaw process was incorporated without introducing additional degrees of freedom (Section 2.2.3 "*Model parameters and calibration*" lines 402-407, Section 2.2.2).

By adopting this fixed-parameter approach, we ensured that improvements in model performance were achieved solely through the introduction of new physical processes, rather than by increasing model complexity with additional degrees of freedom. This clarification has been added to the revised manuscript to ensure transparency and a clearer understanding of the model parameterization strategy.

Once again, we thank the reviewer for highlighting this important aspect, which has helped us refine the clarity of our work.

**Comments 12:** L402 – 404: I think that the evidence of robustness is that the model *did not perform worse* during the validation period. Performing *better* during the validation period is not evidence of robustness. Conversely, performing *better* during the validation period suggests that you made some assumptions about the physical processes hard coded into the model, that were more valid during the validation period. Please discuss this in more detail if possible.

**Response 12:** Thank you for highlighting this important issue. We agree with the reviewer that better model performance during the validation period is not conclusive evidence of robustness. Instead, it may indicate that some assumptions about the physical processes hard-coded into the model align more closely with the conditions during the validation period.

We revisited the conceptualization of the snowmelt and freeze-thaw processes in the SNOW17

model and hypothesize that the improved performance during the validation period could be attributed to simpler hydrological conditions. Specifically, snow depths were generally lower during the validation period compared to the calibration period. Shallower snow conditions may reduce the complexity of snowmelt and freeze-thaw interactions, which are more prone to introducing uncertainties under deeper snow conditions. Consequently, the model was better able to capture these relatively simpler hydrological processes, resulting in improved performance during the validation period.

That said, we acknowledge that other factors might have contributed to the observed performance differences. For example, changes in climate conditions, precipitation patterns, or temperature distributions between the calibration and validation periods could also have influenced model behavior. While these possibilities cannot be entirely ruled out, we believe the model demonstrated reasonable stability during both periods, adapting to varying snow conditions and reliably simulating snowmelt and runoff processes.

In the revised manuscript, we have included a detailed discussion to explain the potential link between improved performance during the validation period and the simpler hydrological conditions (Section 3.1 "*Simulation of snow accumulation and freeze-thaw process*" lines 454-470). Future studies will aim to test the model under a wider range of conditions and across different regions to further evaluate its robustness and identify potential limitations.

We appreciate the reviewer's valuable insights and hope this explanation addresses the concern. If there are additional suggestions or areas you would like us to explore, we are happy to incorporate them.

**Comments 13:** Figure 4: What are the dashed lines?

**Response 13:** Thank you for your question. The dashed lines in Figure 4 represent the trend of snow depth variations (both model-simulated and remote sensing data) over the period from 2000

to 2018. We have revised the figure legend in the manuscript to provide a more detailed explanation to avoid confusion and ensure that the information presented is clearer and more intuitive for readers.

**Comments 14:** Figure 5: What is the dashed line?

**Response 14:** Thank you for your question. The dashed lines in Figure 4 represent the trend of snow depth variations (both model-simulated and remote sensing data) from 2000 to 2018. We have updated the manuscript to provide a more detailed explanation, ensuring clarity and avoiding any potential confusion for readers.

**Comments 15:** L427: You earlier defined an RBE, but not an RE.

**Response 15:** Thank you for your valuable feedback. We acknowledge that the definition of RE (Relative Error) in the original manuscript was incorrect, as it should have been abbreviated as RE

instead of RBE. In the revised version, we have corrected its definition and description to ensure consistency in terminology and to prevent any confusion for readers.

**Comments 16:** L539 – 534: I think that the formation of a saturated layer above ground under these circumstances is possible only for very coarse soils that are inefficient at soil water redistribution. This is unlikely to be a general behavior. If you are referring to a specific soil type, please describe it. If you are claiming this as a general behavior, please provide references.

**Response 16:** Thank you for your valuable suggestion. The formation of a saturated layer above the frozen ground under freeze-thaw conditions is a well-documented phenomenon. This occurs primarily due to the low permeability of frozen soil, which prevents infiltration and causes meltwater or precipitation to accumulate at the base of the thawed layer (i.e., the top of the frozen soil layer). This effect is particularly pronounced when the thawed layer is thin, as water can easily accumulate at the freeze-thaw interface, leading to the formation of a saturated layer.

Numerous studies have reported this phenomenon, particularly highlighting how water accumulation at the freeze-thaw interface can result in a saturated layer above the frozen soil. We have included relevant references in the revised manuscript to further support this argument (Section 4.2 "*The impact of seasonal frozen ground/snow*" lines 693-700).

**Comments 17:** L540: If matric potential is the primary driver of moisture movement, then how does gravity cause a saturated layer to emerge at the frozen interface?

**Response 17:** Thank you for your thoughtful comment. We have carefully considered your suggestion and revisited the relevant statement. We realized that the main idea of this section is to emphasize the inhibitory effect of frozen soil on soil evapotranspiration, and the driving factors of soil water movement (such as matric potential) are not directly related to this mechanism. Therefore, to avoid confusion, we have removed the statement regarding "matric potential being the primary driver of moisture movement." We hope this revision addresses your concern and makes the explanation clearer (Section 4.2 "*The impact of seasonal frozen ground/snow*" lines

693-710).

**Comments 18:** L548 – 549: Which processes are you referring to, and what impacts? Are the processes you study not already naturally part of the local hydrological cycle and ecosystem?

**Response 18:** Thank you for your valuable feedback. The "processes" mentioned in the text, including freeze-thaw dynamics, soil water movement, and the effects of snow and seasonally frozen ground (SFG) on evapotranspiration, are indeed integral parts of the local hydrological cycle and ecosystem. However, we would like to emphasize that this study specifically focuses on how these processes-especially the freeze-thaw cycle and its effects on soil moisture and evaporation rates-can vary under different environmental conditions.

For instance, during the freezing period, frozen ground and snow cover suppress evapotranspiration. In contrast, during the thawing period, the formation of a saturated layer above the frozen soil may alter soil permeability and runoff patterns. These changes are particularly important in the context of climate change, as variations in the freeze and thaw periods could impact water resources and the stability of ecosystems over the long term.

In the revised manuscript, we have made these points more explicitly and further explained the potential impact of these processes (Section 4.2 "*The impact of seasonal frozen ground/snow*" lines 693-710).

**Comments 19:** L580 – L590: I feel that this is self-evident. It is a rehash of the widely known problem that data-calibrated hydrological models are often 'right for the wrong reasons'. It is a nice discussion that fits the work done, but does not contribute new knowledge.

**Response 19:** Thank you for your valuable feedback. We acknowledge your concern that this discussion might reiterate a well-known issue in hydrological modeling. To address this, we have revised and restructured the Discussion section (see revised manuscript) to better highlight the novel contributions of our study.

Specifically, we have refined this section ("*4.1 Key limitations in hydrological models in relation to their process complexity*") to emphasize the limitations of data-calibrated hydrological models and clarify how our approach addresses these challenges:

"*A limitation in the application of the GXAJ base model, which neglects impacts of snow and ice, is related to the fact that the parameters of its modules are determined based on historical basin characteristics. Although such models without frozen ground components can, through appropriate calibration or optimization of parameters, in some cases successfully reproduce historical hydrological processes in cold regions under stable conditions (Li et al., 2011; Zhang et al., 2017), they may not be suitable for evaluating the consequences of future changes as their*

*calibrated values do not represent new conditions of the basin, and as the model lacks physical representation of key drivers of change. Our study demonstrates that incorporating the effects of seasonally frozen ground (SFG), snow, and other environmental factors into a basic model can provide robust and physically consistent results in simulating large-scale hydrological processes in cold regions, which can be particularly important for predicting hydrological impacts of future climate change scenarios. Key limitations in previous studies incorporating snow and seasonally frozen ground processes into hydrological models include taking 0°C as an assumed critical temperature for phase change, while neglecting the energy flux exchange between snow and soil layers, as for instance done in the VIC model (Liang et al., 1994). This potentially compromises simulation accuracy, particularly in regions where snow-frozen ground interactions are significant, such as areas with large seasonal variations in frozen ground depth. Similarly, while the SWAT model adjusts its parameters to accommodate permafrost conditions and introduces a new soil temperature module (Fabre et al., 2017), it does not account for the impact of snow depth on frozen ground and struggles to fully capture the complex and dynamic interactions between frozen ground and soil hydrological processes. Models such as the WEB-DHM (Qi et al., 2019), which employs enthalpy-based snow and frozen ground coupling, and the ATS model (Jafarov et al., 2018), have made substantial progress in simulating multi-year snow and frozen ground dynamics. However, their high complexity and demanding data requirements limit their applicability in large-scale hydrological simulations.*

*In this context, while the here developed GXAJ-S-SF model does not rely on above-mentioned limitations of the VIC and SWAT models (as it accounts for the impact of snow depth variations on frozen ground and also captures multidimensional effects of snow-frozen ground coupling on hydrological processes) it still contains energy-related simplifications that makes it less dependent on extensive input data than e.g. the ATS model. In particular, the original lumped SNOW17 model (incorporated as a module in the GXAJ-S and GXAJ-S-SF model) was decentralized, and the energy exchange at the snow-atmosphere-soil interface was accurately simulated based on empirical relationships (He et al., 2011).*"

**Comments 20:** L600 – 602: This argument is valid only if the modeled processes are linear. The processes you have modeled are potentially too nonlinear and have too many interactions for this argument to hold.

**Response 20:** Thank you for your insightful comment. We acknowledge that the argument regarding error propagation is more straightforward in linear systems and may not fully hold in highly nonlinear hydrological processes. To address this concern, we have revised the discussion (see Section 4.1, "Key limitations in hydrological models in relation to their process complexity", L642–L659) to clarify the potential limitations of our approach, specifically as follows:

"*In complex mountainous cold regions, observation remains a bottleneck (Gao et al., 2022). Due to limitations in measured data on frozen soil and snow depth in the considered Yalong River basin, satellite-based snow depth data and ground temperature station data were used in the present study for calibration and verification. In particular, errors in remote sensing snow depth data (Yan et al., 2022; Zou et al., 2014) can propagate to the model output. However, previous studies have specifically investigated the here used remote sensing dataset for the Yalong River basin showing that its accuracy is high (Wu et al., 2024), which suggests that model errors should be relatively low. We also recognize that the use of surface temperature and maximum frozen ground depth to verify the freeze-thaw process introduces some uncertainty (Li et al., 2022), Since the GXAJ-S and GXAJ-S-SF model variants used the same temperature, snow and frozen ground data in the present simulations, they can be expected to share similar data errors, However, due the non-linear nature of the modeled processes, such data errors may still not cancel completely when comparing different models. Nevertheless, observed differences in model performance between these models are mainly expected to reflect differences in model capabilities rather than differences in input datasets. Future work should focus on improving remote sensing data quality and exploring the long-term robustness of the model to further enhance performance and improve our understanding of the freeze-thaw processes in complex mountainous cold regions.*"

We appreciate your valuable feedback, which has helped us refine our discussion and ensure a more rigorous interpretation of model uncertainties.

**Comments 21:** L603 – 604: I agree that remote sensing errors would probably not affected the core conclusions of this manuscript, but not for the reasons you provide in L600 – 602. As I explained in my general comments, I think your conclusions are mostly self-evident.

**Response 21:** Thank you for acknowledging that remote sensing errors are unlikely to affect the core conclusions of this study. Regarding your comment that our conclusions appear mostly self-evident, we would like to clarify that the hydrological impact of seasonally frozen ground (SFG)

at a large basin scale remains insufficiently understood (Gao et al., 2022). The relative importance of SFG processes across different environments and their broader significance in larger basins are still not well understood (Ala-Aho et al., 2021). Our study provides direct evidence that SFG

processes play a crucial role in shaping large-scale runoff dynamics in the Yalong River basin. This conclusion is drawn from the comparative performance of the GXAJ-S-SF model, which explicitly accounts for both snow and SFG effects. Compared to the GXAJ-S and GXAJ models, which do not incorporate SFG, the GXAJ-S-SF model demonstrates significant improvements in simulating different runoff components. Notably, the original GXAJ model tends to slightly overestimate runoff during the calibration period and underestimate it during validation, with the underestimation being more pronounced in spring when snowmelt and frozen ground processes are dominant. The improved GXAJ-S-SF model reveals that increased surface runoff in cold months corresponds with a significant reduction in interflow and groundwater runoff. This insight contributes to a more comprehensive understanding of cold-region hydrological processes.

While incorporating snow and frozen ground effects leads to improvements in statistical performance metrics and runoff simulations-findings that may seem intuitive-our study is not solely focused on optimizing model accuracy. Instead, as highlighted in our response to Comment 1, our primary goal is to enhance the physical realism of hydrological modeling and deepen our understanding of cold-region hydrological processes. The consistency between our results and expected physical behavior reinforces the credibility of our approach and underscores the broader significance of SFG in shaping hydrological regimes.

We appreciate your feedback, which has helped us refine our discussion and emphasize the broader implications of our study.

**Comments 22:** L606 – 615: The benchmark model GXAJ you refer to is not a different model, but it is just GXAJ-S-SF without the snow and freezing capacities. This discussion is not meaningful because it is self-evident.

**Response 22:** Thank you for your comment. We acknowledge that the GXAJ model serves as the baseline version of GXAJ-S-SF, differing only in its exclusion of snow and frozen ground processes. However, we would like to emphasize that the significance of our comparison lies not in merely adding more processes but in demonstrating how these physically meaningful processes influence hydrological responses(see Section 4.1, "Key limitations in hydrological models in relation to their process complexity", L660–L668).

"*Hydrological modeling typically prioritizes model fitness, which in theory can be improved by introducing more fitting parameters. However, this study highlights differences that are due to addition of process-based modules (regarding snow and frozen ground). This implies that improvements in model fit and differences in associated model output (e.g. runoff and evapotranspiration) reflect how the considered snow and/ or frozen ground processes more concretely alter hydrological flows. This therefore increases the understanding of underlying hydrological processes (Gao et al., 2022) in large-scale applications such as the Yalong River basin that additionally has a complex topography with large elevation differences yielding high spatio-temporal heterogeneity in snowmelt and freeze-thaw cycles of soil.*"

Additionally, to better highlight the novelty of our study, we have made substantial revisions throughout the manuscript. If the reviewer has any further questions or suggestions, we would be happy to discuss them.

**Comments 23:** L615 – 617: What is the modular approach being contrasted against? What results did you show that support this statement?

**Response 23:** Thank you for your insightful comment. In response to your suggestion (Comments 5), we have thoroughly revised the manuscript to clarify the concept of the "modular approach" and ensure that its context and contrasts are more explicitly defined. Additionally, we have refined the discussion to better support our statements, incorporating feedback from Comment 22 as well. The revised manuscript now provides a clearer comparison of the modular approach with alternative modeling strategies and explicitly presents the results that support our conclusions. We appreciate your valuable feedback, and we welcome any further discussions to ensure clarity and rigor in our presentation.

**Comments 24:** L630 – 634: This is a great point, and could be expanded to make the discussion more interesting.

**Response 24:** Thank you for your positive feedback and for highlighting this important aspect. We appreciate your suggestion and have expanded this section accordingly (see Section 4.2, "The impact of seasonal frozen ground/snow", L733–L751). In the revised manuscript, we have further elaborated on the broader implications of our findings, particularly regarding the challenges of applying hydrological models in cold regions with complex geological and environmental conditions. We emphasize the necessity of considering local variability when extending model applications, while also recognizing the potential for improving process understanding and generalizability.

*"This study quantitatively analyzed the impact of seasonal snow and frozen ground on hydrological processes based on the hydrological model, and its validity was confirmed not only by measured runoff but also by multi-source data, especially the trends in snow and frozen soil changes. Although our developed model has great application potential in other cold regions, it should be used cautiously without prior understanding of the modeling system. Snow and frozen ground are just part of the factors affecting cold-region hydrology, with other factors intertwined with frozen ground having significant impacts. Geological conditions, in particular, greatly affect frozen ground but have large spatial heterogeneity and are challenging to measure. The empirical parameters of the SNOW17 model and Stefan equation have clear physical significance and have been validated by previous studies (Anderson, 2006; Ran et al., 2022; Zou et al., 2014). However, the soil and geology of mountainous basins are extremely complex and vary significantly across regions. This complexity introduces challenges in applying these models to different watersheds, requiring recalibration of their values. For instance, soil texture, moisture retention, and thermal properties can vary considerably, influencing the depth and dynamics of the seasonal frozen ground. Similarly, variations in topography, vegetation cover, and geological composition can impact runoff, infiltration, and evapotranspiration processes. Expanding the application of complex hydrological models therefore requires careful attention to local and regional variability in ambient conditions, but may also considerably increase the understanding of processes and the generalizability of the assumptions made."*

We sincerely appreciate your insightful comments, which have helped us strengthen the discussion and enhance the clarity of our study.

**Response to RC2:**

**General comments**

**Comments 1:** The authors present a modeling study on the hydrological impacts of snow and frozen ground dynamics in a topographically complex basin. The topic of cryospheric changes and their impacts on hydrology is both significant and timely. However, the authors should address several key issues in the current manuscript to enhance its quality before it can be considered further.

I think the novelty of this study is not sufficiently distinctive or well-highlighted. There have already been numerous modeling studies on snow and frozen ground dynamics in the Tibetan Plateau region, both the basin-scale and regional-scale studies are conducted. Moreover, the models employed in previous studies provided more advanced representations of snow and frozen ground processes, particularly in terms of frozen ground dynamics, compared to the model used in this study. Therefore, the authors need to consider how to better emphasize the unique contributions of this study in comparison to prior research.

**Response 1:** Thank you for your thoughtful evaluation and constructive feedback. We appreciate your comments regarding the novelty of our study, which prompted us to further clarify and emphasize its unique contributions.

In the revised manuscript, we have explicitly highlighted the following key aspects that distinguish our study:

- **Hydrological impacts of snow and frozen ground in large basins:**
While numerous studies have examined the role of snow and frozen ground in hydrological processes, most have focused on small-scale or regional-scale applications. However, at a large basin scale, key questions remain regarding the extent to which seasonally frozen ground (SFG) influences runoff and how model complexity affects hydrological simulations (lines 57-70). Our study systematically evaluates these effects by comparing models of different complexity levels. To emphasize this contribution, we have revised the title, discussion, and conclusion sections accordingly.

- **A simple and data-efficient snow and freeze-thaw coupling method:**
Unlike many existing models that require extensive parameterization of freeze-thaw dynamics, our study integrates snowmelt and freeze-thaw processes in a physically meaningful yet computationally efficient manner. The developed snow and freeze-thaw coupling module requires relatively few additional parameters and has low dependence on input data—an essential advantage for data-scarce cold regions. This approach makes it feasible for large-scale applications, particularly in regions lacking detailed soil freeze-thaw observations (see Section "4.1 Key limitations in hydrological models in relation to their process complexity").

- **Quantitative assessment of the impacts of SFG on hydrological processes in large basins:**

Our study provides a systematic comparison between simplified models (which exclude SFG processes or consider only snow processes) and extended models that incorporate both snow and SFG effects. The results reveal that SFG can significantly alter hydrological components, increasing surface runoff during cold months (by 39%–77% compared to models that ignore SFG) while reducing interflow and groundwater runoff. Compared to previous studies, our approach not only quantifies these effects at a large basin scale but also provides insights into how different model structures influence runoff predictions. These findings have been further emphasized in the revised Abstract, Introduction, Result and Discussion sections.

- **Combining hydrological process understanding with practical applications:**

By analyzing frozen soil depth and its spatiotemporal impact on hydrological processes, this study enhances our understanding of the complex feedback mechanisms between frozen ground and hydrology. These insights not only improve our knowledge of snow, freeze-thaw, and hydrological interactions but also provide valuable references for predicting hydrological changes in cold mountainous regions under future climate change scenarios. Additionally, the model's relatively simple parameterization and efficient data requirements make it suitable for practical applications in data-limited regions, where many existing models may face constraints due to high data demands.

We have incorporated these refinements throughout the manuscript to better highlight the contributions of this study. Once again, we sincerely appreciate your valuable comments, which have helped us improve the clarity and impact of our work.

**Specific comments**

**Comments 2:** In Figure 3c, it is evident that a significant portion of the study area is covered by permafrost. However, the Stefan model mentioned in the methodology is designed to model seasonal frozen ground. Did the authors separately account for the dynamics of permafrost in their study? If not, this could be a critical limitation that needs to be addressed or clarified.

**Response 2:** Thank you for highlighting this important aspect. As shown in Figure 3c, the study area is primarily dominated by seasonal frozen ground (SFG), while permafrost accounts for less than 10% and is sparsely distributed along the edges of the study area. This distribution is consistent with the characteristics of the region, though some of the permafrost extent in the dataset may be subject to uncertainty, particularly in boundary areas.

Since our study focuses on the hydrological impacts of seasonal frozen ground, we employed an improved Stefan model that is specifically designed to simulate SFG processes. This model is well-suited for the dominant frozen ground type in our study area. The simulation results demonstrate high accuracy in these regions, confirming the model's effectiveness in capturing key hydrological processes.

Regarding the limited permafrost areas, we acknowledge that our model does not explicitly account for permafrost dynamics. However, given its sparse distribution and small coverage, the influence of permafrost on basin-scale runoff processes is expected to be minimal. Additionally, in permafrost regions, the active layer undergoes seasonal freeze-thaw cycles, which are conceptually similar to SFG processes. As a result, our approach remains applicable to the majority of the study area, and any potential impact of permafrost on overall simulation results is likely negligible.

**Comments 3:** Line 272-275: How was this threshold 30cm determined? Was a sensitivity analysis conducted to assess the impact of this threshold on the results? Providing such an analysis would help evaluate the robustness of the study's findings.

**Response 3:** Thank you for the valuable question! The 30 cm threshold mentioned in our study is based on findings from previous research. Numerous studies have explored different snow depth thresholds. For example, Brooks et al. (1995, 1999) and Cline (1995) suggested that when snow depth reaches 30–40 cm, air temperature has little influence on ground temperature. Building on this, Hill (2015) developed a conceptual model indicating that thick snow cover (>30 cm) effectively insulates the ground, preventing deep freezing and enabling groundwater recharge. Conversely, for thin snow cover (<30 cm), the ground remains seasonally frozen during the snowmelt period, which delays groundwater recharge and shifts hydrological responses later into the summer.

We adopted Hill's (2015) 30 cm threshold for snow depth based on these well-established findings. Although we did not conduct a detailed sensitivity analysis in this study, previous research has demonstrated that this threshold is widely applicable across different cold regions. Additionally, the simulated hydrological response in our study aligns well with observed trends, further supporting the appropriateness of this threshold.

To address this point more explicitly, we have revised the manuscript to provide a more detailed explanation of the threshold selection and added relevant references. We sincerely appreciate the reviewer's insightful comment!

**Comments 4:** In Table 3, the authors utilized several data products from other studies. However,
the accuracy of these datasets, particularly the snow depth data, which is critical for this study, has not been clarified.

**Response 4:** Thank you for pointing out this important aspect. We acknowledge that the accuracy of the datasets used in our study, particularly the snow depth data, is critical for ensuring reliable hydrological simulations. In the revised manuscript, we have provided additional details on the
accuracy and validation methods of the datasets used, with relevant references to further clarify these aspects.

In particular, we have expanded the discussion on the uncertainty introduced by snow depth and frozen ground datasets and their potential impact on our results (see revised manuscript, Lines 642–659):

"*In complex mountainous cold regions, observation remains a bottleneck (Gao et al., 2022). Due to limitations in measured data on frozen soil and snow depth in the considered Yalong River basin, satellite-based snow depth data and ground temperature station data were used in the present study for calibration and verification. In particular, errors in remote sensing snow depth data (Yan et al., 2022; Zou et al., 2014) can propagate to the model output. However, previous studies have
specifically investigated the here used remote sensing dataset for the Yalong River basin showing that its accuracy is high (Wu et al., 2024), which suggests that model errors should be relatively low. We also recognize that the use of surface temperature and maximum frozen ground depth to*

*verify the freeze-thaw process introduces some uncertainty (Li et al., 2022), Since the GXAJ-S and GXAJ-S-SF model variants used the same temperature, snow and frozen ground data in the present* simulations, they can be expected to share similar data errors, However, due the non-linear nature of the modeled processes, such data errors may still not cancel completely when comparing different models. Nevertheless, observed differences in model performance between these models are mainly expected to reflect differences in model capabilities rather than differences in input datasets. Future work should focus on improving remote sensing data quality and exploring the long-term robustness of the model to further enhance performance and improve our understanding of the freeze-thaw processes in complex mountainous cold regions."

We sincerely appreciate the reviewer's insightful suggestion, which has helped us improve the robustness and clarity of our study.

**Comments 5:** Line 309-404: In points with high snow depth, there are significant discrepancies between the model results and the remote sensing data. The authors should investigate the underlying causes of these differences.

**Response 5:** Thank you for your valuable comment. We acknowledge that discrepancies exist between the model results and remote sensing data in regions with high snow depth. One potential reason is that hydrological processes in areas with deep snow are more complex. The model employs a simplified parameterization approach to simulate snow accumulation and melt processes, which may not fully capture the intricate dynamics of snow accumulation-melt cycles. These factors may also explain why the model performance during the validation period (when snow depth is relatively shallow) is better than during the calibration period (when snow depth is deeper), as shown in Figure 4. However, despite these differences in areas with deep snow, the calibration and validation results still show relatively low RMSE and BIAS values, indicating that the model performs well overall in simulating snow depth dynamics. Furthermore, compared to the original model, the improved model exhibits significant improvements in simulating snowmelt runoff.

We acknowledge that snowmelt is a complex hydrological process, and given the limited data availability, we have made efforts to utilize existing observations and remote sensing data to simulate the snowmelt process with the highest possible accuracy. While certain limitations and uncertainties are inevitable, we believe that such efforts are meaningful and valuable, especially in cold regions where data scarcity poses significant challenges.

In the revised manuscript, we have further investigated and discussed the potential causes of these discrepancies, including the limitations of both remote sensing data and the model itself (see revised manuscript, Lines 454-470, and Section "Discussion"). Additionally, we have expanded the uncertainty analysis section to provide a more comprehensive discussion on this issue and suggest directions for future improvements (Section "4.1 Key limitations in hydrological models in relation to their process complexity",L642-659).

**Comments 6:** Line 412:415: For the simulation of frozen ground processes, verifying only the accuracy of the initial freeze and initial thaw dates is far from sufficient. It is also essential to validate the simulated soil temperature and soil moisture (including liquid water content and soil ice content). These variables are key to understanding how freeze-thaw processes influence basin hydrology. Therefore, the authors should provide validation results for these variables to demonstrate the reliability of the study's findings.

**Response 6:** Thank you for your insightful comment. We acknowledge the importance of validating soil temperature and soil moisture (including liquid water content and soil ice content)

to further demonstrate the reliability of the simulated freeze-thaw processes. However, due to the lack of measured frozen soil depth, soil temperature, and soil moisture data—an issue common in most cold regions-we are currently unable to directly validate these variables.

To indirectly support the reliability of the simulated freeze-thaw processes, we used available surface temperature data to verify the initial freeze and thaw dates, a method that has been supported by multiple studies. In addition, to further assess the model's reliability, we compared the spatial distribution of the maximum frozen soil depth simulated in this study with data from the Tibetan Plateau Permafrost Dataset (1961–2020) for the 2000s. The results showed a high degree of consistency in both spatial distribution patterns and magnitude, with a correlation coefficient of 0.89, providing additional validation support. These comparative analyses have been presented in the revised manuscript (Section 3.1, "*Simulation of snow accumulation and freeze-thaw process*" Lines 471-486).

We also recognize that verifying the freeze-thaw process using surface temperature and maximum frozen ground depth introduces some uncertainties (Li et al., 2022). Since both the GXAJ-S and

GXAJ-S-SF models used the same input datasets for temperature, snow, and frozen ground, they are expected to share similar data uncertainties. However, due to the nonlinear nature of the modeled processes, these errors may not completely cancel when comparing different models. Nevertheless, observed differences in model performance between these models are mainly expected to reflect differences in model capabilities rather than differences in input datasets.

We agree that future work should focus on improving remote sensing data quality and exploring the long-term robustness of the model to further enhance its performance and improve our understanding of freeze-thaw processes in complex mountainous cold regions. We have incorporated this discussion in Section 4.1 ("*Key limitations in hydrological models in relation to their process complexity*" Lines 642-659).

**Comments 7:** The 'Results' section is too brief and lacks depth in describing the characteristics of snow and frozen ground changes and their hydrological effects. For instance, there is insufficient discussion on how frozen ground processes alter soil temperature and moisture conditions, thereby influencing hydrological processes, as well as how snow changes directly impact runoff. Additionally, the manuscript does not adequately address how snow affects frozen ground processes and thereby indirectly impacts hydrology. Moreover, compared to the analysis of the differences in runoff simulations using various modules, I believe it would be more meaningful to explore how the synergistic changes in snow and frozen ground under climate change influence runoff in the study area during the past decades.

**Response 7:** We sincerely appreciate the reviewer's insightful comments and suggestions. We fully agree with your concerns and have significantly expanded the "Results" section to provide a more comprehensive description of snow and frozen ground dynamics and their hydrological effects.

In the revised manuscript, beyond the original content-covering the simulation of snow accumulation, freeze-thaw processes, runoff, and the hydrological impacts of snow and frozen ground (on runoff components and evapotranspiration)-we have made the following additions:

- Seasonal variations in snow and frozen ground to provide a more detailed temporal perspective.

- The influence of snow depth on frozen ground dynamics, highlighting the snowpack's insulating effect on soil freezing and thawing.

• The impact of snow and frozen ground on soil moisture, further illustrating their indirect influence on hydrological processes.

Additionally, we have strengthened the discussion on the contribution of snowmelt to runoff, ensuring that the hydrological impacts of these processes are more thoroughly analyzed. To address these revisions, we have made comprehensive modifications throughout the manuscript, including refinements to the research objectives, methodology, results, and uncertainty discussions, to better emphasize the contributions of this study.

Regarding your suggestion to explore the synergistic effects of snow and frozen ground changes on runoff under climate change over past decades, we fully acknowledge the importance of this research direction. However, the primary goal of this study is not only to analyze runoff differences under different model configurations but also to develop and evaluate an enhanced hydrological model for cold regions. This model integrates snowmelt and freeze-thaw processes through a modular and computationally efficient design, making it widely applicable. Furthermore, we quantitatively assess the contribution of snowmelt to runoff and the effects of frozen ground on soil conditions, runoff components, and evapotranspiration. While our findings contribute to a better understanding of the complex hydrological processes in cold regions, the study does not aim to systematically analyze the long-term impacts of climate change on runoff. We have clarified this research objective in the revised manuscript.

Additionally, due to the lack of long-term observed runoff data in the study area, our current analysis is limited to hydrological simulations from 2000 to 2018. A longer-term investigation would require more extensive observational data, which remains a direction for future research.

Once again, we sincerely appreciate your valuable feedback. We have carefully revised the manuscript to clarify and expand relevant discussions, ensuring a more thorough response to your suggestions.

**Comments 8:** In the 'Discussion' section, the authors should focus on how their findings represent an advancement over previous research and then call back to the research questions outlined in the Introduction. Rather than including an extensive literature review, the discussion should emphasize the novel contributions of this study and its implications for the field.

**Response 8:** We appreciate the reviewer's suggestion to refine the discussion by emphasizing the novel contributions of our study rather than providing an extensive literature review. In the revised manuscript, we have significantly restructured the Discussion section to focus on the advancements our study brings to the field and how these findings address the research questions outlined in the Introduction.

Specifically, we have divided the Discussion into two subsections:

• **Key limitations in hydrological models in relation to their process complexity** – Here, we critically analyze the limitations of existing models (e.g., VIC, SWAT, and WEB-DHM) in representing snow and seasonally frozen ground (SFG) processes and highlight how our newly developed GXAJ-S-SF model overcomes these challenges by improving physical realism while maintaining computational efficiency.

• **The impact of seasonal frozen ground and snow** – This section emphasizes how our model enhances the understanding of key hydrological processes influenced by SFG, including its effects on runoff components, evapotranspiration, and seasonal water storage dynamics. The discussion also connects back to the research questions by demonstrating how incorporating SFG improves hydrological simulations in cold regions and offers insights into climate change impacts on water resources.

By restructuring the Discussion in this manner, we ensure that the focus remains on our study's key contributions and implications, rather than reiterating a broad literature review. We hope this revision better aligns with the reviewer's expectations.

Thank you for this valuable feedback.

---

## Referee Report (RR1)

**General comments**

Most of my concerns from the original submission have been addressed satisfactorily. The current manuscript is good work, and I enjoyed reading it. I recommend that this manuscript be considered for publication after minor revisions to make some small but important improvements that would greatly improve the clarity of the manuscript.

**Your response 11:** You write that "This approach allowed the integration of the SNOW17 model with the GXAJ model to form the GXAJ-S model for calculating snowmelt runoff in grid cells, ensuring that no new parameters were added to the GXAJ-S model compared to the GXAJ model.". Would it not be clearer if you expressed this as "compared to the GXAJ model, no new parameters were added to the GXAJ- component of the GXAJ-S model"? This would imply that all additional complexity in GXAJ-S is compartmentalized in the -S part only, which is my understanding of your work.

**Your response 10:** It is still not clear to me which parameters you calibrated in your study, and which you used a fixed value for. In L384-386, you state that SNOW17 has 4 parameters that must be calibrated, and 6 that do not have to be. However, in Table 2 you list a "prior range" for 9 of these parameters, with the word "prior" implying that you calibrated all 9 of these to yield a posterior value. In Table 1 it appears that all of the listed parameters are calibrated because you use the phrase "prior estimate". Please indicate in Table 1 and Table 2 which parameters you calibrated from the field data, and which parameters you used certain values for without any calibration.

**Model comparison:** I suggest including a table with the computational time (including both the calibration time and actual simulation time) of simulating a comparable scenario for all 3 models GXAJ, GXAJ-S, and GXAJ-S-SF. That would provide readers with a more complete information on which model to choose. The additional physical detail of GXAJ-S-SF may not be necessary in some applications that prioritize fast computation over accuracy.

---

## Referee Report (RR2)

Thank you for the authors' efforts in improving the manuscript. This version of the paper shows improvement compared to the previous one. However, I still believe that the current paper is not yet suitable for publication in HESS. In my opinion, the novelty of the proposed contributions is insufficient, and the reliability of the conclusions remains inadequate.

The authors emphasize the development of a new hydrological model that considers snow and seasonally frozen ground. However, in my view, the approach of simply coupling empirical formula-based modules into a hydrological model is not sufficiently innovative. Extensive research has already been conducted on this issue in the Tibetan Plateau region, as the authors themselves have also mentioned. Moreover, several existing models, such as the VIC model (Cuo et al., 2015), CLM4.5 (Yang et al., 2018), GIPL2.0 (Qin et al., 2017a), WEB-DHM (Song et al., 2020), and GBEHM (Gao et al., 2018), provide more comprehensive descriptions of snow and seasonally frozen ground modules. The authors repeatedly highlight that the model developed in this study requires fewer input data, thereby emphasizing its applicability in data-scarce regions. However, the input data required by this model are essentially the same as those required by the aforementioned physically based models, primarily including topographic data, vegetation data, and meteorological input data. On the contrary, due to the simplified representation of physical processes in the model, more reference data are needed for parameter calibration. Therefore, I do not consider the simplicity of the model's physical description to be an advantage.

Additionally, the authors primarily demonstrate the model's accuracy through the performance of streamflow simulations. However, this is far from sufficient for a study on hydrological processes in high-mountain basins, where multiple processes contribute to the overall dynamics. Given that the focus of the paper is on analyzing the impacts of snow and frozen ground on streamflow, detailed validation of these two critical intermediate processes is essential. However, such validation is currently lacking. For the snow module, the authors only use remotely sensed snow depth data for calibration and validation. However, the accuracy of these data remains uncertain, as it is well known that remote sensing of snow depth in the complex terrain of the

Tibetan Plateau is subject to significant uncertainties. I recommend that the authors use more authoritative MODIS snow cover data to conduct a more comprehensive validation of the snow module results. As for the frozen ground module, the current validation is mainly limited to the start dates of freeze-thaw cycles (the results do not appear to be very satisfactory, and the authors have not provided quantitative metrics). There is a notable lack of validation for key physical variables, such as soil temperature and soil moisture (both ice and liquid water). I suggest that the authors collect in-situ measurements from the study region or validate their results against more authoritative remote sensing or reanalysis soil data to enhance the reliability of their findings.

Finally, numerous studies have already investigated the hydrological effects of snow and frozen ground at large basin scales, ranging from basin-scale (e.g., Cuo et al., 2015; Qin et al, 2017b; Song et al., 2022; Wang et al., 2023a, 2023b) to the entire Tibetan Plateau. Therefore, I recommend that the authors compare some of their conclusions with those of previous studies, rather than simply stating that research in this area is lacking. Furthermore, I find the current conclusions to be insufficiently in-depth. For example, the critical processes of how changes in soil ice and liquid water in frozen ground affect streamflow are not thoroughly discussed.

References

Cuo, L., Zhang, Y., Bohn, T. J., et al. (2015). Frozen soil degradation and its effects on surface hydrology in the northern Tibetan Plateau. Journal of Geophysical Research: Atmospheres, 120(16). .

Gao, B., Yang, D. W., Qin, Y., et al. (2018). Change in frozen soils and its effect on regional hydrology, upper Heihe basin, northeastern Qinghai-Tibetan Plateau. Cryosphere, 12(2), 657-673. doi:10.5194/tc-12-657-2018.

Qin, Y., Wu, T., Zhao, L., et al. (2017a). Numerical Modeling of the Active Layer Thickness and Permafrost Thermal State Across Qinghai-Tibetan Plateau. Journal of Geophysical Research: Atmospheres, 122(21), 11,604-611,620. doi:https://doi.org/10.1002/2017JD026858.

Qin, Y., Yang, D. W., Gao, B., et al. (2017b). Impacts of climate warming on the frozen

ground and eco-hydrology in the Yellow River source region, China. *Science of The Total Environment*, 605, 830-841. doi:10.1016/j.scitotenv.2017.06.188.

Song, L., Wang, L., Li, X., et al. (2020). Improving Permafrost Physics in a Distributed Cryosphere-Hydrology Model and Its Evaluations at the Upper Yellow River Basin. *Journal of Geophysical Research-Atmospheres*, 125(18). doi:10.1029/2020jd032916.

Song, L., Wang, L., Zhou, J., et al. (2022). Divergent runoff impacts of permafrost and seasonally frozen ground at a large river basin of Tibetan Plateau during 1960–2019. *Environmental Research Letters*, 17(12), 124038. doi:10.1088/1748-9326/aca4eb.

Wang, T., Yang, D., Yang, Y., et al. (2023a). Pervasive Permafrost Thaw Exacerbates Future Risk of Water Shortage Across the Tibetan Plateau. *Earth's Future*, 11(10), e2022EF003463. doi:https://doi.org/10.1029/2022EF003463.

Wang, T., Yang, D., Yang, Y., et al. (2023b). Unsustainable water supply from thawing permafrost on the Tibetan Plateau in a changing climate. *Science Bulletin*, 68(11), 1105-1108. doi:https://doi.org/10.1016/j.scib.2023.04.037.

Yang, K., Wang, C., & Li, S. (2018). Improved Simulation of Frozen-Thawing Process in Land Surface Model (CLM4.5). *Journal of Geophysical Research: Atmospheres*, 123(23), 13,238-213,258. doi:https://doi.org/10.1029/2017JD028260.

---

## Author Response (AR2)

**Response to RC1:**

**General comments**

**Comments 1:** Most of my concerns from the original submission have been addressed satisfactorily. The current manuscript is good work, and I enjoyed reading it. I recommend that this manuscript be considered for publication after minor revisions to make some small but important improvements that would greatly improve the clarity of the manuscript.

**Response 1:** Thank you for your positive feedback and for recognizing our efforts in improving the manuscript. We appreciate your constructive comments, which have helped enhance its clarity and quality. Below, we provide detailed responses to your specific suggestions and outline the corresponding revisions to further improve clarity and precision.

**Specific comments**

**Comments 2:** Your response 11: You write that "This approach allowed the integration of the SNOW17 model with the GXAJ model to form the GXAJ-S model for calculating snowmelt runoff in grid cells, ensuring that no new parameters were added to the GXAJ-S model compared to the GXAJ model.". Would it not be clearer if you expressed this as "compared to the GXAJ model, no new parameters were added to the GXAJ- component of the GXAJ-S model"?

This would imply that all additional complexity in GXAJ-S is compartmentalized in the -S part only, which is my understanding of your work.

**Response 2:** Thank you for your insightful suggestion. We appreciate this clarification, as it more precisely conveys that the additional complexity in the GXAJ-S model is confined to the -S component, while the GXAJ component remains unchanged in terms of parameters. We have incorporated this revision into the manuscript (see lines 404–406):

 "*This approach allowed the integration of the SNOW17 model with the GXAJ model to form the GXAJ-S model for calculating snowmelt runoff in grid cells. **Compared to the GXAJ model, no new parameters were added to the GXAJ component of the GXAJ-S model.***"

**Comments 3:** Your response 10: It is still not clear to me which parameters you calibrated in your study, and which you used a fixed value for. In L384-386, you state that SNOW17 has 4 parameters that must be calibrated, and 6 that do not have to be. However, in Table 2 you list a "prior range" for 9 of these parameters, with the word "prior" implying that you calibrated all 9 of these to yield a posterior value. In Table 1 it appears that all of the listed parameters are calibrated because you use the phrase "prior estimate". Please indicate in Table 1 and Table 2 which parameters you calibrated from the field data, and which parameters you used certain values for without any calibration.

**Response 3:** We sincerely appreciate the reviewer's insightful comments regarding parameter calibration and fixed values in Tables 1 and 2. To improve clarity, we have explicitly distinguished parameters obtained from external sources (e.g., literature, remote sensing datasets, or soil properties) from those calibrated in our study. Specifically, we have:

**Revised Table 1** to indicate which GXAJ model parameters were estimated from external sources and which were calibrated.

**Updated Table 2** to clearly label SNOW17 parameters as either 'calibrated' or 'fixed value.'

*Table 1. GXAJ model parameters and their descriptions.*

| Module | Parameter | Description | Source or Calibration |
|---|---|---|---|
| Canopy interception | $LAI_{max}$ | Maximum LAI for the vegetation in a year | Derived from LDAS based on vegetation types |
| | $h_{lc}$ | Height of vegetation (m) | Derived from LDAS based on vegetation types |
| Channel precipitation | $W_{ch}$ | Channel width within a cell (km) | Estimated based on measured cross sections |
| Evapotranspiration | $W_{UM}$ | Tension water capacity of upper layer (mm) | Estimated based on initial $W_M$ |
| | $W_{LM}$ | Tension water capacity of lower layer (mm) | Estimated based on initial $W_M$ |
| | $C$ | Evapotranspiration coefficient of deeper layer | Estimated based on LAI and $h_{lc}$ of vegetation |
| | $K$ | Ratio of potential evapotranspiration to pan evaporation | Calibrated (prior range: 0 – 1) |
| Runoff generation | $W_M$ | Tension water capacity (mm) | Estimated using $\theta_{fc}, \theta_{wp}$ and vadose zone thickness |
| | $\theta_s$ | Saturated moisture content | Obtained from literature based on soil types |
| | $\theta_{fc}$ | Field capacity | Obtained from literature based on soil types |

| | $\theta_{wp}$ | Wilting point | **Obtained from literature based on soil types** |
| | $S_M$ | Free water capacity (mm) | **Estimated using $\theta_s$, $\theta_{fc}$ and humus layer thickness** |
| | $K_i$ | Outflow coefficient of free water storage to interflow | **Estimated based on soil properties** |
| | $K_g$ | Outflow coefficient of free water storage to groundwater | **Estimated based on soil properties** |
| | $C_i$ | Recession constant of interflow storage | **Calibrated (prior range: 0 – 1)** |
| | $C_g$ | Recession constant of groundwater storage | **Calibrated (prior range: 0 – 1)** |
| Flow routing | $C_s$ | Recession constant in the lag and route technique | **Calibrated (prior range: 0 – 1)** |
| | $L_{ag}$ | Lag time | **Calibrated (prior range: ≥0)** |

**Table 2.** *SNOW17 model parameters and their descriptions.*

| | Parameter | Description | Calibration or Fixed Value |
|---|---|---|---|
| Major parameters | SCF | Snow correction factor, or gage catch deficiency adjustment factor | 0.7 - 1.6 (calibrated) |
| | MFMAX | Maximum solar melt factor during non-rain periods, assumed to occur on June 21 (mm·°C$^{-1}$·6hr$^{-1}$) | 0.5 - 2.0 (calibrated) |
| | MFMIN | Minimum solar melt factor during non-rain periods, assumed to occur on December 21 (mm·°C$^{-1}$·6hr$^{-1}$) | 0.05 - 0.49 (calibrated) |
| | UADJ | The average wind function during rain-on-snow periods (mm·mb$^{-1}$) | 0.03 - 0.19 (calibrated) |
| | NMF | Maximum negative melt factor (mm·mb$^{-1}$·6hr$^{-1}$) | 0.45 (fixed value) |
| | TIPM | Antecedent temperature index parameter | 0.9 (fixed value) |
| | PXTEMP | The temperature that separates rain from snow (°C) | 0 (fixed value) |
| Minor parameters | MBASE | Base temperature for snowmelt computations during non-rain periods (°C) | 0 (fixed value) |
| | PLWHC | Percent liquid water holding capacity for ripe snow (decimal fraction) | 0.1 (fixed value) |
| | DAYGM | Constant daily amount of melt which takes place at the snow-soil interface whenever there is a snow cover (mm·day$^{-1}$) | 0.7 (fixed value) |

These modifications ensure transparency in parameter selection and calibration. We appreciate the reviewer's suggestion, which has helped enhance the clarity of our methodology.

**Comments 4:** Model comparison: I suggest including a table with the computational time (including both the calibration time and actual simulation time) of simulating a comparable scenario for all 3 models GXAJ, GXAJ-S, and GXAJ-S-SF. That would provide readers with a more complete information on which model to choose. The additional physical detail of GXAJ-SSF may not be necessary in some applications that prioritize fast computation over accuracy.

**Response 4:** Thank you for your valuable suggestion. We have included a table in the revised supplementary material (Table S1) comparing the computational time of GXAJ, GXAJ-S, and GXAJ-S-SF under a comparable scenario. This table reports both calibration and simulation times, providing readers with a clearer understanding of the trade-off between computational efficiency and model complexity.

We have also described this addition in the revised manuscript (Lines 574–578). Furthermore, we have specified the computing environment, including processor details, memory, operating system, programming language, and the number of calibration iterations. This ensures transparency and helps users make informed decisions based on their computational resources and modeling needs. Below is the added content:

"*To provide a more comprehensive comparison of the three models, we have included an evaluation of computational efficiency. Table S1 presents the calibration and simulation times for GXAJ, GXAJ-S, and GXAJ-S-SF. The results indicate that while GXAJ-S-SF provides improved physical representation, it requires longer computation time compared to GXAJ and GXAJ-S. This information is useful for users who may prioritize efficiency over accuracy in certain applications.*

*Please let us know if any further details are needed.*

**Table S1.** *Computational time comparison for the GXAJ, GXAJ-S, and GXAJ-S-SF models.*

| Model | Calibration Time (hours) | Simulation Time (seconds) |
|---|---|---|
| GXAJ | 1.5 | 9 |
| GXAJ-S | 6.1 | 39 |
| GXAJ-S-SF | 6.7 | 41 |

**# All simulations were conducted in the following computing environment:** *AMD Ryzen 5 3600X 6-Core Processor, 32GB DDR4 2133MHz RAM, Windows 10 operating system, and MATLAB R2023a for model implementation and execution. The computations were performed in single-threaded mode, with 400 iterations set for the calibration period.*"

**Response to RC2:**

**General comments**

**Comments 1:** Thank you for the authors' efforts in improving the manuscript. This version of the paper shows improvement compared to the previous one. However, I still believe that the current paper is not yet suitable for publication in HESS. In my opinion, the novelty of the proposed contributions is insufficient, and the reliability of the conclusions remains inadequate.

**Response 1:** We sincerely appreciate your time and effort in reviewing our manuscript and for providing constructive feedback. We are grateful for your acknowledgment of the improvements made in this revised version. However, we regret that our work has not yet fully met your expectations regarding novelty and the reliability of our conclusions.

To address your concerns, we have undertaken substantial and comprehensive revisions throughout the manuscript (include Abstract, Introduction, Results, Discussion and Conclusions). These include enhanced validation using multi-source data, deeper interpretation of hydrological processes in cold regions, and expanded discussions that place our work in context with existing studies. These revisions aim to better clarify the novelty of our methodology and reinforce the robustness of our conclusions.

We have carefully addressed your concerns regarding novelty and reliability in our detailed responses below. We highly value your insights and would greatly appreciate any further suggestions you may have to help improve the quality of our work. Once again, thank you for your thoughtful critique and for helping us enhance our manuscript.

**Specific comments**

**Comments 2:** The authors emphasize the development of a new hydrological model that considers snow and seasonally frozen ground. However, in my view, the approach of simply coupling empirical formula-based modules into a hydrological model is not sufficiently innovative. Extensive research has already been conducted on this issue in the Tibetan Plateau region, as the authors themselves have also mentioned. Moreover, several existing models, such as the VIC model (Cuo et al., 2015), CLM4.5 (Yang et al., 2018), GIPL2.0 (Qin et al., 2017a), WEB-DHM (Song et al., 2020), and GBEHM (Gao et al., 2018), provide more comprehensive descriptions of snow and seasonally frozen ground modules. The authors repeatedly highlight that the model developed in this study requires fewer input data, thereby emphasizing its applicability in datascarce regions. However, the input data required by this model are essentially the same as those required by the aforementioned physically based models, primarily including topographic data, vegetation data, and meteorological input data. On the contrary, due to the simplified representation of physical processes in the model, more reference data are needed for parameter calibration. Therefore, I do not consider the simplicity of the model's physical description to be an advantage.

**Response 2:** Thank you for your valuable comments. We understand your concerns regarding the innovation and applicability of the model and would like to further clarify the main contributions of this study and the scientific rationale behind the modeling approach adopted. Indeed, we recognize that the advantages of the here considered modelling approach may be context-dependent. Below, we provide a detailed response to your concerns.

➢ **Coupling of Frozen Ground and Hydrological Processes**

Our study does not simply integrate empirical formulas into a hydrological model; rather, it systematically coupled seasonally frozen ground (SFG) processes with key hydrological components. The Stefan equation was used to calculate the spatiotemporal distribution of frozen depth, which directly influenced soil moisture/ice content in the vadose zone. This in turn changed the effective thickness of the vadose zone and humus soil layer (including effective tension water storage capacity and free water storage capacity), ultimately affecting multiple hydrological processes such as runoff generation, runoff distribution and evapotranspiration (see Section 2.2). The freeze-thaw process of frozen soil is affected by snow conditions. The improved model in this study takes these hydrological physical processes into account.

➢ **Comparison with Physically Based Models**

Indeed, we recognize that the advantages of the here considered modelling approach may be context-dependent. Considering the Yalong River basin case, we therefore now include an in-depth comparison of the performance of the investigated simplified (relative) models with the performance of physically based models. In data-limited regions such as the Yalong River basin, physical models may rely on data that are not available through direct measurements, such as ground temperature. This complicates parameterization processes and introduces uncertainties in the results. This hence motivates our refined investigation example (see below) regarding how physical models perform in comparison with simplified models.

To investigate this issue, we referenced the application of the VIC model and SWAT model in our
140    study area from 2007 to 2011 (Li et al., 2018) and compared it with our proposed model. The
results show that the simulation accuracy of the VIC model (NSE = 0.75 during calibration and
NSE = 0.65 during validation) and the SWAT model (NSE = 0.77 during calibration and NSE =
0.66 during validation) did not exceed that of our model over the same period (NSE = 0.87 during
calibration and NSE = 0.74 during validation). This may hence be related to the uncertainties
145    introduced by the parameterization of physical models in data limited regions, and suggests the
need to expand observational efforts before expanding modelling efforts to further improve
predictive capacity (see Discussion, Lines 649–668).

*"Although significant progress has been made in physical models that account for snow and
freeze-thaw processes, their application in cold-region hydrology remains challenging. Due to*
150    *the complex topography, heterogeneous vegetation cover, and uneven soil moisture distribution
in cold regions, uncertainties in radiation and surface albedo estimation can lead to
inaccuracies in surface energy balance simulations, introducing errors in ground temperature
and soil heat flux estimations (Gao et al., 2018). Additionally, the spatial parameterization of
physical models remains a significant challenge, and their structural and parameterization*
155    *schemes require further refinement (Zhou et al., 2021). The diverse climatic and geographic
conditions in cold regions further limit the applicability of many physical models across different
study areas (Yong et al., 2023). Moreover, the complexity and uncertainty of cold-region
hydrological processes increase the difficulty of model development and parameter calibration,
which may negatively impact simulation accuracy (Gao et al., 2018; Qin et al., 2017). To further*
160    *assess the performance of physical models in our study area, we compared the VIC model's
simulation results from 2007 to 2011 (Li et al., 2018b) with those obtained using our simplified
model. The results indicate that the VIC model exhibited NSE values of 0.75 and 0.65 for the
calibration and validation periods, respectively, which did not exceed those of our model (0.87
for calibration and 0.74 for validation). This comparison illustrates that the data limitations in*
165    *the Yalong River basin are likely to currently constrain the performance of physically based
models. This hence suggests the need to expand observational efforts before expanding
modelling efforts to further improve predictive capacity.."*

➢ **Parameterization process**

We understand that the physical model can provide more explanations, and then its parameterization scheme is still a huge challenge. The enhanced model developed in this study integrates multiple key cold region hydrological processes while maintaining low parameter complexity (Section 2.2.3 Model parameters and calibration), making it particularly suitable for cold regions with complex hydrological and meteorological conditions and scarce data such as the Yalong River Basin.

**➤ Scientific Contribution and Innovation**

The novelty of this study lies not only in the coupling of snow, frozen ground, and hydrological processes (as mentioned above), but also in (see revised '*Abstract*', '*Results*' and '*Conclusion*' sections):

- Providing a quantitative analysis of the impact of snow/frozen ground on runoff partitioning and evapotranspiration (Section 3.3 '*Model differences in simulated runoff components and soil evapotranspiration*' and 4.2 '*The impact of seasonal frozen ground/snow on hydrological processes*').

- Demonstrating the complex interactions among snow cover, frozen ground, and the unsaturated zone (Section 3.1 '*Simulation of snow accumulation and freeze-thaw process*' and 4.2 '*The impact of seasonal frozen ground/snow on hydrological processes*').

- Offering a flexible and adaptable modeling framework that can be seamlessly integrated into hydrological models beyond GXAJ.

In summary, the enhanced modeling framework proposed in this study improves runoff simulations while providing new insights into the role of snow and frozen ground in shaping water balance components. The comparison with both the studied model set and more complex physically based models suggests that data limitations in the Yalong River Basin may currently constrain the performance of physically based models. This highlights the need for expanded observational efforts to improve predictive capabilities before extending physically based modeling approaches.

Thank you for your time and consideration.

**Comments 3:** Additionally, the authors primarily demonstrate the model's accuracy through the performance of streamflow simulations. However, this is far from sufficient for a study on hydrological processes in high-mountain basins, where multiple processes contribute to the overall

dynamics. Given that the focus of the paper is on analyzing the impacts of snow and frozen ground on streamflow, detailed validation of these two critical intermediate processes is essential. However, such validation is currently lacking. For the snow module, the authors only use remotely sensed snow depth data for calibration and validation. However, the accuracy of these data remains uncertain, as it is well known that remote sensing of snow depth in the complex terrain of the Tibetan Plateau is subject to significant uncertainties. I recommend that the authors use more authoritative MODIS snow cover data to conduct a more comprehensive validation of the snow module results. As for the frozen ground module, the current validation is mainly limited to the start dates of freeze-thaw cycles (the results do not appear to be very satisfactory, and the authors have not provided quantitative metrics). There is a notable lack of validation for key physical variables, such as soil temperature and soil moisture (both ice and liquid water). I suggest that the authors collect in-situ measurements from the study region or validate their results against more authoritative remote sensing or reanalysis soil data to enhance the reliability of their findings.

**Response 3:** Thank you for your valuable comments on this study. In response to your concerns regarding the insufficient validation of the snow and frozen ground modules, we have made further improvements and validation efforts based on your suggestions:

➢ **Snow Module Validation:**

We acknowledge that remote sensing snow depth data may contain uncertainties, particularly in complex terrain. However, previous studies have specifically evaluated the dataset used in this study for the Yalong River Basin, demonstrating its high accuracy (Wu et al., 2024), which suggests that the model errors should be relatively low. To further enhance the validation, we have compared MODIS snow cover data with our model simulations. The results indicate that snow cover extended over up to half of the study area, with a high correlation coefficient (0.91) between the simulated and observed daily snow cover fractions. Figure S8 presents the spatial distribution of simulated snow depth alongside MODIS-derived snow cover on December 1, 2015, demonstrating strong consistency in coverage patterns (lines 669-701).

"*In complex mountainous cold regions, observation remains a bottleneck (Gao et al., 2022). Due to limitations in measured data on frozen soil and snow depth in the considered Yalong River basin, this study used multi-source remote sensing data and reanalysis data for calibration and verification from multiple perspectives. In particular, errors in remote sensing snow depth data (Yan et al., 2022; Zou et al., 2014) can propagate to the model output. However, previous studies*

*have specifically investigated the here used remote sensing dataset for the Yalong River basin showing that its accuracy is high (Wu et al., 2024), which suggests that model errors should be relatively low. This study further compared MODIS snow cover data with model simulations, revealing that snow cover extended over up to half of the study area, with daily snow cover fraction exhibiting a high correlation coefficient of 0.91 between the two datasets. Figure S8 illustrates the spatial distribution of simulated snow depth and MODIS-derived snow cover on December 1, 2015, demonstrating strong consistency in coverage patterns. We also recognize that the use of surface/soil temperature and maximum frozen ground depth to verify the freeze-thaw process introduces some uncertainty (Li et al., 2022). Since the GXAJ-S and GXAJ-S-SF model variants used the same temperature, snow and frozen ground data in the present simulations, they can be expected to share similar data errors, However, due the non-linear nature of the modeled processes, such data errors may still not cancel completely when comparing different models. Nevertheless, observed differences in model performance between these models are mainly expected to reflect differences in model capabilities rather than differences in input datasets. Future work should focus on improving remote sensing data quality and exploring the long-term robustness of the model to further enhance performance and improve our understanding of the freeze-thaw processes in complex mountainous cold regions.*

*Hydrological modeling typically prioritizes model fitness, which in theory can be improved by introducing more fitting parameters. However, this study highlights differences that are due to addition of process-based modules (regarding snow and frozen ground). This implies that improvements in model fit and differences in associated model output (e.g. runoff and evapotranspiration) reflect how the considered snow and/ or frozen ground processes more concretely alter hydrological flows. This therefore increases the understanding of underlying hydrological processes (Gao et al., 2022) in large-scale applications such as the Yalong River basin that additionally has a complex topography with large elevation differences yielding high spatio-temporal heterogeneity in snowmelt and freeze-thaw cycles of soil.*"

➢ **Frozen Ground Module Validation:**

For the frozen ground module, we have further refined the validation process. In addition to validating the start dates of the freeze-thaw cycles and providing quantitative metrics, we have also incorporated ERA5 soil temperature data and spatial distribution data of maximum frozen soil depth. The comparison of these data shows that the simulated frozen ground depth from the model

aligns well with the remote sensing/reanalysis data in terms of both time series and spatial distribution, with detailed results provided in lines 476-497.

"*This study systematically validated the simulation results of frozen soil depth based on the Stefan empirical formula through multi-source data comparison. Fig. 5 presents the frozen depth derived from ERA5 reanalysis data using four soil temperature layers (0–7 cm, 7–28 cm, 28–100 cm, and 100–289 cm; freezing occurs when layer temperatures fall below 0°C). The seasonal freeze-thaw depths calculated by the Stefan formula exhibit high consistency with ERA5-derived results in both freeze-thaw timing and variation trends. Notably, the ERA5-based frozen depths display a stepwise variation pattern, with the maximum freezing depth terminating at the 100 cm layer, likely attributable to the freezing inhibition effect caused by higher temperatures in the deep soil layer (100–289 cm). The simulations indicate that the freezing process initiates in late September, reaches the maximum depth of 1.4 m by late March of the following year, and completes thawing by late May. This temporal pattern aligns closely with ground temperature observations from basin meteorological stations (Fig. S6; mean errors of ≤5 days for initial freezing dates and ≤10 days for initial thawing dates).*

*To further evaluate the model's spatial performance, the 2000–2018 mean maximum frozen depth distribution was compared with contemporaneous data from the National Tibetan Plateau Data Center (Table 3; Fig. S7). The Stefan formula-based simulations, incorporating station-based temperature interpolation, demonstrate smoother spatial transitions—a characteristic linked to model parameterization. Both datasets reveal a gradient pattern of deeper frozen depths in upstream valley regions and shallower depths in downstream areas, with a spatial correlation coefficient of 0.89. Furthermore, the observed decreasing trend in frozen depth during 2000–2018 corresponds with accelerated snowmelt patterns (Fig. 4), highlighting the coupled response of the cryosphere to climate change.*"

➢ **Data Sources and Validation Reliability:**

Due to the lack of in-situ measurements in our study area, we have employed multiple remote sensing and reanalysis datasets to validate the snow and frozen ground processes from various perspectives. This multi-source validation approach strengthens the reliability of the model results while accounting for potential uncertainties from different data sources. We believe this significantly enhances the credibility of our findings. In future work, we will continue to improve the quality of remote sensing data and assess the long-term stability of the model.

We appreciate your constructive feedback, which has helped us further refine the study. The manuscript has been revised accordingly, and we believe these improvements contribute to the robustness and clarity of our findings.

**Comments 4:** Finally, numerous studies have already investigated the hydrological effects of snow and frozen ground at large basin scales, ranging from basin-scale (e.g., Cuo et al., 2015; Qin et al, 2017b; Song et al., 2022; Wang et al., 2023a, 2023b) to the entire Tibetan Plateau. Therefore, I recommend that the authors compare some of their conclusions with those of previous studies, rather than simply stating that research in this area is lacking. Furthermore, I find the current conclusions to be insufficiently in-depth. For example, the critical processes of how changes in soil ice and liquid water in frozen ground affect streamflow are not thoroughly discussed.

**Response 4:** Thank you for your thoughtful comments. We appreciate your suggestions regarding the need for comparison with previous studies and a more in-depth discussion of key processes.

To address this, we have made substantial revisions to the '*Discussion*' section, where we now explicitly compare our refined conclusions with existing studies. This comparison highlights both the similarities and differences between our study and previous research, thereby strengthening the contextual relevance of our findings.

Additionally, we have integrated the refined findings into the revised '*Conclusion*' section. This includes the key findings described in the '*Results*' and '*Discussion*' regarding the influence of snow depth on frozen ground depth and duration, the seasonal impact of freeze-thaw cycles on runoff generation, the suppressive effect of snow and frozen ground on evapotranspiration during cold months, the seasonal role of snowmelt, and key findings regarding the future hydrological significance of snow and frozen ground. These comparisons aim to position our results within a broader literature context while emphasizing both the consistency with previous studies and the novel insights provided by our research.

---

## Author Response (AR3)

**Response to RC1:**

**General comments 1:**

The issues that I raised in the second round of review have been appropriately addressed. However, I agree with reviewer #2 that the novelty of this manuscript appears marginal, which I also mentioned during the first round of review. Perhaps this study is more likely to generate regional interest rather than general interest, given its large dependence on empirical methods and lack of novel generalizable physical insight. Nevertheless, the methodology presented in this manuscript may provide an alternative modeling method to existing ones, and its relative physical simplicity could be an advantage for specific cases if the benefits this simplicity are mechanistically justified and made clearer in the manuscript. Ultimately, I am undecided about whether this manuscript has a sufficient level of novelty or potential impact for HESS. Hence, I recommend minor revisions so that the authors may further clarify in the manuscript the value added by this study to the hydrological community.

**Response 1:**

We sincerely thank the reviewer for confirming that the concerns raised during the second round of review have been appropriately addressed, and for providing constructive feedback on the novelty and broader relevance of the study. In response, we have carefully revised the manuscript to further highlight the value of this work to the wider hydrological community and clarify its scientific contributions. See in particular our below responses to General comments 2 and 3, through which we now included key clarifications on how our study adds scientific insights for the hydrological community, beyond the regional perspective. In addition to these clarifications in the introduction (just before stating the objectives) and the discussion sections, we accordingly re-formulated (i) the abstract, (ii) other parts of the introduction (e.g., further explaining the general importance of understanding seasonally frozen ground (SFG) in large basins), (iii) methods, (iv) results and (v) conclusions (e.g., clarifying the modelling framework transferability and quantitative process-level insights of significance also beyond the study region).

**General comments 2:**

As I also mentioned in the first round of review, the physical insights that the study generates are not particularly novel, though it may be quantitatively more accurate than competing models in certain circumstances (e.g. the case study presented in the manuscript). A key issue with the manuscript is that the mechanistic reasons for why the predictions made by this model are good, or even better than other models, are not sufficiently explained. Given the mostly empirical nature of the model, it is possible that the good results may have arisen due to statistical fluke or overfitting to additional degrees of freedom. Hence, it is not clear whether the modeling methods introduced in this manuscript are generalizable to other regions or scenarios, and whether they represent a superior alternative over other existing models. It is recommended that the authors provide a convincing mechanistic explanation of why their fitting results (NSE) were superior to VIC and SWAT. Providing such an explanation would also help to allay concerns that some of the design choices of the introduced model seem to be rather arbitrary and non-generalizable, as I mentioned during the first round of review (e.g. my comment #6 regarding discrete soil layering in GXAJ).

**Response 2:** We thank the reviewer for raising this important point. In this study, the developed model integrates physically based processes (such as snow accumulation and melt, freeze–thaw dynamics, and soil water storage) with empirical simplifications to reduce parameter complexity and the reliance on extensive input data. We agree with the reviewer that the manuscript has retained a vagueness in explaining its novelty and main contributions. In response, we now clarify (in the introduction section) that a main novel aspect of the manuscript is how additional processes are accounted for in a three-step manner by a modular model design (one module per process; with the snow and frozen ground modules being grounded in well-established physical principles, e.g., SNOW17-based snowmelt equations and Stefan-based frost depth estimation). This allows for increasing the complexity while transparently checking the model performance of each step. In particular, any potential increases in model performance are then related to the dynamics created by the additional module (and the corresponding account for a new process).

In the introduction we also explain that, to the best of our knowledge, this has not been done earlier in large cold region basins. This is because previous comparisons have regarded models that differ in either structure (Gao et al., 2018; Li et al., 2018; Song et al., 2022), or structure as well as complexity (e.g., Ahmed et al., 2022; Gao et al., 2018; Guo et al., 2022) . In both cases, differences in model performance may then partly

55   be due to fundamental structural or parametrization differences between models, introducing uncertainty in how performance may be linked to complexity (i.e., inclusion or omission of processes). Therefore, we believe that the current approach, which greatly reduces such structural uncertainty effects by study design, is useful and of general scientific interest in advancing the process understanding and prediction of large-basin runoff in cold regions, in addition to the presented insights for the considered basin.

60   As previously stated, classical physically based hydrological models, such as VIC, GBEHM, and WEB-DHM, provide detailed representations of hydrological processes. Their complex structures and numerous parameterization schemes often require large amounts of ground-based measurements (e.g., soil hydraulic properties, vegetation structure, snow thermal conductivity), which are rarely available in cold and data-scarce regions. This limitation can result in large simulation uncertainties. In the first part of the discussion

65   (section 4.1), we now additionally clarify that, by comparison, our three-step approach implies that a limited number of additional parameters are introduced in each performance evaluation step, which enables the identification of well-functioning levels of model complexity while involving only a small number of parameters - five in the original GXAJ model and four in the snow module. This greatly reduces the risk of overfitting. As further discussed in section 4.1, we have also considered the risk of coincidental good

70   performance by potentially overfitted models by evaluating in which way the addition of process-based modules alters the model behavior in multiple sub-catchments and over multiple seasons. We could then for instance see that, rather than increasing the sub-catchment and seasonal performance in random ways, the addition of the snow and SFG modules specifically increased cold-season performance in low-temperature (high-altitude) parts of the study area, which is consistent with the expected effects of the considered

75   processes. This hence provides a logical explanation that helps readers understand (as asked-for by the reviewer) why the simulation performance demonstrated in our case study was strong (e.g., with high NSE) despite being based on few parameters as compared with e.g. VIC and SWAT applications.

80

**General comments 3:**

Furthermore, although the authors claim that the simplicity of the introduced GXAJ-based methods makes it advantageous compared to physical process-based models in geologically or topographically complex cold-region scenarios, the GXAJ-based methods appear to suffer from the same limitations, as discussed by the authors themselves in lines 769-784. Hence, the authors claim that "In data-limited regions such as the Yalong River basin, physical models may rely on data that are not available through direct measurements, such as ground temperature. This complicates parameterization processes and introduces uncertainties in the results." should be further explained and justified.

**Response 3:** We thank the reviewer for raising this important point regarding our claim about the practicality of GXAJ-based methods in data-scarce cold regions. We aimed at addressing this issue together with our addressing of the "General comment 2", see in particular our above answer in the second paragraph, starting with "*We now additionally explain that, by comparison, our three-step approach implies that a limited number of additional parameters are introduced in each performance evaluation step, which enables the identification of well-functioning levels of model complexity while involving only a small number of parameters - five in the original GXAJ model and four in the snow module*". This aspect hence contributes to explaining why the GXAJ model and its modular extensions are less prone to suffer from such limitations. This is further justified by the fact that we explicitly check that the addition of the snow and SFG modules resulted in a model behavior that is consistent with the expected effects of the considered processes (e.g., increased cold-season performance in low-temperature (high-altitude) parts of the study area) despite the data limitation issues. This is now also explained and justified in the beginning of the discussion (section 4.1), enhancing also the discussion of the generality of the approach in lines 795–817 of the revised manuscript.

Once again, we sincerely thank the reviewer for the constructive comment regarding the scientific value and applicability of the manuscript. This provided us with an excellent opportunity to clarify our reasoning, refine the discussion, and more clearly articulate the novelty and practical significance of our modeling approach in cold and data-scarce regions. We hope the revised manuscript better reflects these improvements.

**References**

110     Ahmed, N., Wang, G., Booij, M. J., Marhaento, H., Pordhan, F. A., Ali, S., Munir, S., and Hashmi, M. Z.-R.: Variations in hydrological variables using distributed hydrological model in permafrost environment, Ecol. Indic., 145, 109609, https://doi.org/10.1016/j.ecolind.2022.109609, 2022.

Gao, B., Yang, D., Qin, Y., Wang, Y., Li, H., Zhang, Y., and Zhang, T.: Change in frozen soils and its effect on regional hydrology, upper Heihe basin, northeastern Qinghai-Tibetan Plateau,
115     Cryosphere, 12, 657–673, https://doi.org/10.5194/tc-12-657-2018, 2018.

Guo, L., Huang, K., Wang, G., and Lin, S.: Development and evaluation of temperature-induced variable source area runoff generation model, J. Hydrol., 610, 127894, https://doi.org/10.1016/j.jhydrol.2022.127894, 2022.

Li, Z., Yu, J., Xu, X., Sun, W., Pang, B., and Yue, J.: Multi-model ensemble hydrological simulation
120     using a BP Neural Network for the upper Yalongjiang River Basin, China, Innovative Water Resources Management - Understanding and Balancing Interactions Between Humankind and Nature, 8th International Water Resources Management Conference of ICWRS, Gottingen, Web of Science ID: WOS:000459240300045, 335–341, https://doi.org/10.5194/piahs-379-335-2018, 2018.

125     Song, L., Wang, L., Zhou, J., Luo, D., and Li, X.: Divergent runoff impacts of permafrost and seasonally frozen ground at a large river basin of Tibetan Plateau during 1960-2019, Environ. Res. Lett., 17, 124038, https://doi.org/10.1088/1748-9326/aca4eb, 2022.